# Disentangling temporal and population variability in plant root water uptake from stable isotopic analysis: when rooting depth matters in labeling studies

Valentin Couvreur[1]*, Youri Rothfuss[2]*, Félicien Meunier[3], Thierry Bariac[4], Philippe Biron[4], Jean-Louis Durand[5], Patricia Richard[4], and Mathieu Javaux[1,2]

[1]Earth and Life Institute (ELI), Université catholique de Louvain (UCL), Louvain-la-Neuve, 1348, Belgium
[2]Institute of Bio- and Geosciences, IBG-3 Agrosphere, Forschungszentrum Jülich GmbH, Jülich, 52425, Germany
[3]CAVElab - Computational and Applied Vegetation Ecology, Faculty of Bioscience Engineering, Ghent University, Campus Coupure links 653, Gent, 9000, Belgium
[4]Institute of Ecology and Environmental Sciences (IEES) – Paris, UMR 7618, CNRS-Sorbonne Université, Campus AgroParisTech, Thiverval-Grignon, 78850, France
[5]UR P3F (INRA), Lusignan, 86600, France

*Correspondence to*: Valentin Couvreur (valentin.couvreur@uclouvain.be) and Youri Rothfuss (y.rothfuss@fz-juelich.de)

* These authors contributed equally to this work.

**Abstract.** Isotopic labeling techniques have the potential to minimize the uncertainty of plant root water uptake (RWU) profiles estimated through multi-source (statistical) modeling, by artificially enhancing soil water isotopic gradient. On the other end of the modelling continuum, physical models can account for hydrodynamic constraints to RWU if simultaneous soil and plant water status data is available.

In this study, a population of tall fescue (*Festuca arundinacae* cv Soni) was grown in a macro-rhizotron and monitored for a 34-hours long period following the oxygen stable isotopic ($^{18}$O) labeling of deep soil water. Aboveground variables included tiller and leaf water oxygen isotopic compositions ($\delta_{\text{tiller}}$ and $\delta_{\text{leaf}}$) as well as leaf water potential ($\psi_{\text{leaf}}$), relative humidity, and transpiration rate. Belowground profiles of root length density (RLD), soil water content and isotopic composition were also sampled. While there were strong correlations between hydraulic variables as well as between isotopic variables, the experimental results underlined the partial disconnection between temporal dynamics of hydraulic and isotopic variables.

In order to dissect the problem, we reproduced both types of observations with a one-dimensional physical model of water flow in the soil-plant domain, for 60 different realistic RLD profiles. While simulated $\psi_{\text{leaf}}$ followed clear temporal variations with little differences across plants as if they were "on board of the same rollercoaster", simulated

$\delta_{\text{tiller}}$ values within the plant population were rather heterogeneous ("swarm-like") with relatively little temporal
variation and a strong sensitivity to rooting depth. The physical model thus explained the discrepancy between isotopic
and hydraulic observations: the variability captured by $\delta_{\text{tiller}}$ reflected the spatial heterogeneity in rooting depth in the
soil region influenced by the labeling and may not correlate with the temporal dynamics of $\psi_{\text{leaf}}$. In other words, the
strong variations of RWU as deduced from isotopic changes in the tiller water may not translate into significant
variations of leaf water potential value.
For comparison purposes, a Bayesian statistical model was also used to simulate RWU. While they predicted relatively
similar cumulative RWU profiles, the physical model could differentiate spatial from temporal dynamics of the isotopic
composition. An important difference between the two types of RWU models was the ability of the physical model to
simulate the occurrence of hydraulic lift in order to explain concomitant increases of soil water content and isotopic
composition observed overnight above the soil labeling region.
**List of variables with symbols and units**

| 42 | Name | Symbol | Units |
|---|---|---|---|
| 43 | Leaf water potential/head: | $\psi_{\text{leaf}}$ | MPa |
| 44 | Soil water potential/head: | $\psi_{\text{soil}}$ | MPa |
| 45 | Water volumetric mass: | $\rho_{\text{w}}$ | kg m$^{-3}$ |
| 46 | Soil apparent density: | $\rho_b$ | kg m$^{-3}$ |
| 47 | Soil gravimetric water content: | $\theta_{\text{grav}}$ | kg kg$^{-1}$ |
| 48 | Soil volumetric water content: | $\theta$ | m$^3$ m$^{-3}$ |
| 49 | Intensity of water uptake (sink term): | $S$ | d$^{-1}$ |
| 50 | Transpiration rate per unit soil area: | $T$ | m d$^{-1}$ |
| 51 | Air relative humidity | RH | % |
| 52 | Soil horizontal area: | $A_{soil}$ | m$^2$ |
| 53 | Soil layer depth (for each layer): | $z$ | m |
| 54 | Soil layer thickness (for each layer): | $\Delta Z$ | m |
| 55 | Root length (for each soil layer): | $l_{root}$ | m |
| 56 | Relative Root Water Uptake | rRWU | dimensionless |
| 57 | Best run | $br$ | dimensionless |
| 58 | Root Length Density: | RLD | m m$^{-3}$ |
| 59 | Soil water oxygen isotopic composition: | $\delta_{\text{soil}}$ | ‰ |
| 60 | Tiller water oxygen isotopic composition: | $\delta_{\text{tiller}}$ | ‰ |
| 61 | Leaf water oxygen isotopic composition: | $\delta_{\text{leaf}}$ | ‰ |
| 62 | Soil-root system conductance: | $K_{\text{soil-root}}$ | m$^3$ MPa$^{-1}$ s$^{-1}$ |
| 63 | Soil-root radial conductance: | $K_{\text{radial}}$ | m$^3$ MPa$^{-1}$ s$^{-1}$ |
| 64 | Root radial conductivity: | $L_{\text{pr}}$ | m MPa$^{-1}$ s$^{-1}$ |
| 65 | Root axial conductance: | $K_{\text{axial}}$ | m$^3$ MPa$^{-1}$ s$^{-1}$ |
| 66 | Equivalent root axial conductivity: | $k_{\text{axial}}$ | m$^4$ MPa$^{-1}$ s$^{-1}$ |
| 67 | Soil hydraulic conductivity: | $k_{\text{soil}}$ | m$^2$ MPa$^{-1}$ s$^{-1}$ |
| 68 | Saturated soil hydraulic conductivity: | $k_{\text{sat}}$ | m$^2$ MPa$^{-1}$ s$^{-1}$ |
| 69 | Soil hydraulic conductivity parameter | $\lambda$ | dimensionless |

| 70 | Soil relative water content | $S_{e,j}$ | dimensionless |
| 71 | | | |

## 1 Introduction

Since the seminal work of Washburn and Smith [1934] where it was first reported that willow trees did not fractionate hydrogen stable isotopes in a hydroponic water solution during root water uptake (RWU), water stable isotopologues ($^1H^2H^{16}O$ and $^1H_2^{18}O$) have been used as indicators for plant water sources in soils. In their review, Rothfuss and Javaux [2017] reported in the period 2015-2016 about no less than 40 publications in which RWU was retrieved from stable isotopic measurements. Novel measuring techniques (e.g., cavity ring-down spectroscopy – CRDS and off-axis integrated cavity output spectroscopy – ICOS) providing ways for fast and cost-effective water stable isotopic analyses certainly enable and emulate current research in that field. Water stable isotopologues are no longer powerful tracers waiting for technological developments [Yakir and Sternberg, 2000] but are on the verge to be used to their full potential for addressing eco-hydrological research questions and identify processes in the soil-plant-atmosphere continuum [Werner et al., 2012; Dubbert and Werner, 2019; Sprenger et al., 2016].

The isotopic determination of RWU profiles is based on the principle that the isotopic composition of xylem water at the outlet of the root system (i.e., in the first aerial and non-transpiring node of the plant) equals the sum of the product between the soil water isotopic composition and relative contribution to RWU across plant water sources. Results come only with reasonable precision when (i) the soil water isotopic composition depth gradient is strong and monotonic (thus avoiding issues of identifiability) and (ii) the temporal dynamics of RWU and soil water isotopic composition is relatively low. Condition (i) is fulfilled mostly at the surface of the soil, while soil water isotopic composition gradients become usually lower or null with increasing depth (due to the isotopic influence of the groundwater table and increasing dispersion with depth). As illustrated by Oerter and Bowen [2019], the lateral variability of the soil water isotopic composition profiles can become significant in the field and could have great implications on the representability and meaningfulness of isotopic-derived estimate of RWU profiles. Condition (ii) is often neglected but is required due to the instantaneous nature of the sap flow samples.

To overcome these limitations, labeling pulses have been increasingly used in recent works to artificially alter the natural isotopic gradients [e.g., Beyer et al., 2016; Beyer et al., 2018; Grossiord et al., 2014; Jesch et al., 2018; Volkmann et al., 2016b]. However, a precise characterization of the artificial spatial (i.e., lateral and vertical) and temporal distributions of the soil water isotopic composition (driven by e.g., soil isotopic water flow) is crucial. The punctual assessments of the isotopic composition profiles following destructive sampling in the field and subsequent extraction of water in the laboratory might neither be spatially nor temporally representative and can lead to erroneous estimates of RWU profiles [Orlowski et al., 2018; Orlowski et al., 2016a].

The vast majority of isotopic studies use statistical (e.g., Bayesian) modeling to retrieve RWU profile solely from the isotopic composition of water extracted in the soil and the shoot [Rothfuss and Javaux, 2017]. However, when data on soil and plant water status is available, hydraulic modeling tools can also be used to connect different data types in a process-based manner and estimate root water uptake profiles [Passot et al., 2019]. Some of the most simplistic models

use 1-D relative root distribution and plant-scale hydraulic parameters [Sulis et al., 2019], while the most complex rely
on root architectures and root segment permeabilities [Meunier et al., 2017c]. Only a handful of studies coupled
isotopic measurements in plant tissues and soil material with models describing RWU in a mechanistic manner. For
instance, Meunier et al. [2017a] could both locate and quantify the volume of redistributed water by *Lolium multiflorum*
by labeling of the soil with $^{18}$O enriched water under controlled conditions.
Building on the work of Meunier et al. [2017a], the objective of the present study is to (i) model in a physically-based
manner (i.e., by accounting for soil and plant and environmental factors) the temporal dynamics of the isotopic
composition of RWU of a population of *Festuca arundinacae* cv Soni. (tall fescue) during a semi-controlled
experiment following an isotopic labeling of deep soil water, (ii) investigate the implication of the model-to-data fit
quality in terms of meaningfulness of the isotopic information to reconstruct RWU profiles, and finally (iii) confront
the simulated root water uptake profiles with estimations obtained on basis of isotopic information alone (i.e., provided
by a Bayesian mixing model).
**2 Material and methods**
Our experiment consisted in supplying labeled water from the bottom to a macro-rhizotron in which tall fescue was
grown. Data on soil and plant oxygen stable isotopic composition and hydraulic status were monitored for 34 hours.
In the following, the oxygen isotopic composition of water will be expressed in per mil (‰) on the "delta" ($\delta^{18}$O) scale
with respect to the international water standard V-SMOW [Gonfiantini, 1978].
**2.1 Rhizotron experimental setup**
The macro-rhizotron (dimensions: 1.6 m x 1.0 m x 0.2 m, see picture in Appendix A) was placed inside a glasshouse
(INRA Lusignan, France), where it was continuously weighed (KE1500, Mettler-Toledo, resolution: 20 g) to monitor
water effluxes (i.e., bare soil evaporation or evapotranspiration). Underneath the soil compartment and in contact with
it, a water reservoir (height: 0.1 m) filled with gravel acted as water table and allowed the supply of water to the
rhizotron. The rhizotron was equipped with two sets of CS616 time domain reflectometer (TDR) profiles (Campbell
Scientific, USA) with 30 cm long probe rods positioned at six depths (–0.05, –0.10, –0.30, –0.60, – 1.05 and –1.30 m)
and one profile of tensiometers (SMS 2000, SDEC-France) located at four depths (–0.05, –0.10, –0.30, and –0.60 m)
in order to monitor the evolution of soil water volumetric content ($\theta$, in m$^3$ m$^{-3}$) and matric potential ($\psi_{soil}$, in MPa).
Finally, relative humidity (RH, %) was recorded above the vegetation with one humidity and temperature probe
(HMP45D, Vaisala, Finland). The transparent polycarbonate sides (front and back) allowed the daily observations of
root maximal depth. The experimental setup allowed precisely controlling the amount and $\delta^{18}$O of soil input water.
Another important feature was the soil depth (i.e., 1.60 m) which minimized the influence of the water table on
superficial layers water content and $\delta^{18}O$.

**2.2 Soil properties and installation**

The soil substrate originates from the Lp horizon of an agricultural field part of the Observatory of Environment
Research (ORE), INRA Lusignan, France (0°60W, 46°250N) which is classified as District Cambisol (particle size
distribution: sand 15%, silt 65%, clay 20%). Prior installation in the rhizotron, the substrate was sieved at 2 mm and
dried out in an air oven at 110 °C during 48 h to remove most of the residual water. 450 kg of soil was filled in the
rhizotron by 0.10 m increment and compacted in order to reach a dry bulk density value of $\rho_b = 1420$ kg m$^{-3}$. The
closed-form soil water retention curve of van Genuchten [1980] was derived in a previous study by Meunier et al.
[2017a] from synchronous measurements of soil water content and matric potential from saturated to residual water
content (see Appendix B for its hydraulic parameters). It was used to compute the soil water matric potential ($\psi_{soil}$, in
MPa) on basis of volumetric water content data during the present experiment.

**2.3 Experimental protocol**

After installation, the soil was gradually flooded with local water ($\delta^{18}O = -6.8$ ‰) from the bottom reservoir up to the
top of the profile for a period of three days in order to reduce as much as possible the initial lateral and vertical
heterogeneities in water content and $\delta^{18}O$. The tall fescue (*Festuca arundinaceae* cv Soni) was sown at a seeding
density of 3.6 g m$^{-2}$ (which corresponds for the rhizotron surface area of 0.2 m$^2$ to roughly 300 plants) when soil water
content reached 0.25 m$^3$ m$^{-3}$ (corresponding to pF 2.3) at $-0.05$ m, as measured by the soil water sensors, and emerged
12 days later. During a period of 165 day following seeding, the tall fescue cover was exclusively watered from the
reservoir with local water in order to (i) keep the soil bottom layer ($< -1.3$ m) close to water saturation, and to (ii) not
to disrupt the natural soil water $\delta^{18}O$ profile.
166 days after seeding (DaS 166) the following conditions were fulfilled: (i) there was a strong soil water content
gradient between the soil deep [$-1.5$ m, $-1.0$ m] and superficial [$-0.3$ m, 0 m] layers, (ii) the tall fescue roots had
reached a depth of $-1.5$ m (observed through polycarbonate transparent sides). That same day at 17:00, the reservoir's
water was labelled and its $\delta^{18}O$ measured at $+470$ ‰. Soil was sampled before (DaS 166 - 15:45) and after labeling on
DaS 167 - 07:00, DaS 167 - 17:00 and DaS 168 - 05:00 using a 2 cm diameter auger through the transparent
polycarbonate side of the rhizotron on four occasions from the surface down to $-1.3$ m for the determination of soil
gravimetric water content ($\theta_{grav}$, in kg kg$^{-1}$) and oxygen stable isotopic composition ($\delta_{soil}$, in ‰). Gravimetric water
content was then converted to volumetric water content ($\theta = \theta_{grav} * \rho_b / \rho_w$, in m$^3$ m$^{-3}$, where $\rho_b$ is the bulk soil density
and $\rho_w$ is the water density). The hypothesis of a constant value for $\rho_b$ across the reconstructed soil profile was further
validated from the quality of the linear fit (coefficient of determination $R^2 = 1.0$) between the $\theta$ values measured by the
sensors at the six available depths and (–0.05, –0.10, –0.30, –0.60, – 1.05 and –1.30 m) and those computed from $\theta_{grav}$.
On 40 occasions during a 34-hour long period three whole plants were sampled from the vegetation (i.e., 120 plants
were sampled in total from the cover). Each plant's tiller and leaves were pooled into two separate vials. Dead material
as well as the oldest living leaf around each tiller were removed in order not to contaminate tiller samples with
transpiring material [Durand et al., 2007]. In addition, air water vapor was collected from the ambient atmosphere
surrounding the rhizotron. The air was run at a flow rate of $1.5\ \mathrm{l\ min^{-1}}$ through two glass cold traps in series immersed
in a mixture of dry ice and pure ethanol at - 80°C. Water from plant (i.e., tillers and leaves) and soil samples was
extracted by vacuum distillation for 14 to 16 hours depending on the sample mass (e.g., ranging between 18 to 28 g
for soil) at temperatures of 60 and 90°C, respectively. The residual water vapor pressure at the end of each successful
extraction procedure invariably reached $10^{-1}$ mbar. The oxygen isotopic compositions of tiller, leaf, and soil water
(i.e., $\delta_{tiller}$, $\delta_{leaf}$, and $\delta_{soil}$) together with that of atmospheric water vapor ($\delta_{atm}$) were measured with an IRMS (Isoprep
18 - Optima, Fison, Great-Britain, precision accuracy of 0.15 ‰). Finally, leaf water potential ($\psi_{leaf}$, in MPa) was
monitored with a pressure chamber on two leaves per sampled plant, and evapotranspiration rate (in $\mathrm{m\ d^{-1}}$) was derived
from the changes in mass of the rhizotron at the same temporal scale as plant sampling.
Root biomass was determined from the horizontal sampling of soil between the polycarbonate sides using a 2 cm
diameter auger at –0.02, –0.08, –0.10, –0.40, –0.55, –0.70, –0.90, –1.10, and –1.30 m soil depth. Each depth was
sampled once to thrice. Each soil core was washed of soil particles and roots were collected over a 0.2 mm mesh filter,
and dried at 60°C for 48 hours. Finally, Root Length Density (RLD, in m root $(\mathrm{m\ soil})^{-3}$) distribution was determined
from the root dry mass using the specific root length determined by Gonzalez-Dugo et al. [2005] specifically for tall
fescue (95 m $\mathrm{g^{-1}}$). The reader is referred to Appendix C for an overview of the type and timing of the different
destructive measurements during the intensive sampling period.
**2.4 Modeling of RWU and $\delta_{tiller}$**
The experimental setup included about 300 tall fescue plants. In order to limit the computational requirement in the
inverse modelling loop, we only generated 60 virtual root systems whose rooting depths ranged from –1.30 to –1.60
m depth [based on our own observations and those of the literature, e.g., Schulze et al., 1996; Fan et al., 2016] with
the root architecture simulator CRootBox [Schnepf et al., 2018], so that the simulated RLD matched observations (Fig.
1a). In order to reach a total number of virtual plants representative of the number of plants in the experimental setup,
each root system was replicated 5 times, forming a "group". Each group was assumed to occupy one sixtieth of the
total horizontal area, and considered as a "big root" hydraulic network (5 identical plants per "big root") with equivalent
radial and axial hydraulic conductances (thus neglecting architectural aspects but accounting for each group's
respective root length density profile).
The radial soil-root conductance between the bulk soil and each group's ($i$) root surfaces in soil layer $j$ ($K_{radial,j}$, m$^3$
MPa$^{-1}$ d$^{-1}$), as derived by Meunier et al. [2017a], was assumed as variable in time ($t$):

$$K_{radial,i,j}(t) = \frac{2\pi r_{root} \cdot l_{root,i,j} \cdot B_j \cdot L_{pr} \cdot k_{soil,j}(t)}{B_j \cdot k_{soil,j}(t) + r_{root} \cdot L_{pr}} \qquad (1)$$

with $r_{root}$ (m) the root radius, $l_{root,i,j}$ (m) the root length of plants of group $i$ in soil layer $j$, $L_{pr}$ (m MPa$^{-1}$ d$^{-1}$) the root
radial hydraulic conductivity, $k_{soil,j}$ (m$^2$ MPa$^{-1}$ d$^{-1}$) the soil hydraulic conductivity in layer $j$, and $B_j$ (dimensionless) a
geometrical factor simplifying the horizontal dimensions into radial domains between the bulk soil and root surfaces,
as given by Schroeder et al. [2009]:

$$B_j = \frac{2(1-\rho_j)(1+\rho_j)}{2\rho_j^2 \ln \rho_j - \rho_j^2 + 1} \qquad (2)$$

where $\rho$ (dimensionless) represents the ratio of the distance between roots and the root averaged diameter. It can be
deduced from the observed root length density (RLD$_j$, m m$^{-3}$):

$$\rho_j = \frac{\sqrt{\frac{1}{\pi RLD_j}}}{r_{root}} \qquad (3)$$

The soil hydraulic conductivity function of Mualem [1976] and van Genuchten [1980] was used:

$$k_{soil,j}(t) = k_{sat} \cdot S_{e,j}{}^\lambda(t) \left(1 - \left(1 - S_{e,j}^{\frac{1}{m}}\right)^m\right)^2 \qquad (4)$$

where $k_{sat}$ (m$^2$ MPa$^{-1}$ d$^{-1}$), m (dimensionless) and $\lambda$ (dimensionless) are soil hydraulic parameters (with m = 1 − 2/n)
and $S_{e,j}$, the relative water content (dimensionless), is computed from the saturated ($\theta_{sat}$, m$^3$ m$^{-3}$) and residual ($\theta_{res}$, m$^3$
m$^{-3}$) water contents as:

$$S_{e,j} = \frac{\theta_j - \theta_{res}}{\theta_{sat} - \theta_{res}} \qquad (5)$$

Unlike the geometrical parameter $B$, which defines a domain geometry between the bulk soil and roots of the overall
population, the $l_{root}$ term is group specific ($i$) and uses the simulated root length density profiles over an area
corresponding to one sixtieth of the total setup horizontal area:

$$l_{root,i,j} = \frac{\Delta Z_j \cdot A_{soil} \cdot RLD_{i,j}}{60} \qquad (6)$$

with $\Delta Z$ (m) and $A_{soil}$ (m$^2$) the soil layer thickness and horizontal surface area, respectively.
To finalize the connection between root xylem and shoot, axial conductances per root system group ($K_{axial}$, m$^3$ MPa$^{-1}$
d$^{-1}$) were calculated as equivalent "big root" specific axial conductance per root system group ($k_{axial}$, m$^4$ MPa$^{-1}$ d$^{-1}$, to
be optimized by inverse modelling) as:

$$K_{axial,j} = \frac{k_{axial}}{\Delta Z_j} \qquad (7)$$

At each time step, both the total soil-root system conductance ($K_{\text{soil-root}}$, $m^3\ MPa^{-1}\ d^{-1}$) and the standard sink distribution
($SSF$, dimensionless, summing up to 1), were calculated from $K_{\text{radial}}$ and $K_{\text{axial}}$, using the algorithm of Meunier et al.
[2017b]. The variable $SSF$ is the relative distribution of water uptake in each soil layer under vertically homogeneous
soil water potential conditions [Couvreur et al., 2012], and $K_{\text{soil-root}}$ represents the water flow per unit water potential
difference between the $SSF$-averaged bulk soil water potential and the "big leaf" (assuming a negligible stem hydraulic
resistance [Steudle and Peterson, 1998]).
Adding soil hydraulic conductance to the one-dimensional hydraulic model of Couvreur et al. [2014] yields the
following solutions of leaf water potential ($\psi_{\text{leaf}}$, MPa) and water sink terms ($S$, $d^{-1}$) whose formulation approaches that
of Nimah and Hanks [1973]:
$$\psi_{leaf}(t) = -\frac{T(t)}{K_{soil-root}(t)} + \sum \quad SSF_j(t) \cdot \psi_{soil,j}(t) \tag{8}$$
Where one sixtieth of the overall transpiration rate ($T$, $m\ d^{-1}$) is allocated to each group, and $\psi_{\text{soil,j}}$ (Mpa) is the soil
water potential in soil layer $j$.
$$S_{i,j}(t) = \frac{K_{soil-root,i}(t) \cdot SSF_{i,j}(t) \cdot (\psi_{soil,j}(t) - \psi_{leaf,i}(t))}{A_{soil} \cdot \Delta Z_j} \tag{9}$$
where $K_{soil-root}$ was assumed to control the compensatory RWU which arise from a heterogeneously distributed soil
water potential, due to large axial conductances [Couvreur et al., 2012].
Finally, the tiller water oxygen isotopic composition ($\delta_{\text{tiller}}$) was calculated as the average of local soil water oxygen
isotopic compositions ($\delta_{\text{soil}}$) weighted by the relative distribution of positive water uptakes (i.e., not accounting for $\delta_{\text{soil}}$
at locations where water is exuded by the root), assuming a perfect mixture of water inside the root system [Meunier
et al., 2017a]:
$$\delta_{tiller} = \frac{\sum_{S_j>0} S_j \cdot A_{soil} \cdot \Delta Z_j \cdot \delta_{soil}(t)}{\sum_{S_j>0} S_j(t) \cdot A_{soil} \cdot \Delta Z_j} \tag{10}$$
Like in the experiment, $\delta_{\text{tiller}}$ from three plants were randomly pooled at each observation time. A hundred pools of 3
plants (possibly including several plants of the same group) were randomly selected in order to obtain the pooled
simulated $\delta_{\text{tiller}}$ by arithmetic averaging.
The unknown parameters of the soil-root hydraulic model, i.e., the root radial conductivity ($L_{\text{pr}}$), the root axial
conductance ($k_{\text{axial}}$), the soil saturated hydraulic conductivity ($k_{\text{sat}}$), and the soil tortuosity factor ($\lambda$) were finally
determined by inverse modeling. For details on the procedure, the reader is referred to Appendix D.
In order to evaluate the robustness of the hydraulic model predictions (parametrized solely based on the reproduction
of shoot observations in the inverse modeling scheme) from independent perspectives, we also compared predictions
and measurements over 4 quantitative "soil-root domain" criteria: (i) the depth at which the transition between
nighttime water uptake and exudation ($S_{\text{i,j}}<0$, i.e. release of water from root to soil) takes place, (ii) quantities of exuded
water and overnight increase of soil water content, (iii) the enrichment of labelled water at the depth where water
content increase is observed overnight, and (iv) the order of magnitude of the optimal root radial conductivity value as
compared to literature data in tall fescue.
Finally, and as a comparison point, the Bayesian inference statistical model SIAR [Parnell et al., 2013] was used to
determine the profiles of water sink terms of ten identified potential water sources. These water sources were defined
to originate from 10 distinct soil layers (0.00-0.03, 0.03-0.07, 0.07-0.15, 0.15-0.30, 0.30-0.60, 0.60-0.90, 0.90-1.20,
1.20-1.32, 1.32-1.37, and 1.37-1.44 m) for which corresponding $\delta_{soil}$ values were computed [Rothfuss and Javaux,
2017]. SIAR solely bases its estimates from the comparison of $\delta_{tiller}$ observations to the isotopic compositions of the
soil water sources ($\delta_{soil}$). For this, $\delta_{tiller}$ measurements were pooled in twelve groups corresponding to different time
periods, selected to best reflect the observed temporal dynamics of $\delta_{tiller}$. The reader is here referred to Appendix E for
details on the model parametrization and running procedure.
**3 Results and discussion**
**3.1 Experimental data**
**3.1.1 Soil profiles**
Figure 2a and b show a very stable soil water content profile and a more variable $\delta_{soil}$ profile from DaS 166 - 15:45 to
DaS 168 - 05:00. Soil was dry at the surface (0.058 m$^3$ m$^{-3}$ < $\theta$ < 0.092 m$^3$ m$^{-3}$ for layer 0.015 - 0.040 m) whereas
closer to saturation at depth –1.30 m ($\theta$ = 0.34 m$^3$ m$^{-3}$ ± 0.012 m$^3$ m$^{-3}$, estimated $\theta_{sat}$ = 0.40 m$^3$ m$^{-3}$, see Appendix A).
According to the measured soil matric potentials (Fig. 2c), soil water was virtually unavailable ($\leq$ –1.5 MPa) above –
0.5 m depth. Soil moisture remained unchanged in the top 25 cm during the sampling period ($\theta$ = 0.08 ±0.00 m$^3$ m$^{-3}$)
as well as at –1.30 m from DaS 166 - 15:45 to DaS 168 - 05:00 ($\theta$ = 0.33 ±0.01 m$^3$ m$^{-3}$), showing that roots were
predominantly extracting water from deep soil layers.
Water in the top soil layers (–0.040 m < z < –0.015 m) was isotopically enriched (–3.2 ‰ < $\delta_{soil}$ < 0.3 ‰) as opposed
to the deepest layer ($\delta_{soil}$ = –7.34 ‰ ± 0.30 ‰ at –1.30 m). Following labeling of the reservoir water on DaS 166 -
17:00, $\delta_{soil}$ reached a value of 36.9 ‰ at –1.50 m on DaS 167 - 17:00. The development of the vegetation on DaS 166-
168 (LAI = 5.6) and the observed surface $\theta$ values lead us to assume that the rhizotron water losses were due to
transpiration flux solely (i.e., evapotranspiration = transpiration). The soil water oxygen isotopic exponential-shaped
profiles were the product of fractionating evaporation flux, and to a great extent when the soil was bare or when the
tall fescue cover was not fully developed. The differences in soil water oxygen isotopic profile observed at the four
different sampling dates were therefore either due to lateral heterogeneity (e.g., upper soil layers), to the soil capillary
rise of labelled water from the reservoir (deep soil layers), or to the hydraulic redistribution of water through roots (to
the condition that the isotopic composition of the redistributed water differs from that of the soil water at the release

location). We noted an isotopic enrichment of 1.0 ‰ of soil water observed on DaS 168 - 05:00 at –0.9 m with respect to the mean $\delta_{soil}$ value across previous sampling dates. This could partly be due to, e.g., upward preferential flow of labelled water from the bottom soil layers and therefore be the sign of the lateral heterogeneity of the soil. Another reason for this would be hydraulic redistribution of labelled water by the roots. It was however not possible to evaluate the relative importance of these three processes (lateral heterogeneity, capillary rise/preferential flow, and hydraulic redistribution) in the setting of the soil water isotopic profile since the physically-based soil-root model presented in section 2.4 does not account for soil liquid and vapor flow. This was also not the primary intent of the present study. The observed RLD profile (Fig. 1a) showed a typical exponential shape, i.e., maximum at the surface (5.42 ± 0.34 cm cm$^{-3}$) down to a minimum at –1.10 m (0.540 ± 0.35 cm cm$^{-3}$), while it increased again from the latter depth up to a value of 1.660 cm cm$^{-3}$ at –1.30 m. This significant trend was most probably a direct consequence of the high soil water content value in this deeper layer.

### 3.1.2 Plant water and isotopic temporal dynamics

The temporal variation of $\delta_{tiller}$ (Fig. 3a) was found to be either (i) moderate during day and night, i.e., from DaS 167 - 06:00 to 11:00 ($\delta_{tiller} = –2.6 ± 1.4$ ‰) and from DaS 167 - 21:30 to DaS 168 - 00:00 ($\delta_{tiller} = –2.7 ± 0.4$ ‰), or (ii) strong during the day, i.e., from DaS 167 - 11:00 to 18:00 (maximum value of 20.9 ‰ at DaS 167 - 12:40), or else (iii) strong during the night, i.e., from DaS 167 - 04:00 to 06:00 (max = 36.4 ‰ at DaS 167 - 05:15) and from DaS 168 - 00:00 to 06:00 (max = 14.6 ‰ at 28:00, DaS 168). Note that transpiration (Fig. 3b) occurred also at night during the sampling period, due to relatively high temperature in the glasshouse leading to a value of atmospheric relative humidity smaller than 85%, Fig. 3b). From 12:00 to 14:00 and from 16:00 to 17:00 on DaS 167 (case (ii)) high values of leaf transpiration corresponded to high values of $\delta_{tiller}$.

### 3.1.3 Partial decorrelation between water and isotopic state variables

Figure 4 shows that variables describing plant water status, i.e., $T$ and RH (Fig. 4a) and $T$ and $\psi_{leaf}$ (Fig. 4b) were well correlated: coefficient of determination $R^2$ was equal to 0.78 and 0.70 for the entire experimental duration, respectively. However, linear relationships between water status and isotopic variables were either nonexistent, e.g., between $T$ and $\delta_{tiller}$ ($R^2$=0.01, Fig. 4c) and between $\psi_{leaf}$ and $\delta_{tiller}$ ($R^2$=0.00, Fig. 4h) or characterized by a low $R^2$ and high p-value (e.g., between $T$ and $\delta_{leaf}$, $R^2$=0.43, p>0.05, Fig. 4d). The partial temporal disconnection between $\delta_{leaf}$ and $T$ could not be attributed to problems of the isotopic methodology, during e.g., the vacuum distillation of the water from the plant tillers and leaves: water recovery rate was always greater than 99 % and Rayleigh distillation corrections [Dansgaard, 1964; Galewsky et al., 2016] were applied to standardize the observed oxygen isotopic composition values to a 100 % water recovery (based on the comparison of sample weight loss during distillation and mass of collected distilled water). The evolution of $\delta_{leaf}$ was strongly correlated with that of $\delta_{tiller}$ during the day ($R^2 = 0.90$) whereas non-correlated

during the night ($R^2 = 0.00$, Fig. 4j). These observed correlations are in agreement with the Craig and Gordon [1965] model revisited by Dongmann [1974] and later by Farquhar et al. [2007; 2005]. The model, which is extensively used in the current literature [e.g., Dubbert et al., 2017] states that, at isotopic steady-state, $\delta_{leaf}$ is a function of the input water oxygen isotopic composition ($\delta_{tiller}$) among other variables, i.e., leaf temperature (not measured during the experiment), stomatal and boundary layer conductances, oxygen isotopic composition of atmospheric water vapor, and relative humidity.

It is generally difficult to observe a statistically significant $\delta_{leaf}$-$\delta_{tiller}$ (Fig. 4j) relationship at this temporal scale under natural abundance conditions in the field since the soil water isotopic weak gradient translates into weaker $\delta_{tiller}$ temporal dynamics. The quality of linear fit between $\delta_{leaf}$ and $\delta_{tiller}$ data collected during the day ($R^2=0.90$) was made possible in this specific experiment by the artificial isotopic labeling pulse that enhanced the soil water isotopic gradient, which in turn increased the range of variation of $\delta_{tiller}$, ultimately highlighting the $\delta_{leaf}$-$\delta_{tiller}$ temporal correlation. Air relative humidity is a driving variable of $\delta_{leaf}$ in the model of Dongmann [1974] via the competing terms $(1-RH)\cdot\delta_{tiller}$ and $RH\cdot\delta_{atm}$, where $\delta_{atm}$ is the atmospheric water vapor isotopic composition inside the glasshouse. An overall significant linear correlation was observed between RH and $\delta_{leaf}$ during the experiment ($R^2=0.57$, Fig. 4g). During the two night periods (i.e., from 04:00-06:00 and from 20:30-07:00), as relative humidity increased in the glasshouse (51 % < RH < 85 %, Fig. 3b), the influence of the isotopic labeling of the tiller water (due to the labeling of deep soil water) through term $(1-RH)\cdot\delta_{tiller}$ decreased to the benefit of term $RH\cdot\delta_{atm}$ (with $\delta_{atm}$ values ranging from $-15.9$ to $-10.7$ ‰, mean $= -13.1\pm1.6$‰, data not shown). This was especially visible between 04:50 and 06:00 on DaS 167 and between 01:00 to 06:00 on DaS 168, when $\delta_{tiller}$ reached greater values than $\delta_{leaf}$.

From a different perspective, as three plant water samples were pooled to reach a workable volume for the isotopic analysis at each observation time without replicates, the isotopic signal fluctuations may reflect both its temporal dynamics and its variability within the plant population.

**3.2 Simulations**

**3.2.1 Rooting depth and transpiration rate control $\delta_{tiller}$ and $\psi_{leaf}$ fluctuations, respectively**

Despite the use of a global optimizer and 4 degrees of freedom ($L_{pr}$, $k_{axial}$, $k_{sat}$, $\lambda$, see optimal values in Table 1) specifically aiming at matching the simulated and observed temporal dynamics of $\delta_{tiller}$, none of the 60 root system groups or average population could reproduce the measured fluctuations in time ($R^2=0.00$, Fig. 5a), regardless of the weight attributed to this criterion in the objective function. The predicted versus observed $\delta_{tiller}$ distributions including all plant groups and observation times differed noticeably but not significantly (6.6 ± 8.4 ‰ and 3.7 ± 8.4 ‰, respectively) when pooling 3 simulated $\delta_{tiller}$ randomly at each observation time (P>0.01 in 92 cases out of 100 repeated drawings), as in measurements. Besides, the simulated $\psi_{leaf}$ fitted well the observations ($R^2=0.67$, overall distributions: $-0.175 \pm 0.053$ MPa and $-0.177 \pm 0.053$ MPa, respectively, Fig. 5c). When analyzing the distributions of $\psi_{leaf}$ and $\delta_{tiller}$

per maximum root system depth (Fig. 5b and d), it appears that the $\psi_{leaf}$ signal is not sensitive to the rooting depth (Fig.
5d), while $\delta_{tiller}$ is more sensitive to rooting depth than to the temporal evolution of the plant environment (Fig. 5b).
This leaves us with two hypotheses. The "rollercoaster hypothesis": $\delta_{tiller}$ rapidly goes up and down with all
individuals on board of the same car (i.e. little variability within the population, unlike predictions in Fig. 5a, but like
the simulated $\psi_{leaf}$ in Fig. 5c). If that is correct, the physical model lacks a process that would capture the observed
temporal fluctuations of $\delta_{tiller}$. The "swarm pattern hypothesis": $\delta_{tiller}$ is rather stable in time but its values within the
plant population are dispersed like in a flying swarm, so that $\delta_{tiller}$ values sampled at different times fluctuate, not due
to temporal dynamics but to the fact that different individuals are sampled (Fig. 5a).
The model suggests that the tall fescue population $\psi_{leaf}$ follows a "rollercoaster" dynamics driven by transpiration rate,
while the population $\delta_{tiller}$ follows a "swarm" pattern driven by the maximum rooting depth of the sampled plants. As
no correlation could be expected between the drivers (the maximum rooting depth of the sample plants and canopy
transpiration rate), our analysis explains the absence of correlation between $\delta_{tiller}$ and $\psi_{leaf}$ or transpiration rate.
In future experiments and in the specific context of labeling pulses, sampling more plants at each observation time
would help disentangle the spatial from temporal sources of variability of $\psi_{leaf}$ and $\delta_{tiller}$. It would however be at the
cost of the temporal resolution of observations, or would necessitate a larger setup with more plants in the case of
controlled conditions experiments.

### 3.2.2 Independent observations support the validity of the hydraulic model predictions

In the last 12 hours of the experiment (DaS 167 – 17:00 to DaS 168 – 05:00), the measured soil water content increased
by 0.029 $m^3$ $m^{-3}$ at –0.9 m depth, which could be a sign of nighttime hydraulic redistribution. During the same period,
the physical model predicted a cumulative water exudation sufficient to increase soil water content by 0.003 $m^3$ $m^{-3}$,
as soil water potential was sufficiently low to generate reverse flow, but high enough not to disrupt the hydraulic
continuity between soil and roots [Carminati and Vetterlein, 2013; Meunier et al., 2017a]. While this increase is smaller
than the observed water content change, it is only a component in the soil water mass balance. Given the soil water
potential vertical gradient, upward soil capillary water flow may have accounted for another part of the observed
moisture change. Experimental observations also show that $\delta_{soil}$ increased by 1.0 ‰ at 0.9 m depth during that time (–
6.2 ‰, a value significantly higher than –7.1 ‰ $\pm$ 0.1 ‰ at earlier times based on ANOVA analysis, P<0.01), while
our simulations of hydraulic redistribution generated an increase of $\delta_{soil}$ by 0.34 ‰. As soil capillary flow may not
generate local maxima of $\delta_{soil}$ (no enrichment observed at surrounding depths, see Fig. 2b), and soil evaporation is
assumed negligible at that depth, it is likely that the observed local enrichment was entirely due to hydraulic
redistribution, which would then be underestimated by a factor of about 3 in our simulations. Increasing water
exudation by a factor 3 would imply a simulated water content change due to exudation of 0.0090 $m^3$ $m^{-3}$ absolute
water content, which remains compatible with the experimental observation. Between –1.1 m and –0.9 depths, the

nighttime water flow pattern transitioned from exudation to uptake in both measurements and predictions. At –1.1 m, the model predicted a cumulative water uptake sufficient to decrease soil water content by 0.0101 $m^3 m^{-3}$, as compared to the observed 0.0141 $m^3 m^{-3}$ total soil water content decrease. The remaining 0.004 $m^3 m^{-3}$ water content decrease may have contributed to the recharge to the soil layers above through capillary flow, which was not simulated. Therefore, all relevant measurements (local increase of soil water content, local enrichment of water isotopic composition) and simulation results ($S<0$, i.e. local water release from roots) clearly converge to the conclusion that hydraulic lift occurred in the vicinity of -0.9 m depth in the early morning of DaS 168.

As far as fitted parameter values are concerned, $L_{pr}$ (2.3 $10^{-7}$ m $MPa^{-1}$ $s^{-1}$) was in the range found by Martre et al. [2001] in tall fescue (2.2 $10^{-7}$ ± 0.1 m $MPa^{-1}$ $s^{-1}$) and falls in the range obtained by Meunier et al. [2017a] for another grass (*Lolium multiflorum* Lam., 6.8 $10^{-8}$ to 6.8 $10^{-7}$ m $MPa^{-1}$ $s^{-1}$). Our $k_{axial}$ value cannot be compared to values of axial root conductance from the literature as it transfers the water absorbed by roots in a single "big root" per group of 5 identical plants. The optimal value of $k_{sat}$ was quite high (Table 1) but reportedly very correlated to $\lambda$ (i.e. soil unsaturated hydraulic conductivity is proportional to $k_{sat}$, but also to $S_e^{\lambda}$ [van Genuchten, 1980]), so that the low value of the latter compensated the high value of the former, thus they should be considered as effective rather than physical parameters.

### 3.2.3 Other sources of variability and observational error

Our treatment of the soil medium in this experiment (sieving, irrigation from the bottom) makes it laterally more homogeneous than natural soils. This method allowed us to study specifically the impact of the vertical gradients of $\delta_{soil}$ on $\delta_{tiller}$. It also justified the use of a simplistic 1-D model adapted to the vertically resolved measurements. If lateral heterogeneity of soil water content remained and was accounted for, our predictions of root water uptake distribution, $\delta_{tiller}$ and $\psi_{leaf}$ would be altered. Observational errors in the gravimetric soil water content measurement (turned into soil water potential using the soil water retention curve) would as well alter these predictions. In order to quantify the sensitivity of our simulated results to such heterogeneity or observational error, we varied the soil water content input by ± 0.02 $m^3 m^{-3}$ at three critical depths (–0.9, –1.1 and –1.3 m, before interpolation), at the last observation time, during which measurements and simulations suggested that hydraulic lift occurred. Our results were mostly sensitive to soil water content alterations at –0.9 m, and barely differed in response to alterations at –1.1 and –1.3 m, though the conclusions were not affected qualitatively. No statistically significant difference between predicted and observed $\delta_{tiller}$ distributions for the overall dataset could be found when pooling 3 simulated $\delta_{tiller}$ randomly at each observation time (predicted and observed $\delta_{tiller}$ distributions were closest to differ when soil water content was reduced by 0.02 $m^3 m^{-3}$ at 0.9 m depth; P>0.01 in 76 cases out of 100 repeated drawings). Measured and simulated $\psi_{leaf}$ remained very correlated in all cases (from $R^2$=0.69 to 0.74 when adding or removing 0.02 $m^3 m^{-3}$ at 0.9 m depth, respectively). Furthermore, when adding or removing 0.02 $m^3 m^{-3}$ at 0.9 m depth, cumulative water exudation at –0.9 m varied

between 0.0019 and 0.0035 $m^3$ $m^{-3}$, uptake at $-1.1$ m varied between 0.0080 and 0.0108 $m^3$ $m^{-3}$, and the simulated change of $\delta_{soil}$ ranged between 0.28 and 0.40 ‰, respectively.

Lateral heterogeneity of soil water isotopic composition may as well occur at the microscopic scale. As water in micropores is less mobile than water in meso- and macropores [Alletto et al., 2006], it is likely that, in the lower half of the profile, the capillary rise of labelled water affected the composition of water in meso- and macropores more than in micropores. If roots have more access to meso- and macropore water, then the water absorbed by roots would be isotopically enriched, as compared to the "bulk soil water" characterized experimentally. The importance of this possible bias depends on soil texture and heterogeneity (e.g. existence of more isolated "pockets" of soil or compact clusters), as well as on the speed of water mixing between mobile and immobile water fractions [Gazis and Feng, 2004]. Including this process in the modelling would necessitate sufficient observations to estimate the aforementioned properties, and ideally some quantification of the lateral heterogeneity of soil water isotopic composition at the micro-scale.

The lateral heterogeneity of soil hydraulic properties and root distribution may also have participated to the generation of lateral soil water potential heterogeneities, particularly in undisturbed soils. If one had access to data on lateral heterogeneity of soil properties and rooting density, it would be possible to simulate 3D soil-root water flow with a tool such as R-SWMS [Javaux et al., 2008], using a randomization technique for soil properties distribution as in Kuhlmann et al. [2012], in order to obtain estimations of the relative importance of this type of heterogeneity on $\delta_{tiller}$ and $\psi_{leaf}$ variability.

Unlike the tiller water isotopic composition, leaf water potential turned out to be very sensitive to transpiration rate in our simulations (see temporal fluctuations of grey lines in Figure 5 panel c) and not very sensitive to root distribution (see small variations of leaf water potential across individuals in Figure 5 panel d). This suggests that in this setup the hydraulic conductance of the soil-root system limited shoot water supply more than the distribution of roots, as in Sulis et al. [2019]. Simulated baseline (i.e. for uniform transpiration rates) leaf water potentials are shown as grey lines in Figure 5 panel c, and measured leaf water potentials as a green line in the same panel. The fact that they match well, despite the high sensitivity of leaf water potential to transpiration rate, reinforces the idea that transpiration rate was likely not spatially heterogeneous among the plant population. Therefore, the tiller water isotopic composition, whose sensitivity to transpiration rate is already very low, was likely not affected by transpiration rate heterogeneity.

**3.2.4 Do root water uptake profiles predicted by hydraulic and Bayesian models differ?**

The root water uptake dynamics predicted by the mechanistic model are shown in Fig. 6a. The overall pattern of peaking water uptake in the lower part of the profile during daytime matched that of the statistical model, and the correlation coefficient of both model predictions was relatively high ($R^2$=0.53) in average over the simulation period, see Figure 7. The main differences were the following: (i) in the upper soil layers where the soil water potential was

lower –1.5 MPa, the statistical model predicted water uptake, which is theoretically impossible given the leaf water
potential above –0.4 MPa [van Den Honert, 1948]; (ii) In the upper half of the profile, the physical model predicted
exudation at a rate limited by the low hydraulic conductivity between root surface and bulk soil, with a peak at night,
at –0.9 m depth (quantitative analysis in previous section); (iii) Below –1.0 m depth, the water uptake rate predicted
by the statistical model steadily increased with depth while that of the physical model was more uniform, likely due to
axial hydraulic limitation [e.g., Bouda et al., 2018] counteracting the increasing soil water potential with depth. Note
that the outcome of the statistical model may significantly depend on the definition of the a priori relative RWU
(rRWU) profile. In the present study, we set it to follow a "flat" uniform distribution (i.e., $rRWU_j = 1/10$, see Appendix
E), in other words, each layer was initially defined to contribute equally to RWU. To the contrary of other studies [e.g.,
Mahindawansha et al., 2018], where the a priori rRWU profile was empirically constructed on basis of soil water
content and root length density profiles, we decided not to further arbitrarily constrain the Bayesian model for the sake
of comparison with the physically-based soil-root model.

**3.3 Progresses and Challenges in soil water isotopic labeling for RWU determination**

Often in the field, the vertical dynamics of both soil water oxygen and hydrogen isotopic compositions are not strong
enough (or show convolutions leading to issues of identifiability) for partitioning RWU among different contributing
soil water sources. As a consequence, we unfortunately cannot make use of the natural variability in isotopic
abundances for deciphering soil-root transfer processes [Beyer et al., 2018; Burgess et al., 2000]. To address this
limitation of the isotopic methodology, labeling pulses have been applied locally at different depths in the soil profile
[e.g., Beyer et al., 2016] or at the soil upper/lower boundaries under both lab and field conditions by mimicking rain
events [e.g., Piayda et al., 2017] and/or rise of the groundwater table [Meunier et al., 2017a; Kühnhammer et al., 2019].
After labeling, we are faced with two problems: (i) the labeling pulse might enhance RWU at the labeling location if
the volume of added water significantly changes the value of soil water content. It therefore poses the question of the
meaningfulness of the derived RWU profiles, and this independently from the model used (i.e., physically-based soil-
root model or statistical multi-source mixing model). In other worlds: are we observing a natural RWU behavior of the
plant individual or population or are we seeing the influence of the labeling pulse? Certainly a way to move forward
is environmental observatories such as ecotron and field lysimeters [e.g., Groh et al., 2018; Benettin et al., 2018] that
provide means to better constrain hydraulic boundary conditions and reduced their isotopic heterogeneity. They allow
for a mechanistic and holistic understanding of soil-root processes from stable isotopic analysis.
Another topic of concern is (ii) the difficulty to properly observe in situ (1) the propagation of the labeling pulse in the
soil after application and (2) the temporal dynamics of the plant RWU isotopic composition. Beyer and Dubbert [2019]
presented a comprehensive review on recent isotopic techniques for non-destructive, online, and continuous
determination of soil and plant water isotopic compositions [e.g., Rothfuss et al., 2013; Quade et al., 2019; Volkmann

et al., 2016a] as alternatives of the widely used combination of destructive sampling and offline isotopic analysis following cryogenic vacuum extraction [Orlowski et al., 2016b] or liquid-vapor direct equilibration [Wassenaar et al., 2008]. These techniques have the potential for a paradigm change in isotopic studies on RWU processes to the condition that, e.g., isotopic effects during sample collection are fully understood.

The present study highlights the need not to "trust" our isotope data alone and always complement them by information on environmental factors as well as on soil and plant water status to go beyond the simple application of statistical models. This is especially the case in the framework of labeling studies where strong soil water isotopic gradients may induce strong dynamics of the RWU isotopic composition from a low variability of rooting depths.

## 4 Conclusion

In the present study, light could be shed on RWU of *Festuca arundinacae* by specifically manipulating the lower boundary conditions for water content and oxygen isotopic composition. The new version of the one-dimensional model of Couvreur et al. (2014) implemented here accounted for both root and soil hydraulics in a population of "big" root systems of known root length density profile. This approach underlined the high sensitivity of $\delta_{tiller}$ to rooting depth and suggested that if $\delta_{tiller}$ is measured on a limited number of individuals, its variations in time may reflect the heterogeneity of rooting depth within the population, rather than temporal dynamics which was minor in our simulations. The model avoided the prediction of water uptake at locations where it was physically unavailable (e.g., in the top half of the soil profile), by accounting for water potential differences observed between the leaves and the soil, and explained quantitatively the local isotopic enrichment of soil water as the occurrence of nighttime Hydraulic Lift at –0.9 m depth. On the other hand, the Bayesian statistical approach tested for comparison, which was driven by isotopic information solely, naturally translated the observed changes of $\delta_{tiller}$ into profound temporal dynamics of RWU, at the expense of eco-physiological consideration (e. g., temporal dynamics of leaf water potential and transpiration rate).

This case study highlights (i) the potential limitations of water isotopic labeling techniques for studying RWU: the soil water isotopic artificial gradients induced from water addition result in an improvement in RWU profiles determination to the condition that they are properly characterized spatially and temporally. As already pointed out in the review of Rothfuss and Javaux (2017), the study also (ii) underlines the interest of complementing in-situ isotopic observations in soil and plant water with information on soil water status and plant ecophysiology; it finally (iii) calls for the use of simple soil-root models (though requiring additional water status measurements and making more explicit assumptions on the description of the soil-plant system, as compared to the traditional Bayesian approach) for inversing isotopic data and gain insights into the RWU process.

**Acknowledgements**

The experiment was part of the ASCHYD ("Biogeochemical characterization of Hydraulic Lift") project and supported by the French EC2CO/BIOHEFFECT program (CNRS – INSU, ANDRA, BRGM, CNES, IFREMER, IRSTEA, IRD, INRA and Météo France). During the preparation of this manuscript, V.C. was supported by the Belgian National Fund for Scientific Research (FNRS, FC 84104) the Interuniversity Attraction Poles Programme-Belgian Science Policy (grant IAP7/29), and the "Communauté française de Belgique-Actions de Recherches Concertées" (grant ARC16/21-075); FM was first funded by the BAEF and the WBI,then by the FWO as a junior postdoc and is thankful to these organizations for their support.

**Data sets**

Upon acceptance, all research data needed for creating plots will be available in reliable FAIR-aligned data repositories with assigned DOIs.

**Author contribution**

TB, JLD, and PB designed the experiments and TB, JLD, PB, and YR carried them out. VC, FM, and MJ developed the physically-based root water uptake model code and VC and FM performed the simulations. YR performed the statistical simulations. VC, YR, FM, and MJ prepared the manuscript with contributions from all co-authors.

**Competing interests**

The authors declare that they have no conflict of interest.

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

**5 Tables**

|  | $L_{pr}$ (m MPa$^{-1}$s$^{-1}$) | $k_{axial}$ (m$^4$ MPa$^{-1}$ s$^{-1}$) | $k_{sat}$ (m$^2$ MPa$^{-1}$ s$^{-1}$) | $\lambda$ (-) |
|---|---|---|---|---|
| Lower limit | $10^{-11}$ | $10^{-13}$ | $10^{-5}$ | $-5$ |
| Upper limit | $10^{-6}$ | $10^{-8}$ | $10^{-2}$ | $2$ |
| Value at best fit | $2.3\ 10^{-7}$ | $4.5\ 10^{-11}$ | $9.5\ 10^{-3}$ | $-4.9$ |

**Table 1. Optimum and limits of the four-dimensional parametric space explored by the global optimization algorithm aiming**
**at minimizing the difference between simulated and observed $\delta_{tiller}$ and $\psi_{leaf}$, as well as their standard deviation from average**
**values during the full experiment.**
**6 Figures**

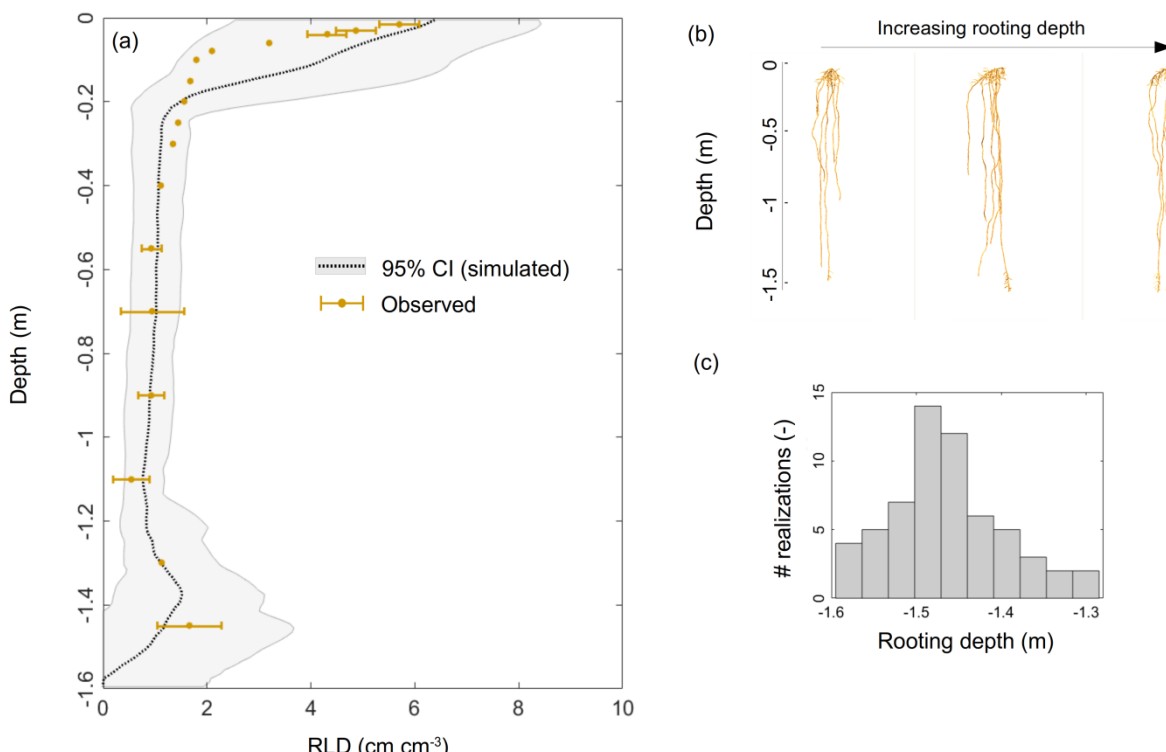


**Figure 1. (a) Simulated (grey envelopes) and observed (brown dots) root length density profiles. Panels (b) and (c) illustrate**
**the variability in modelled root system architectures and rooting depths, respectively.**

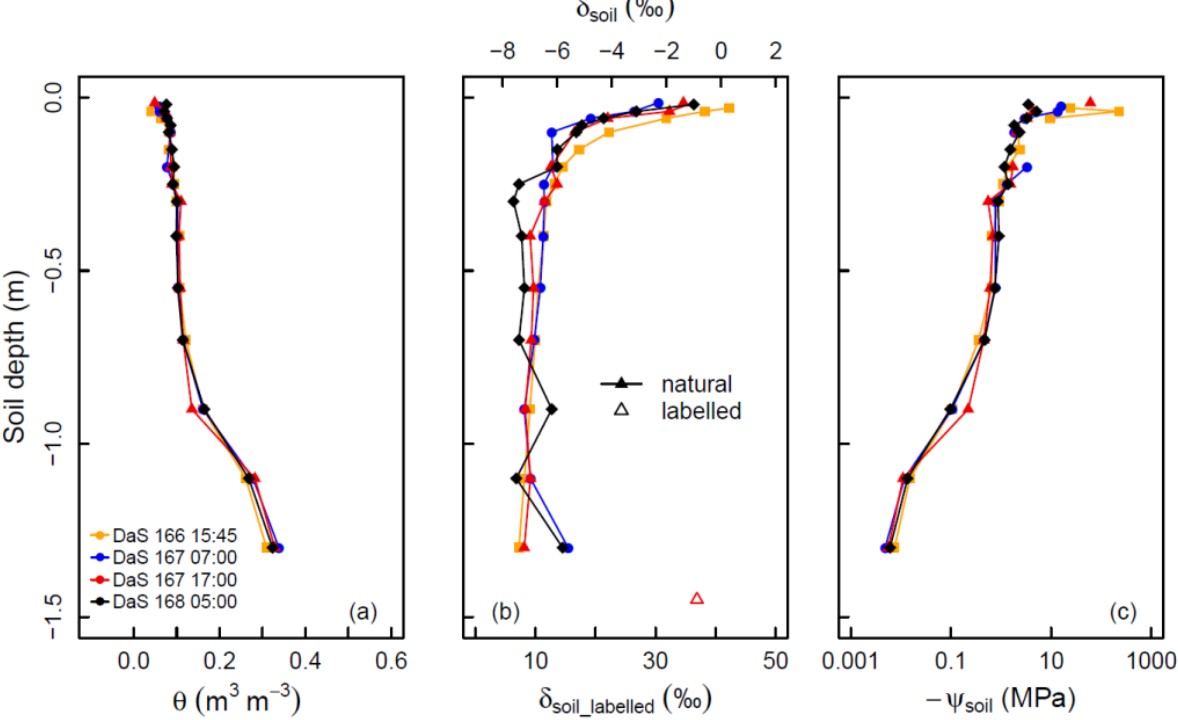


**Figure 2. Measured soil volumetric water content ($\theta$, panel a), oxygen isotopic composition ($\delta_{soil}$, panel b), and calculated**
**soil matric potential ($\psi_{soil}$, panel c) profiles during the sampling period.**

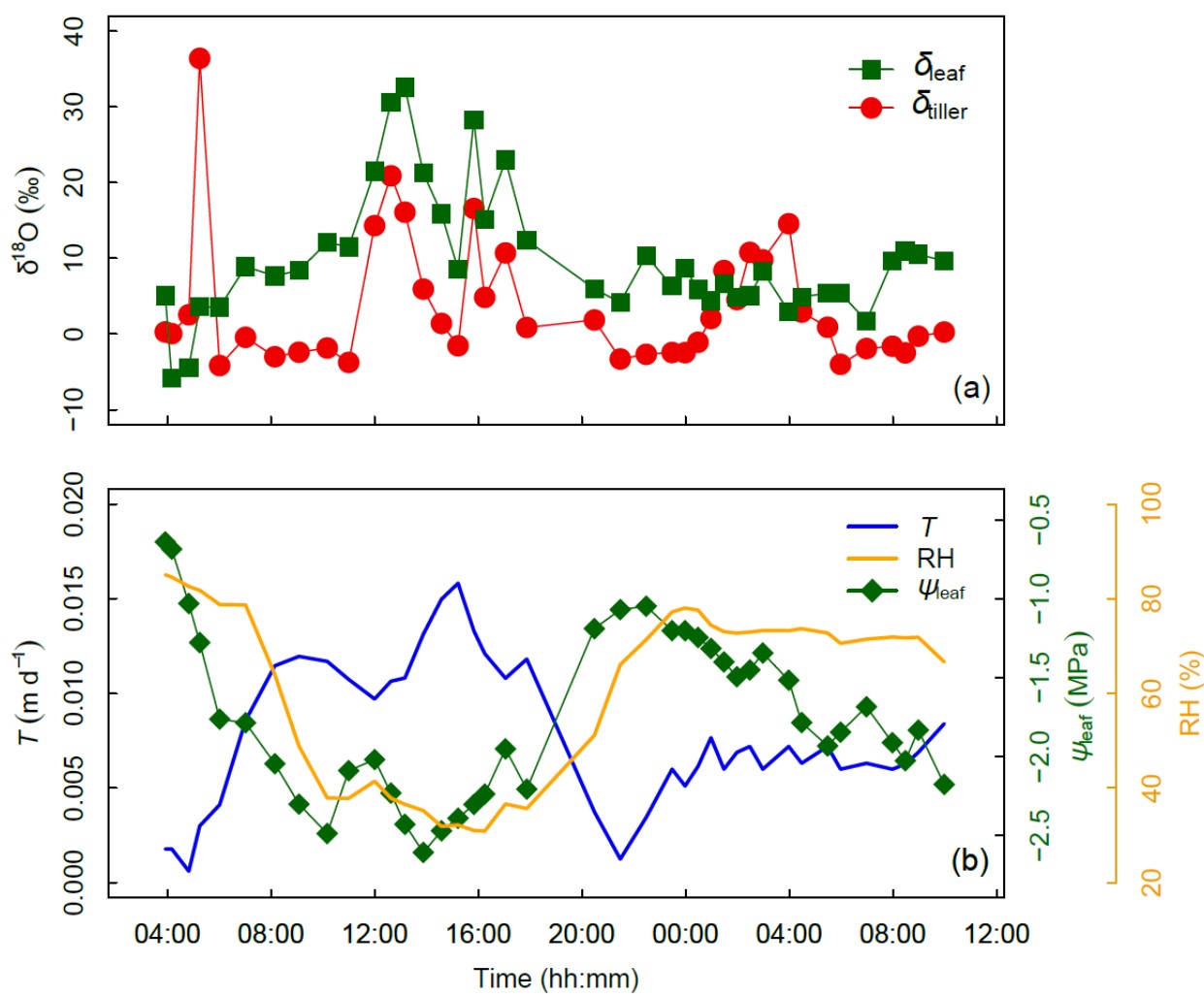


Figure 3. (a) Time series of tiller and leaf water oxygen isotopic compositions ($\delta_{tiller}$ and $\delta_{soil}$, ‰). (b) Transpiration flux ($T$,
in m d$^{-1}$), relative humidity (HR, %), and leaf water potential ($\psi_{leaf}$, in MPa, panel b) from days after seeding DaS 167 –
04:00 to DaS 168 – 11:00. Time of Labeling was DaS 166 – 17:00.

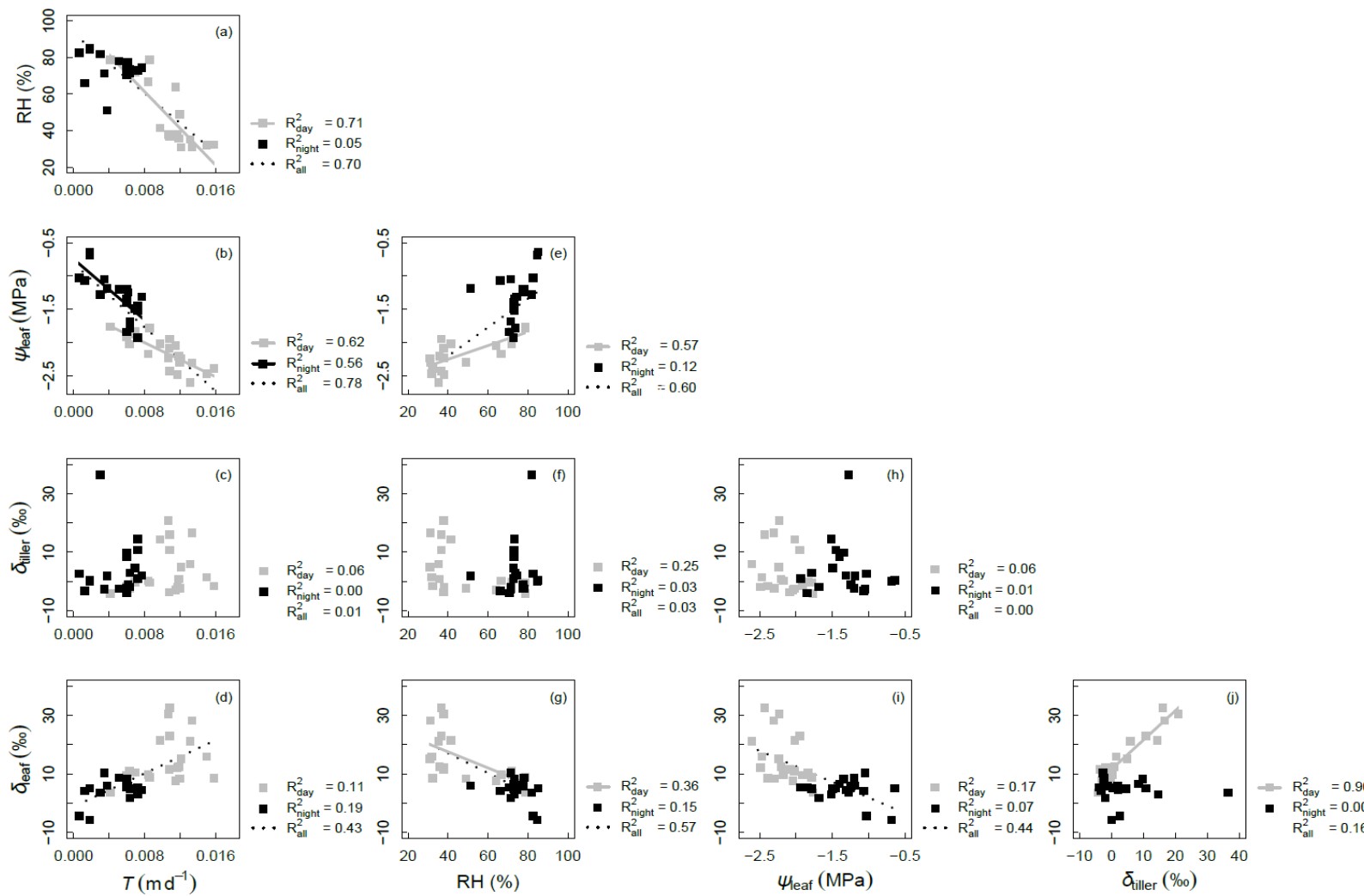


**Figure 4. Correlations between measured variables: oxygen isotopic compositions of xylem and leaf waters ($\delta_{tiller}$ and $\delta_{leaf}$,**
**in ‰), transpiration rate ($T$, in m d$^{-1}$), relative humidity (RH, %), and leaf water potential ($\psi_{leaf}$, in MPa). Coefficient of**
**determinations ($R^2$) are reported for all data, and separately for 'day' data (gray symbols) and 'night' data (black symbols)**
**(see Appendix C for definition of 'day' and 'night' experimental periods). Regression lines are drawn for linear models with**
**p-value < 0.01**

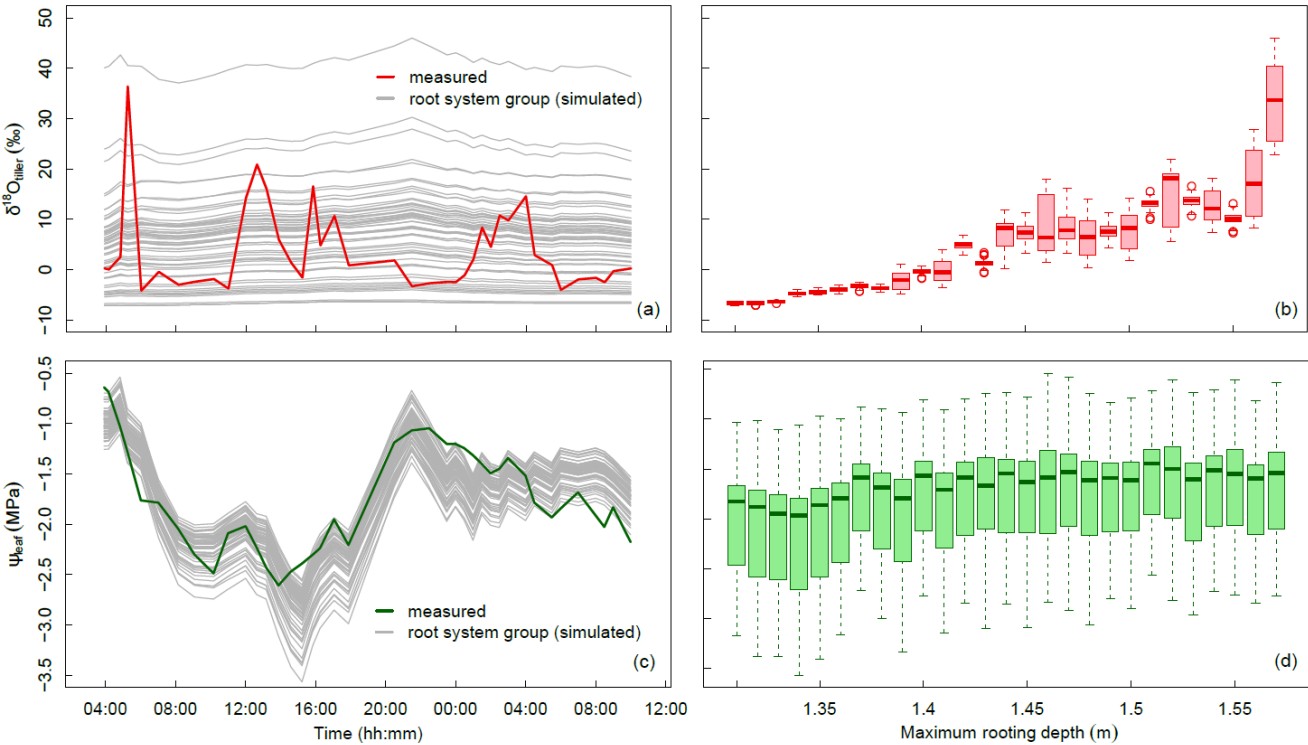


Figure 5. Variation of $\delta_{tiller}$ and $\psi_{leaf}$ in time and across the 60 groups of simulated root systems. (a) Temporal dynamics of $\delta_{tiller}$ measured (thick red line) and simulated (thin grey lines, one line per root system group, following a "swarm" pattern). (b) Boxplot of simulated $\delta_{tiller}$ values for each root system maximum depth, by 1 cm increment. (c) Temporal dynamics of $\psi_{leaf}$ measured (thick green line) and simulated (thin grey lines, one line per root system group, following a "rollercoaster" pattern). (d) Boxplot of simulated $\psi_{leaf}$ values for each root system maximum depth, by 1 cm increment.

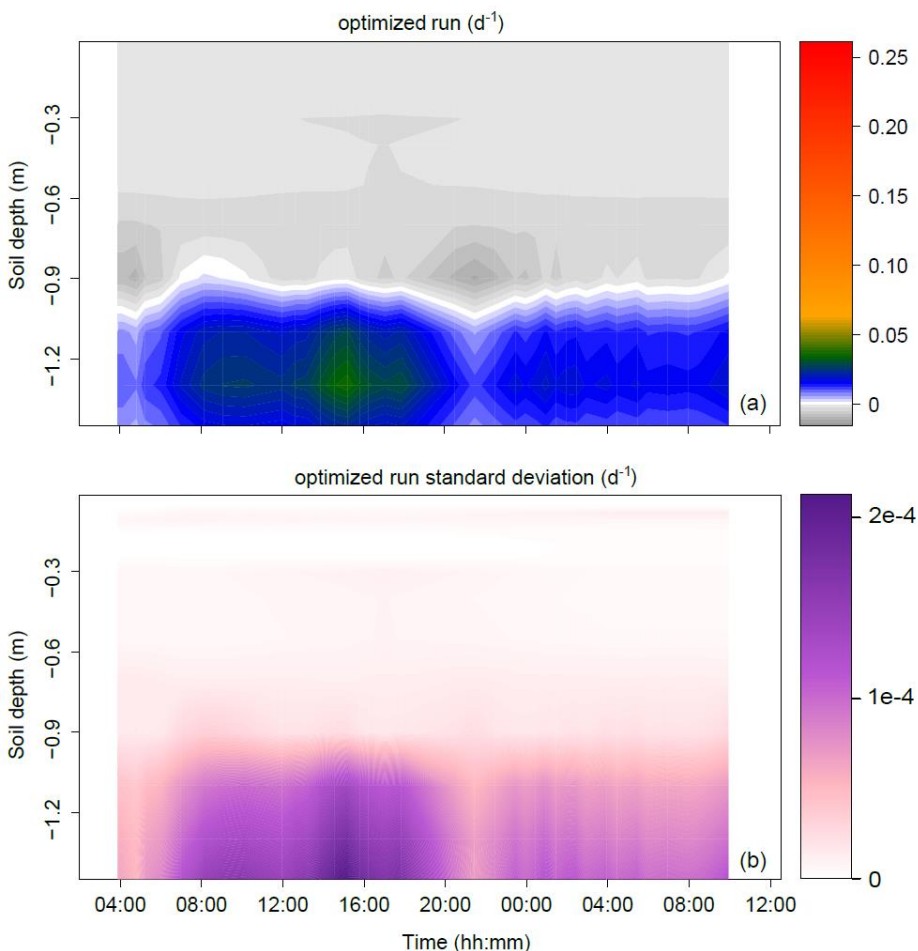


**Figure 6. Time series of the profiles of root water uptake per unit soil volume (sink term, d⁻¹) computed with the physically-**
**based model. (a) Sum of sink terms across the 60 groups of the population. (b) Variability of sink terms within the 60 groups**
**of the population (1 standard deviation).**

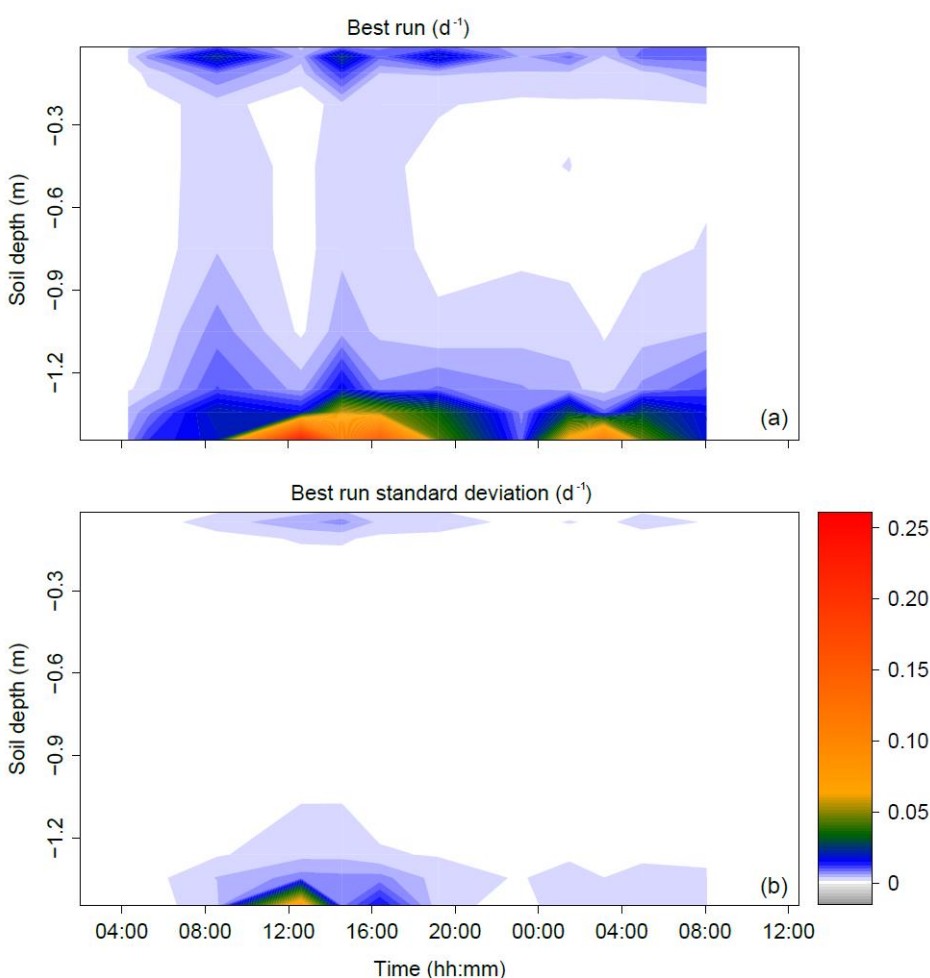


**Figure 7. Time series of the profiles of root water uptake per unit soil volume (sink term, $d^{-1}$) computed with the statistical model SIAR (a). Panel (b) reports the variance of the estimated sink term (1 standard deviation).**



**7 Appendix**

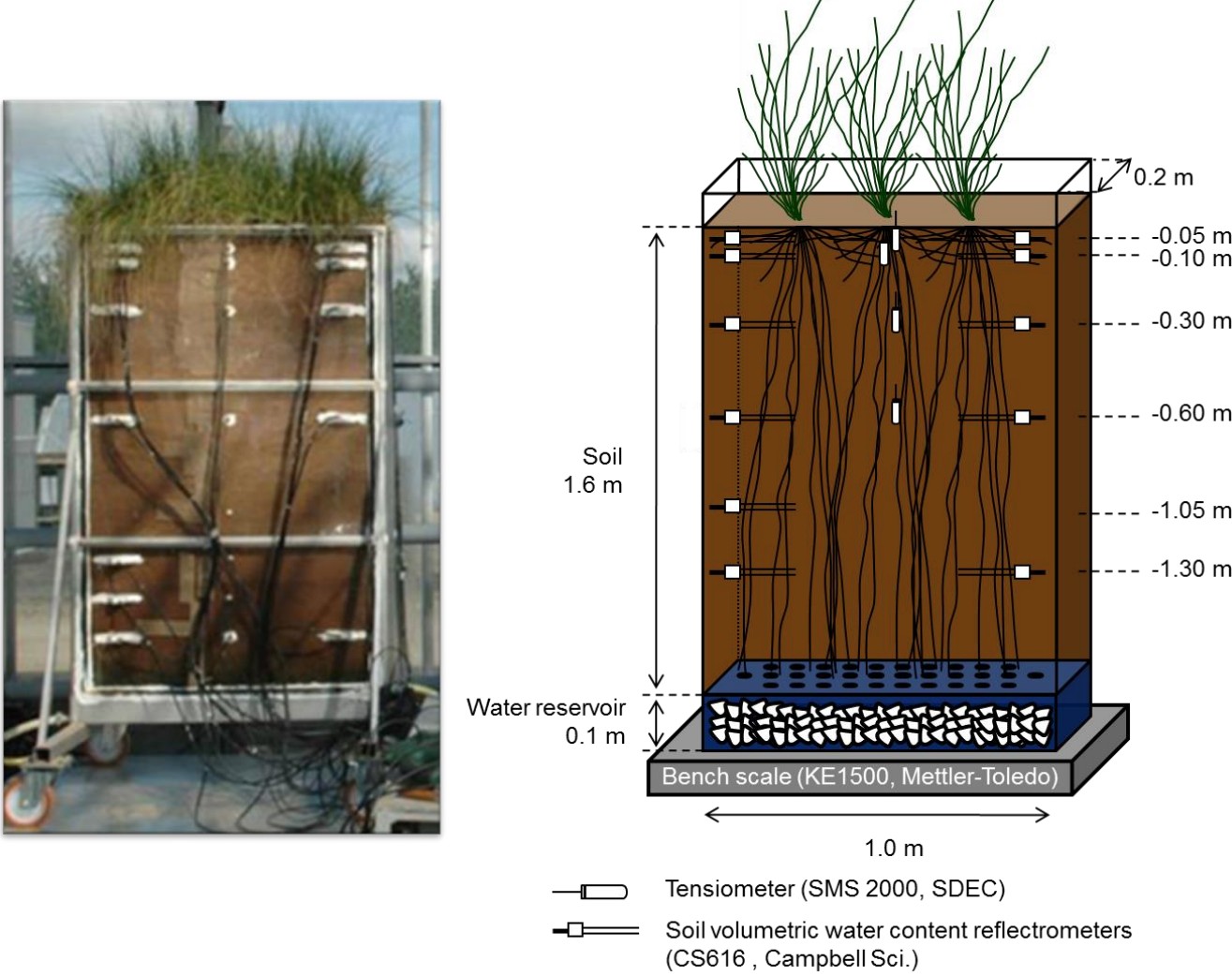


**Appendix A. Soil macro-rhizotron experimental setup with tall fescue cover**

| $\theta_{sat}$ (m$^3$ m$^{-3}$) | $\theta_{res}$ (m$^3$ m$^{-3}$) | α (m$^{-1}$) | n (-) |
|---|---|---|---|
| 0.4 | 0.044 | 0.0285 | 2.29 |

**Appendix B. Soil retention curve and parameters optimized values [van Genuchten, 1980 - Burdine] [Meunier et al., 2017a]**

# Appendix C. Timeline of destructive sampling

| | DAS 166 | DAS 167 | | | | | | | | | | | | | | | | | | | | | | |
|---|---|---|---|---|---|---|---|---|---|---|---|---|---|---|---|---|---|---|---|---|---|---|---|---|---|
| | | 'night' data | | | | | 'day' data | | | | | | | | | | | | | | | 'night data' | | | |
| Time | 15:45 | 03:55 | 04:10 | 04:50 | 05:15 | 06:00 | 07:00 | 08:10 | 09:05 | 10:10 | 11:00 | 12:00 | 12:40 | 13:10 | 13:55 | 14:35 | 15:15 | 15:50 | 16:15 | 17:00 | 17:50 | 20:30 | 21:30 | 22:30 | 23:30 |
| Soil | x | | | | | | x | | | | | | | | | | | | | x | | | | | |
| Leaves | | x | x | x | x | x | x | x | x | x | x | x | x | x | x | x | x | x | x | x | x | x | x | x | x |
| roots | x | | | | | | | | | | | | | | | | | | | | | | | | |

| | DAS 168 | | | | | | | | | | | | | | | | |
|---|---|---|---|---|---|---|---|---|---|---|---|---|---|---|---|---|---|
| | 'night data' | | | | | | | | | | | | 'day' data | | | | |
| Time | 00:00 | 00:30 | 01:00 | 01:30 | 02:00 | 02:30 | 03:00 | 04:00 | 04:30 | 05:00 | 05:30 | 06:00 | 07:00 | 08:00 | 08:30 | 09:00 | 10:00 |
| Soil | | | | | | | | | | x | | | | | | | |
| Leaves | x | x | x | x | x | x | x | x | x | | x | x | x | x | x | x | x |

**Appendix D. Inverse modeling scheme**
The parametrization method was inverse modeling, with four targets: (i) minimizing the differences between observed
and predicted $\delta_{tiller}$ in each pool $p$, (ii) minimizing the difference between the standard deviations of observed and
predicted $\delta_{tiller}$ (temporal and population deviations altogether), (iii) minimizing the differences between observed and
predicted $\psi_{leaf}$ in each root system group $i$, (iv) minimizing the difference between the standard deviations of observed
and predicted $\delta_{tiller}$ (temporal and population deviations altogether). These targets translated as an objective function
($OF$) to be minimized, where differences were normalized by the standard deviation ($SD$) of observations in order to
make the error function dimensionless:
$OF$
$$
= \sqrt{\frac{1}{2}\left( \frac{1}{N_p N_t} \sum_i \sum_t \left( \frac{\delta_{tiller,obs}(t) - \delta_{tiller,p,sim}(t)}{SD\left(\delta_{tiller,obs}(t)\right)} \right)^2 + \frac{1}{N_i N_t} \sum_i \sum_t \left( \frac{\psi_{leaf,obs}(t) - \psi_{leaf,i,sim}(t)}{SD\left(\psi_{leaf,obs}(t)\right)} \right)^2 \right)}
$$

$$
+ \left| \frac{SD\left(\delta_{tiller,obs}(t)\right) - SD\left(\delta_{tiller,p,sim}(t)\right)}{SD\left(\delta_{tiller,obs}(t)\right)} \right| + \left| \frac{SD\left(\psi_{leaf,obs}(t)\right) - SD\left(\psi_{leaf,i,sim}(t)\right)}{SD\left(\psi_{leaf,obs}(t)\right)} \right| \qquad \text{(D1)}
$$

where $N_p$ is the number of $\delta_{tiller}$ pools simulated (100) at each observation time, $N_i$ is the number of plant groups
simulated (60), and $N_t$ the total number of observation times (40).
The global optimizer Multistart heuristic algorithm OQNLP (Optimal Methods Inc.) of the MATLAB (The
MathWorks, Inc., USA) optimization toolbox was used to minimize the error function within the lower and upper
limits of the parametric space reported in Table 1.

 **Appendix E. Statistical determination of relative RWU profiles with SIAR**

The Bayesian inference statistical model SIAR [Parnell et al., 2013] was used to determine the profiles of relative
contributions to RWU (rRWU, dimensionless) of ten identified potential water sources. These water sources were
defined to originate from the soil layers 0.00-0.03, 0.03-0.07, 0.07-0.15, 0.15-0.30, 0.30-0.60, 0.60-0.90, 0.90-1.20,
1.20-1.32, 1.32-1.37, and 1.37-1.44 m. Their corresponding isotopic compositions were obtained from the measured
soil water isotopic compositions ($\delta_{soil}$) and volumetric content ($\theta$) values following Eq. (E1) [Rothfuss and Javaux,
2017]:
$$\delta_{soil,J} = \frac{\sum_{j \in J} \delta_{soil,j} \cdot \theta_j \cdot \Delta Z_j}{\sum_{j \in J} \theta_j \cdot \Delta Z_j} \tag{E1}$$

where J is the soil layer index, j is the soil sub-layer index, and $\Delta Z_j$ is the thickness of the soil sub-layer j. Therefore,
equation (E1) translates the soil water isotopic composition measured across sub-layers j into representative isotopic
compositions of the different sources (i.e., across layers J). The computed $\delta_{soil,J}$ were compared to $\delta_{tiller}$ values. For this,
$\delta_{tiller}$ measurements were pooled in twelve groups corresponding to different time periods. These groups were defined
to best reflect the apparent temporal dynamics of $\delta_{tiller}$.
For each of the twelve time periods:
(i)   the function *siarmcmcdirichletv4* of the SIAR R package (https://cran.r-

project.org/web/packages/siar/index.html) was run 500,000 times with prescribed burnin and thinby

equal to 50000 and 15, respectively. The output of the model (i.e., the *a posteriori* rRWU distribution

across the ten soil water sources J) was obtained from a flat Dirichlet *a priori* rRWU distribution (i.e.,

$rRWU_J=1/10$);

(ii)  the 'best run' (*br*, dimensionless) was selected from SIAR's output. It was defined as the closest solution

of relative contributions across sources from the set of most frequent values (*mfv*, dimensionless), i.e.,

the relative contribution with the greatest probability of occurrence. The best run was identified as

minimizing the objective function below, i.e., the RMSE (root mean square error) with respect to the set

of $mfv_J$:

$$OF = \sqrt{\frac{\Sigma_{J=1}^{10} \; (mfv_J - br_J)^2}{10}}$$    (E2)
(iii)  *br* was then multiplied by transpiration rate (in m d$^{-1}$) and divided by soil layer thicknesses ($\Delta Z_J$, in m)
to obtain sink terms ($S_J$, i.e. root water uptake rate per unit soil volume, expressed in d$^{-1}$). The interest
of sink terms in a comparison is that they do not vary with soil vertical discretization.
Steps (i)-(iii) were repeated a 1,000 times to estimate the variance of the best run for each time period and soil water
source J.