# Peer review of "Disentangling temporal and population variability in plant root"

_Hydrology and Earth System Sciences, 2019_

## Referee Comment (RC1) · Anonymous Referee #1 · 20 Nov 2019

Couvreur and colleagues present an interesting isotopic labelling experiment and innovative simulations of the processes in the soil-roots interactions. Their study is addressing current research gaps and will thus be of interest to the readership of HESS. The manuscript is well prepared and the figures are mostly informative. I provide two general recommendations and several minor technical comments below. I recommend publication after addressing these comments.

General aspects:

The "rollercoaster hypothesis" and the "swarm pattern hypothesis" both focus on the variation of $\delta^{18}O$ in tiller across plants and/or over time, respectively. However, the studied system is likely to be more complex due to heterogeneity of the water

flow/capillary rise. Do you see a chance to improve the modelling results when moving from a uniform flow/capillary rise to some kind of dual-permeabilty approach accounting for potential subsurface isotopic heterogeneity?

I was missing a discussion of the uncertainties regarding for example soil moisture estimates and the impact of such uncertainties for the interpretation regarding potential processes (i.e., hydraulic lift).

I further think that the implications of their interesting findings (i.e., no match between the ensemble of various simulations and the observations; Fig. 5) for both field studies labelled or with natural isotope compositions and the modelling of the soil-root interactions could be made clearer. This way, the manuscript might have a higher impact and could provide recommendations to overcome limitations in observation techniques and modelling approaches.

I appreciate that the authors will upload the data of the study. Are they further intending to make the model code available?

Technical comments:

L 77: monotonic gradient? Consider sinusoidal variability across the depth, which would cause issues of identifiability

L 80: Not only GW, also due to increasing dispersion with depth – even if the GW table is several meters deep

L 100: This paragraph is kept quite general after a very informative introduction. I suggest to be more specific and especially pose hypothesis or specific research questions.

L 117: Since you provide the variable and unit for soil moisture, you probably should also add that to matric potential.

L 140: replace "isotopic" with "$\delta^{18}O$"

L 140: How was the sampling done? Soil corer? How much soil was sampled?

L 149: provide info about temperature, applied vacuum and time of extraction

158: Not sure what "(95 m root (g root)$^-$ − 1)." Means

Figure 1: The circles connecting the bottom of the profile of Figure 1a and the histogram of 1c are more confusing than helping. I suggest to get rid of them. The same would apply for the arrow connecting to 1b.

L 172: All variables should be explained here. For example $L_p r$ is explained in L 216

L 181: The variable "n" should be briefly explained as one of the MVG parameters. Also, consider adding n and Sej to the list of variables.

L 209: Please define conditions for exudation. I believe it is for $S_j$<0, but not sure.

L 239: I do not see how the soil moisture varied notably at 1.3 m depth. What do you mean here? How comes that you refer to 12:00 and 20:00 on DaS 167, while that is not shown in Figure 2a?

L 243: Again, you refer to a time (7:30), which is not shown in the Figure and you should refer to it as soil$_{labelled}$" and not "soil" to be consistent with Figure 2.

L 244: "lead us to assume" or "leads to the assumption"

L 262: It is unclear which of the correlations are describe a significant relationship. I suggest to only draw the regression lines for significant relationships in Figure 4.

L 281: replace "et" with "and"

L 298: Unclear what is meant with "over all dataset". I believe you mean the 60 different root system classes. Please be more specific.

L 315: It seems to me that in-situ measurements would overcome these limitations. One could sample in parallel several plants and thus, observe the temporal dynamics at individual plant level.

L 319: What is the expected accuracy of your volumetric soil moisture measurements. Given that you derived this from gravimetric water content and a bulk density, which was assumed to be constant in the repacked soil. However, relatively small differences in bulk density of just a few g cm$^{-3}$ will affect the estimates of the volumetric water content. It would be good to account for such uncertainties in this discussion.

L 325: What do you mean with "significantly higher"? Did you apply a statistical test? I

believe that you mean that the difference is higher than the measurement uncertainty.
Figure 6b: The title says "standard error", but the caption says "standard deviation". Which one is it? Please correct.
L 360: the upper half of the soil profile
L 367: "water addition is localized and not broadcasted in the soil" is unclear. What do you mean with "broadcasted"?
L 370: "simple"? In addition to the usual struggle of assessing meaningful MVG parameters to describe the soil water transport, also like for example $L_{pr}$ and $K_{axial}$ are needed, which are not easily derived, but its estimation adds to the uncertainty of the uptake depths.

---

## Referee Comment (RC2) · Matthias Beyer (Referee) · 24 Nov 2019

The manuscript hess-2019-543 'Disentangling temporal and population variability in plant root water uptake from stable isotopic analysis: a labeling study' by Couvreur et al. present a lab-/field- and model-based study of root water uptake during an artificial tracer experiment, where the soil is wetted from below (as opposed to often, via irrigation). They support their isotope analysis by hydraulic measures in order to provide a holistic understanding of RWU.

The authors address the urgent and contemporary need for increasing the reliability of RWU models and improve the understanding root water uptake patterns. It has been

often proposed to combine hydraulic, water isotope and other information in order to do so, the presented study is in my opinion a holistic and promising approach. The rollercoaster vs. swarm hypothesis is also a good idea, though (as the authors state themselves) it should be validated further. It is also great that both data from an experiment and modeling are provided, rather than only one of the two. This manuscript is well prepared, and the topic is highly relevant. The figures are suitable and well-explained.

I highly recommend this manuscript for HESS, though I have a number of comments/questions that might help to improve this manuscript further. In brief, a few general comments, which are all rather minor:

- The discussion on hydraulic redistribution should be strengthened. Do the authors see a clear sign or not? I think strengthening this part would be of utmost interest for many people from the ecohydrological community.

- When reading the results and discussion section, I realized that there are very small differences discussed in the manuscript (e.g. 0,41 per mill, 1 per mil, etc. . . ..). I think it is necessary to think about uncertainties in that respect and really decide which of the differences are likely 'true' differences or simply within the variability/uncertainty.

- I find the discussion of the physical experiment slightly too weak compared to the results drawn from the modeling (I also indicated this in the detailed comments below).

- The authors use $\delta$tiller, etc. without providing the water isotope (e.g. $\delta$tiller18O). I think this is important to clarify (it was only 18O used, correct?) starting with the symbol description. Why was only oxygen-18 used? (and not 2H in addition?)

- Will the model be made publicly available? It would be very interesting to apply the model with other datasets (e.g. some in situ datasets of joint soil and plant water isotopes) I wish the authors good luck and look forward to the final publication.

Greetings and best wishes, Matthias Beyer

[Figure]

Detailed comments:

- Abstract is very well written

- L.78/79: depends on how deep the groundwater table is. In thick unsaturated zones, often mixing of old water is also a reason. Further, over short time periods a seasonal pattern might persist in the soil

- L.140: in oxygen-18 I guess? Could the authors please add this information?

- L.149: Can the authors please add specifics on the extraction? (Extraction temperature and time for soil and plant samples, how was complete extraction assessed?) The community has been asking in many occasions to provide more transparency of extraction procedures; hence it would be appreciable to add this information.

- L.152/153: the loss of mass would also include evaporation; was this neglected (please clarify) [I see that this is mentioned later in the text, but perhaps better to clarify here]

- L.163: literature

- Chapter 2.4: The explanation and equations make sense to me, but for a detailed evaluation and/or comments on the equations a true modeler might be considered (e.g. M. Cuntz, Wingate/Ogee group)

- Chapter 3.1.1: Figure 2 is mentioned first in the text, then Figure 1….hence, those might be switched Results/Discussion

- -l.243-247 and Fig. 2: There were two soil moisture profiles measured, but only one is shown in Fig.2 (or is that averaged over the two?) I am not sure if that justification that no evaporation was present is sufficient, as the moisture profiles oscillate greatly and over one or two days the effect of evaporation might be minimal (which on the other hand supports the assumption ET=T). Still, evaporation is probably occurring (though at a low rate).

- L.247-249: Yes, but these three options should be discussed by the authors

- L. 250: minimal minimum instead maximal maximum - L. 252: delete level - 3.1.2: Again, if results/discussion is mixed here, those differences and diurnal patterns should be discussed and explained here

- L.269: 'Rayleigh distillation corrections' – this is not explained in the methods. Could the authors provide details on these corrections and/or provide a citation?

- 3.1.3: Well-written and explained

- L.298: yes, but still: 3 per mill is notable for 18O. . .

- L.325: I am not sure if an 0,9 per mil increase is significant. . .were replicates taken for each soil depth? What is the std of those (-often this can be in that range already). . . .if no replicates were taken, this might be well within the uncertainty rather than a true increase

- L.327 depths instead heights - L.345 model instead models - L. 360 upper half instead first half - L.363-365: But couldn't this be implemented to the Bayesian approach via the construction of priors?

Figure 2:

- it's 18O data shown, could this be added to the title (instead of only delta). . .OK it's in the figure description,still. . .

- why is matric potential 'calculated' shown if it was measured?

- Not sure if the inset graphic for the water content is helping the figure

---

## Referee Comment (RC3) · Anonymous Referee #3 · 27 Nov 2019

Review of Manuscript HESS-2019-543

Disentangling temporal and population variability in plant root water uptake from stable isotopic analysis: a labeling study by Couvreur et al.

This manuscript compares two alternative modeling strategies for deriving the sink term (root water uptake) in a controlled ecotron experiment. Strategy 1 uses a simplified root water uptake model which however incorporates the main features of the three dimensional soil water flow, including hydraulic redistribution. The unknown model parameters are calibrated based on isotope data in the tiller and leaf water potentials. Strategy 2 derives root water uptake based on isotope data using Bayesian inference.

The authors find that the results between the two strategies diverge. They show that Bayesian inference yields unphysical fluxes. Based on the model results they conclude that spatial variation ("swarm-like") in tiller isotopic signal is misinterpreted as a strongly fluctuating time series, whereas it actually reflects the different rooting depths of plant individuals. Additionally, they argue that both the root water uptake model and the soil moisture time series suggest hydraulic lift, which cannot be captured by the Bayesian inference based on isotope data alone. Therefore, they conclude that the results obtained based on Bayesian inference could be due to an artifact.

This is a valuable contribution illustrating how sampling choice may affect the interpretation of isotope data. Especially the application of a straightforward process model for comparison with the Bayesian inference together with the dense measurements are extremely helpful to explain the shortcomings of deriving uptake profiles based on isotope data alone. The case is well argued and the methods are sound. I feel the manuscript has potential to making an impact and will find strong interest in the readership of HESS. The paper is mostly well structured, although I have some concerns with the Abstract and Methods section, as well as with some formulations (see below).

I have two general concerns, and a number of editorial remarks (below): The investigated case is a particular one, e.g. with a strong labelling pulse added below the rooting zone. This needs to be explicitly stated and the manuscript should discuss in which other situations such a strong influence of spatial variation is to be expected (and where it is not a concern).

The study suffers from lack of opportunity for validation: The heterogenous rooting depths cannot be measured in situ and therefore it remains a hypothesis. This is ok. But it requires diligent consideration of other assumptions of the model that may have had a similar effect on the model result. How about the inherent assumptions of a big leaf? Could individual differences in leaf development incur similar results? Those considerations need to enter more than now into the discussion. I propose adding a section dealing with the effect of inherent model assumptions.

[Figure]

Detailed comments:

Abstract: Line 18-19: This sentence sounds vague, e.g. "semi-controlled" and "such variables", please formulate more specifically. Line 23-24: "results underlined the discrepancy .. " At this point unclear what is meant Line 29-30: The sentence starting with "The physical model .. " is difficult to understand, please reformulate Line 33: "local increase . . ." this results is not stated earlier and at this position confusing.

Lines 35-62 List of variables Some variables are missing, please complete. Also I propose erasing all the repetitions of "units of". Later in the paper, it will be useful to express volumetric water content as vol-% and I propose adding it here.

Introduction Line 75-76: The description of the "mean value of . . . weighted by" is confusing, please rephrase

Material and Methods Line 114: Please mention that CS616 is a time domain sensor (TDR). Also, reflectometer is a correct, but awkward term for soil moisture sensor. I propose using the latter, just to avoid confusion.

Line 123: The description of the soil is confusing. Is District Cambisol a typo for "Dystric Cambisol". Otherwise, I am not aware what a District Cambisol would refer to, please explain. Besides, Cambisol refers to a soil in situ and after specific pedogenesis which is completely removed in your experiment. Maybe say "The soil originates from a xx Cambisol". Also, does the bulk density refer to the original soil, and is it required to be mentioned?

Line 125: Add "layer" before "by"

Line 128: sols-PST55 sensors are missing above, where installation depths were mentioned. Add there. Please add where they are installed.

Line 130: "between its position and measured soil water content" unclear, please rephrase

Lines 132-136: Please shortly state: Were the plants watered? What was the lower boundary condition?

Line 139: Do you mean "sides" instead of "slides"

Line 144: "three plants were sampled" - does it mean the entire plant or some leafs?

Line 148-149 I believe you mean "from the atmosphere surrounding the rhizotron". Also, I am assuming the latter means the ambient air in the lab? Would be good to specify. Line 162: "60 tall festucae root systems .." Why 60 plants?

Eq. (1) Personally, I do not find this equation obvious. Please motivate the origin. Are there any other assumptions involved besides the big root one?

Line 173-174: "dimension of the domain . . . " I do not understand this statement

Line 176-177: Sentence starting with "The averaged distance .." seems wrong. Maybe erase the last words?

Eq (4): Se is not part of the List of variables, plus the S stated there refers to the sink term not saturation. Please use a different abbreviation.

Line 188: Could not the measured root length density profiles be used?

Line 191: "were derived" unclear how this was derived? Also, where was k_axial in Eq. (7) taken from? Please explain. Ok, I learn later this was calibrated. Maybe mention this here.

Line 191: "root system class" Unclear what is meant with "class".

Line 194: standard sink distribution is not a standard term and requires a bit more explanation to be convincing.

Line 195-196: "potential difference between soil and leaf": You are dealing with a soil profile and a leaf canopy. Thus, where in the soil and leaf are you referring to. Please also translate to what this means for your experimental setup either here or in the

discussion.

Line 196: "assuming negligible stem conductance": Does this imply that all conductance / resistance happens in the root system? Is this a reasonable assumption? ÂăÂăÂăÂăÂăÂăÂăÂăÂăÂăÂăÂăÂăÂăÂăÂăÂăÂăÂăÂă Âă Line 202: "class" unclear

Line 205: "where axial conductances" this comes too late, please move up.

Line 218: I propose moving the inverse modeling procedure out the the appendix and add it to the main text. It is important information.

Line 226: Not sure what is meant with "ten identified potential water sources" .. "10 distinct soil layers" Can you be more specific?

Results and discussion: Lines 335-252: Small issue: Please add some paragraphs in this section.

Line 281: Do you mean "and" instead of "et"

Line 304: With "all the population" do you mean all individuals?

Line 311-313: Sentence is difficult to understand, please rephrase.

Lines 329ff: Since there is repeatedly reference to increasing by xx% this may be strongly confusing. Better use vol-% to be on the safe side.

Line 340 Lambda is not in the list of variables. Also, this information is very compact, and difficult to understand. Please elaborate.

Lines 345ff: I strongly recommend bringing Fig E into the main text. It is discussed and seems therefore sufficiently important. Also, because this is one of the two alternative water uptake profiles which comparison is the main motivation of the manuscript.

Line 360: Replace "first" with "top"

Line 367: Not sure what is meant with "broadcasted"

Figure 2: I was confused about the positive water potentials. Since they were named psi I was instinctively assuming to see matric potential, but plotted are the water potentials. I propose renaming to h. If you want to stick to psi_soil because of psi_leaf (although I seriously think it would not be an issue), please obviously state the reference elevation to avoid this type of confusion.

---

## Author Comment (AC1) · 4 Dec 2019

Authors responses to Anonymous Referee #1

RC| Couvreur and colleagues present an interesting isotopic labelling experiment and innovative simulations of the processes in the soil-roots interactions. Their study is addressing current research gaps and will thus be of interest to the readership of HESS. The manuscript is well prepared and the figures are mostly informative. I provide two general recommendations and several minor technical comments below. I recommend publication after addressing these comments.

[Figure]

AC| Dear reviewer, we thank you for your general comments as well as technical corrections of our manuscript! You will find our answers listed below:

RC| General aspects:

RC| The "rollercoaster hypothesis" and the "swarm pattern hypothesis" both focus on the variation of $\delta18O$ in tiller across plants and/or over time, respectively. However, the studied system is likely to be more complex due to heterogeneity of the water flow/capillary rise. Do you see a chance to improve the modelling results when moving from a uniform flow/capillary rise to some kind of dual-permeability approach accounting for potential subsurface isotopic heterogeneity?

AC| This is an excellent comment. Other sources of variability may indeed have affected the variability of measured $\delta18O_{tiller}$ and $\psi_{leaf}$, such as: - The lateral heterogeneity of soil water isotopic composition (as mentioned by the referee). The idea is that water in micropores is less mobile than water in meso- and macropores, so that it is likely that, in the lower half of the profile, the capillary rise of labelled water affected the signature of water in meso- and macropores more than in micropores. If roots have more access to meso- and macropore water, then the water absorbed by roots would be isotopically enriched, as compared to the "bulk soil water" characterized experimentally. The importance of this possible bias depends on soil texture and heterogeneity (e.g. existence of more isolated "pockets" of soil or compact clusters), as well as on the speed of water mixing between mobile and immobile water fractions. Including this process in the modelling would necessitate sufficient observations to estimate the aforementioned properties, and ideally some quantification of the lateral heterogeneity of soil water isotopic composition at the micro-scale. We think it would be an excellent idea for a future study, but including it in the model in this study would involve extrapolating simulations beyond what we can justify with the measured dataset;

- The lateral heterogeneity of bulk soil water potential and soil water content (or the observational errors) may have slightly affected our estimation of soil water potential, and

in turn our predictions of root water uptake distribution. The experiment was designed to maximize vertical gradients and minimize lateral bulk soil water potential gradients by wetting soil from the bottom and letting it drain, so we consider that any lateral heterogeneity must be small. However, in the revised version of the MS we will test the impact of deviations of soil water potential, that could be due to observational errors, on our results;

- The lateral heterogeneity of soil hydraulic properties and root distribution may also have participated to the generation of lateral soil water potential heterogeneities, particularly in undisturbed soils. If one had access to data on lateral heterogeneity of soil properties and rooting density, it would be possible to simulate 3-D soil-root water flow with a tool such as R-SWMS (Javaux et al., 2008), using a randomization technique for soil properties distribution as in Kuhlmann et al. (2012), in order to obtain estimations of the relative importance of this type of heterogeneity on $\delta$18Otiller and $\psi$leaf variability. However, in this experiment we consider that the substrate and rooting heterogeneity were minimized by the sieving of the soil, and thus focused on the vertical profiling in measurements and modelling. - Overall, our treatment of the soil media in this experiment (sieving, irrigating from the bottom) makes it different from soils in natural systems, which are most likely more heterogeneous laterally. This method allowed us to study specifically the impact of the vertical component of soil water isotopic signature on tiller water isotopic signature. It also justified the use of a simplistic 1-D model adapted to the vertically resolved measurements. This will be clarified in our revisions, and the perspective of comparing bulk soil water isotopic signature to the signature of "mobile water" in meso- and macropores will be discussed.

RC| I was missing a discussion of the uncertainties regarding for example soil moisture estimates and the impact of such uncertainties for the interpretation regarding potential processes (i.e., hydraulic lift).

AC| We agree that this should be added (see second bullet point in our reply to the previous comment).

[Figure]

RC| I further think that the implications of their interesting findings (i.e., no match between the ensemble of various simulations and the observations; Fig. 5) for both field studies labelled or with natural isotope compositions and the modelling of the soil-root interactions could be made clearer. This way, the manuscript might have a higher impact and could provide recommendations to overcome limitations in observation techniques and modelling approaches.

AC| We will remove the regression lines in Figure 4 for which the p-value of the linear model was higher than 0.01, hoping that it will clarify the absence of significant linear correlation between given hydraulic (e.g., Transpiration flux T) and isotopic variables (e.g., oxygen stable isotopic composition of tiller water, $\delta$tiller). We will provide in a separate discussion section 3.3 "Progresses and Challenges in soil water isotopic labeling for RWU determination" recommendations for overcoming the aforementioned limitations.

RC| I appreciate that the authors will upload the data of the study. Are they further intending to make the model code available?

AC| We are indeed, as it may be useful to the scientific community working on such data. We will upload it as soon as the MS is accepted.

RC| Technical comments:

RC| L 77: monotonic gradient? Consider sinusoidal variability across the depth, which would cause issues of identifiability

AC| You are right! It will read: "...the soil water isotopic composition depth gradient is strong and monotonic (thus avoiding issues of identifiability)"

RC| L 80: Not only GW, also due to increasing dispersion with depth – even if the GW table is several meters deep

AC| We agree, this will be mentioned as well at this point of the introduction
RC| L 100: This paragraph is kept quite general after a very informative introduction. I suggest to be more specific and especially pose hypothesis or specific research questions.

AC| Indeed, the objectives were not clearly stated in our initial submission. We will write: "Building on the work of Meunier et al. (2017a), the objective of the present study is to (i) model in a physically-based manner (i.e., by accounting for soil and plant environmental factors) the temporal dynamics of the isotopic composition of RWU of a population of Festuca arundinacae cv Soni (tall fescue) during a semi-controlled experiment following an isotopic labeling pulse of deep soil water, (ii) investigate the implication of the model-to-data fit quality in terms of meaningfulness of the isotopic information to reconstruct RWU profiles, and finally (iii) confront the simulated root water uptake profiles with estimations obtained on basis of isotopic information alone (i.e., provided by a Bayesian mixing model)."

RC| L 117: Since you provide the variable and unit for soil moisture, you probably should also add that to matric potential.

AC| It will be done, thank you!

RC| L 140: replace "isotopic" with "$\delta 18O$"

AC| Consider it done as well.

RC| L 140: How was the sampling done? Soil corer? How much soil was sampled?

AC| Soil was sampled before (DaS 166 - 15:45) and after labeling on DaS 167 - 07:00, DaS 167 - 17:00 and DaS 168 - 05:00 using a 2 cm diameter auger through the transparent polycarbonate side of the rhizotron. This will be reported in the revised version of the manuscript.

RC| L 149: provide info about temperature, applied vacuum and time of extraction

AC| Water from plant (i.e., tillers and leaves) and soil samples were extracted by vacuum distillation for 14 to 16 hours depending on the sample mass (e.g., ranging between 18 to 28 g for soil) at temperatures of 60 and 90°C, respectively. The residual water vapor pressure at the end of each successful extraction procedure invariably reached 10–1 mbar. This will be specified as such in the revised version of the manuscript.

RC| L158: Not sure what "(95 m root (g root)−−1)." Means

AC| We will remove "root" from the mention of the dimension for clarifications, so it will simply read "m g–1".

RC| Figure 1: The circles connecting the bottom of the profile of Figure 1a and the histogram of 1c are more confusing than helping. I suggest to get rid of them. The same would apply for the arrow connecting to 1b.

AC| Done. Fig. 1 will be changed accordingly!

RC| L 172: All variables should be explained here. For example Lpr is explained in L 216

AC| That is right. We will explain the meaning of Lpr higher up.

RC| L 181: The variable "n" should be briefly explained as one of the MVG parameters. Also, consider adding n and Sej to the list of variables.

AC| We agree with the referee and will make the suggested changes in the revised version of the MS.

RC| L 209: Please define conditions for exudation. I believe it is for Sj<0, but not sure.

AC| The referee is correct. This will be clarified in the revised version of the MS.

RC| L 239: I do not see how the soil moisture varied notably at 1.3 m depth. What do you mean here? How comes that you refer to 12:00 and 20:00 on DaS 167, while that is not shown in Figure 2a?

AC| Thank you, this will be corrected as: "Soil moisture remained unchanged in the top 25 cm during the sampling period ($\theta$ = 0.08 ±0.00 m3 m–3) as well as at –1.30 m from DaS 166 - 15:45 to DaS 168 - 05:00 ($\theta$ = 0.33 ±0.01 m3 m–3)."

RC| L 243: Again, you refer to a time (7:30), which is not shown in the Figure and you should refer to it as soil labelled" and not "soil" to be consistent with Figure 2.

AC| Indeed! This will be also corrected as "$\delta$soil reached a value of 36.9 ‰ at –1.50 m on DaS 167 - 17:00."

RC| L 244: "lead us to assume" or "leads to the assumption" AC| Thank you. We will take you first proposition.

RC| L 262: It is unclear which of the correlations are describe a significant relationship. I suggest to only draw the regression lines for significant relationships in Figure 4.

AC| Thank you for this suggestion: we will remove the regression lines for which the p-value of the linear model was higher than 0.01 and indicate this also in the caption of Figure 4:

RC| L 281: replace "et" with "and"

AC| Thank you, this will be done.

RC| L 298: Unclear what is meant with "over all dataset". I believe you mean the 60 different root system classes. Please be more specific.

AC| That is right. We will clarify the sentence as follows: "However the predicted versus observed average $\delta$tiller and its standard deviation including all plant classes and observation times were not significantly different (. . .)".

RC| L 315: It seems to me that in-situ measurements would overcome these limitations. One could sample in parallel several plants and thus, observe the temporal dynamics at individual plant level.

AC| We could not agree more! We will mention these new methodological developments in a dedicated new subsection 3.3 "Progresses and Challenges in soil water isotopic labeling for RWU determination"

RC| L 319: What is the expected accuracy of your volumetric soil moisture measurements. Given that you derived this from gravimetric water content and a bulk density, which was assumed to be constant in the repacked soil. However, relatively small differences in bulk density of just a few g cm$-3$will affect the estimates of the volumetric water content. It would be good to account for such uncertainties in this discussion.

AC| The hypothesis of a constant value for b across the reconstructed soil profile could be validated from the quality of the linear fit (coefficient of determination R2 = 1.0) between the $\theta$ values measured by the sensors at the six available depths and (–0.05, –0.10, –0.30, –0.60, – 1.05 and –1.30 m) and those computed from $\theta$grav. We will add this information to the text. Yet, the impact of observational errors will be investigated as a sensitivity analysis in the revised MS.

RC| L 325: What do you mean with "significantly higher"? Did you apply a statistical test? I believe that you mean that the difference is higher than the measurement uncertainty.

AC| Yes, the p-value is 1.4e-04, which we will add to the revised version of the MS.

RC| Figure 6b: The title says "standard error", but the caption says "standard deviation". Which one is it? Please correct.

AC| It is standard deviation, thank you for spotting this typo. We will update Figure 6b accordingly.

RC| L 360: the upper half of the soil profile

AC| Done, thank you!

RC| L 367: "water addition is localized and not broadcasted in the soil" is unclear. What
do you mean with "broadcasted"?

AC| We propose not to use the term "broadcasted" anymore and to write instead: "This case study highlights (i) the potential limitations of water isotopic labeling techniques for studying RWU: the soil water isotopic artificial gradients induced from water addition result in an improvement in RWU profiles determination to the condition that they are properly characterized spatially and temporally."

RC| L 370: "simple"? In addition to the usual struggle of assessing meaningful MVG parameters to describe the soil water transport, also like for example Lpr and Kaxial are needed, which are not easily derived, but its estimation adds to the uncertainty of the uptake depths.

AC| We meant "simple soil-root model", relative to (i) complex soil-root models, which include more parameters (e.g. profile of root hydraulic properties changing with root segment age, etc.), and (ii) absent soil-root models, in the typical Bayesian approach. We will clarify that more measurements are needed than with no soil-root model. Extra measurements could be limited if appropriate assumptions on the model parameters can be done (e.g. using soil pedotransfer functions, root hydraulic properties reported in the literature, etc.).

Javaux, M., Schroder, T., Vanderborght, J., and Vereecken, H.: Use of a three-dimensional detailed modeling approach for predicting root water uptake, Vadose Zone J., 7, 1079-1088, doi:10.2136/Vzj2007.0115, 2008. Kuhlmann, A., Neuweiler, I., van der Zee, S. E. A. T. M., and Helmig, R.: Influence of soil structure and root water uptake strategy on unsaturated flow in heterogeneous media, Water Resour. Res., 48, doi:10.1029/2011wr010651, 2012. Meunier, F., Rothfuss, Y., Bariac, T., Biron, P., Durand, J.-L., Richard, P., Couvreur, V., J, V., and Javaux, M.: Measuring and modeling Hydraulic Lift of Lolium multiflorum using stable water isotopes, Vadose Zone J., doi:10.2136/vzj2016.12.0134, 2017a.

[Figure]

543, 2019.

---

## Author Comment (AC2) · 10 Dec 2019

Authors responses to Referee Dr. Matthias Beyer

RC| The manuscript hess-2019-543 'Disentangling temporal and population variability in plant root water uptake from stable isotopic analysis: a labeling study' by Couvreur et al. present a lab-/field- and model-based study of root water uptake during an artificial tracer experiment, where the soil is wetted from below (as opposed to often, via irrigation). They support their isotope analysis by hydraulic measures in order to provide a holistic understanding of RWU.

[Figure]

The authors address the urgent and contemporary need for increasing the reliability of RWU models and improve the understanding root water uptake patterns. It has been often proposed to combine hydraulic, water isotope and other information in order to do so, the presented study is in my opinion a holistic and promising approach. The rollercoaster vs. swarm hypothesis is also a good idea, though (as the authors state themselves) it should be validated further. It is also great that both data from an experiment and modeling are provided, rather than only one of the two. This manuscript is well prepared, and the topic is highly relevant. The figures are suitable and well-explained.

I highly recommend this manuscript for HESS, though I have a number of comments/questions that might help to improve this manuscript further. In brief, a few general comments, which are all rather minor:

AC| Dear Matthias, we thank you for the time you spent in carefully revising our manuscript! We hope that we have sufficiently addressed the issues you raised in our revised version.

RC| - The discussion on hydraulic redistribution should be strengthened. Do the authors see a clear sign or not? I think strengthening this part would be of utmost interest for many people from the ecohydrological community.

AC| This is indeed an important part of the discussion. In the revised manuscript, we will clarify in section 3.2.2 that all measurable signs of hydraulic redistribution are positive (local increase of soil water content, local enrichment of water isotopic signature) and converge with independent simulated results (water exuded at the same time and location, at a rate compatible with measurements) to yield a robust "yes we think that hydraulic lift was happening at that time at that depth". Our approach will also be strengthened by evaluating the impact of the observational error on our predictions.

RC| - When reading the results and discussion section, I realized that there are very small differences discussed in the manuscript (e.g. 0,41 per mill, 1 per mil, etc....). I

think it is necessary to think about uncertainties in that respect and really decide which of the differences are likely 'true' differences or simply within the variability/uncertainty.

AC| We will discuss in further detail (please see our answers to your specific comments) the problematic of meaningfulness of our isotope data, i.e., whether these differences of isotopic compositions are the result of given processes or the mere translation of e.g., soil lateral heterogeneity.

RC| - I find the discussion of the physical experiment slightly too weak compared to the results drawn from the modeling (I also indicated this in the detailed comments below).

AC| We will strengthen the discussion of the experimental experiment (see answer to your specific comment to L247-249)

RC| - The authors use $\delta$tiller, etc. without providing the water isotope (e.g.$\delta$tiller18O). I think this is important to clarify (it was only 18O used, correct?) starting with the symbol description. Why was only oxygen-18 used? (and not 2H in addition?)

AC| For clarification, we will systematically add "oxygen" before "isotopic composition" throughout the manuscript as well as in the "List of variables with symbols and units" (Page 2).

We only measured the water $\delta$18O with our IRMS ("Isoprep 18 - Optima, Fison, Great-Britain") and not water $\delta$2H and $\delta$18O simultaneously with, e.g., a laser spectrometer for two reasons: (1) to the contrary of laser spectrometers, IRMS are not affected by the presence of volatile organic substances which should be present in the distillated water from soil and plant samples. (2) The added information on $\delta$2H profiles should not be discriminating for determination of RWU profiles as $\delta$2H remains constant in the lower half of the soil profile (mostly contributing to RWU) which is influenced by labeling.

RC| - Will the model be made publicly available? It would be very interesting to apply the model with other datasets (e.g. some in situ datasets of joint soil and plant water

isotopes)

AC| We are indeed willing to make the code open source, as it may be useful to the scientific community working on such data. We will upload it as soon as the MS is accepted.

RC| I wish the authors good luck and look forward to the final publication. Greetings and best wishes, Matthias Beyer

Detailed comments:

RC| - Abstract is very well written

AC| Thank you :)

RC| - L.78/79: depends on how deep the groundwater table is. In thick unsaturated zones, often mixing of old water is also a reason. Further, over short time periods a seasonal pattern might persist in the soil

AC| We agree! If soil water (and eventually groundwater) is replenished by rain events of which the isotopic compositions is highly dynamic in time, it can generally lead to issues of identifiability. This will be added in the revised version.

RC| - L.140: in oxygen-18 I guess? Could the authors please add this information?

AC| Consider it done!

RC| - L.149: Can the authors please add specifics on the extraction? (Extraction temperature and time for soil and plant samples, how was complete extraction assessed?)The community has been asking in many occasions to provide more transparency of extraction procedures; hence it would be appreciable to add this information.

AC| Water from plant (i.e., tillers and leaves) and soil samples was extracted by vacuum distillation (applied vacuum: 10–3 mbar) at temperatures of 60 and 90°C, respectively.

In addition, complete extraction was assessed based on the comparison of sample weight loss during distillation and mass of collected distillated water. This information will be added in the revised version of the manuscript.

RC| - L.152/153: the loss of mass would also include evaporation; was this neglected (please clarify) [I see that this is mentioned later in the text, but perhaps better to clarify here]

AC| Yes, thank you. It will be clarified, i.e. "transpiration (m d–1)" will be replaced by "evapotranspiration rate loss (in m d–1)" in the revised text.

RC| - L.163: literature- Chapter 2.4: The explanation and equations make sense to me, but for a detailed evaluation and/or comments on the equations a true modeler might be considered (e.g. M. Cuntz, Wingate/Ogee group)

AC| This is true! We have already received a comprehensive review from referee #1 on the modeling aspects of our work which we hope to have properly addressed in our answer.

RC| - Chapter 3.1.1: Figure 2 is mentioned first in the text, then Figure 1....hence, those might be switched Results/Discussion

AC| We make reference to Fig. 1 at in Chapter "2.4 Modeling of RWU and $\delta$tiller"), thus before citing Fig. 2 (Chapter 3.1.1).

RC| - L.243-247 and Fig. 2: There were two soil moisture profiles measured, but only one is shown in Fig.2 (or is that averaged over the two?) I am not sure if that justification that no evaporation was present is sufficient, as the moisture profiles oscillate greatly and over one or two days the effect of evaporation might be minimal (which on the other hand supports the assumption ET=T). Still, evaporation is probably occurring (though at a low rate).

AC| There was 1 profile taken per sampling time, thus four profiles are shown in Figure 2: DaS 166 - 15:45 (orange line), DaS 167 - 07:00 (blue), DaS 167 - 15:45 (red), and

[Figure]

DaS 168 - 05:00 (black) We agree with the reviewer that evaporation could have been partly the reason of the observed differences in water content at the soil surface across sampling times, the other reason being the lateral heterogeneity. We can only make the assumption that evapotranspiration = transpiration, assumption that we carefully mention, based also on the high computed value for soil water surface tension.

RC| - L.247-249: Yes, but these three options should be discussed by the authors

AC| We will strengthen the discussion at carefully discuss these options, thank you.

RC| - L. 250: minimal minimum instead maximal maximum

AC| Thank you. It will be done!

RC| - L. 252: delete level

AC| We propose to replace "level" by "value". Thank you.

RC| - 3.1.2: Again, if results/discussion is mixed here, those differences and diurnal patterns should be discussed and explained here

AC| We agree that section 3.1.2 (as well as 3.1.1) stays rather descriptive. It is the case because we choose to discuss both soil and plant isotopic data in section 3.1.3 by cross-comparing them with soil and plant hydraulic data.

RC| - L.269: 'Rayleigh distillation corrections' – this is not explained in the methods. Could the authors provide details on these corrections and/or provide a citation?

AC| We will add two references to these corrections and how they should be applied: "Galewsky, J., Steen-Larsen, H. C., Field, R. D., Worden, J., Risi, C., and Schneider, M.: Stable isotopes in atmospheric water vapor and applications to the hydrologic cycle, Rev. Geophys., 54, 809-865, doi:10.1002/2015rg000512, 2016." "Dansgaard, W.: Stable Isotopes in Precipitation, Tellus, 16, 436-468, doi:10.1111/j.2153-3490.1964.tb00181.x, 1964."

RC| - 3.1.3: Well-written and explained

AC| Thank you!

RC| - L.298: yes, but still: 3 per mill is notable for 18O...

AC| Indeed, we agree with this comment. We will mention that a difference of 2.9 ‰ between simulated and measured mean $\delta$tiller is notable, though relatively small compared to the datasets standard deviations (8.4 ‰ and to the isotopic ratio of the labelled water (464 ‰ non-labelled soil water isotopic ratio between −7.4 ‰ and 1.3 ‰. Statistically we could not systematically conclude that simulated and measured $\delta$tiller differed. By drawing randomly simulated $\delta$tiller in 3 plants at each time step (as in the measurements), comparing the overall distributions of measured and simulated pooled $\delta$tiller with an ANOVA analysis, and repeating the random drawings for all 40 observation times 100 times, measured and simulated $\delta$tiller distributions were not statistically different in 92% of drawings (P>0.01). We will reformulate the sentence as: "The predicted versus observed $\delta$tiller distributions for the overall dataset differed noticeably but not significantly (6.6 $\pm$ 8.4 ‰ and 3.7 $\pm$ 8.4 ‰ respectively) when pooling 3 simulated $\delta$tiller randomly at each observation time, as in measurements (P>0.01 in 92 cases out of 100 repeated drawings)"

RC| - L.325: I am not sure if an 0,9 per mil increase is significant...were replicates taken for each soil depth? What is the std of those (-often this can be in that range already)....if no replicates were taken, this might be well within the uncertainty rather than a true increase

AC| The observed $\delta$soil at the first three observation times are -7.17 ‰ -7.00 ‰ and -7.21 ‰We confirm that it differs from -6.2 ‰ with an ANOVA analysis (P<0.01). The p-value will be provided in the revised version of the manuscript

RC| - L.327 depths instead heights

AC| It will be done. Thank you!

RC| - L.345 model instead models

AC| It will be done. Thank you!

RC| - L. 360 upper half instead first half

AC| It will be done. Thank you!

RC| - L.363-365: But couldn't this be implemented to the Bayesian approach via the construction of priors?

AC| This is a very keen remark. It is true that we decided to go for a "flat Dirichlet a priori rRWU distribution (i.e., rRWUJ=1/10)" and we were missing an explanation on why we did not implement the construction of priors. The outcome of the statistical model may indeed significantly depend on the definition of the a priori relative RWU profile. In the present study, we set it to follow a "flat" distribution (i.e., rRWUJ = 1/10, see Appendix E), in other word, each layer was initially defined to contribute equally to RWU. To the contrary of other studies (e.g., Mahindawansha et al., 2018), where the a priori rRWU profile was empirically constructed on basis of soil water content and root length density profiles, we decided not to further arbitrarily constrain the Bayesian model for the sake of comparison with the physically-based soil-root model.. This will be added in the revised version of our manuscript.

RC| Figure 2: RC| - it's 18O data shown, could this be added to the title (instead of only delta)...OK it's in the figure description, still...

AC| We hope mention of "oxygen" in the title and now repeatedly throughout the manuscript will clarified this.

RC| - why is matric potential 'calculated' shown if it was measured?

AC| $\psi$soil was calculated on basis of $\theta$ data, and not directly measured. We propose to clarify this confusion by moving the mention of soil matric potential to section 2.2. In addition, we will add "Measured" in Fig. 2's caption.

RC| - Not sure if the inset graphic for the water content is helping the figure

AC| This is true now that you mention it! The inset will be removed from Fig. 2 in the revised version.

Mahindawansha, A., Orlowski, N., Kraft, P., Rothfuss, Y., Racela, H., and Breuer, L.: Quantification of plant water uptake by water stable isotopes in rice paddy systems, Plant Soil, doi:10.1007/s11104-018-3693-7, 2018.

---

## Author Comment (AC3) · 18 Dec 2019

Authors responses to Referee#3

RC| This manuscript compares two alternative modeling strategies for deriving the sink term (root water uptake) in a controlled ecotron experiment. Strategy 1 uses a simplified root water uptake model which however incorporates the main features of the three dimensional soil water flow, including hydraulic redistribution. The unknown model parameters are calibrated based on isotope data in the tiller and leaf water potentials. Strategy 2 derives root water uptake based on isotope data using Bayesian inference.

The authors find that the results between the two strategies diverge. They show that Bayesian inference yields unphysical fluxes. Based on the model results they conclude that spatial variation ("swarm-like") in tiller isotopic signal is misinterpreted as a strongly fluctuating time series, whereas it actually reflects the different rooting depths of plant individuals. Additionally, they argue that both the root water uptake model and the soil moisture time series suggest hydraulic lift, which cannot be captured by the Bayesian inference based on isotope data alone. Therefore, they conclude that the results obtained based on Bayesian inference could be due to an artifact.

This is a valuable contribution illustrating how sampling choice may affect the interpretation of isotope data. Especially the application of a straightforward process model for comparison with the Bayesian inference together with the dense measurements are extremely helpful to explain the shortcomings of deriving uptake profiles based on isotope data alone. The case is well argued and the methods are sound. I feel the manuscript has potential to making an impact and will find strong interest in the readership of HESS. The paper is mostly well structured, although I have some concerns with the Abstract and Methods section, as well as with some formulations (see below).

AC| Dear reviewer, we thank you for the detailed list of specific comments, for which we hope you will find our answers satisfactory.

RC| I have two general concerns, and a number of editorial remarks (below). The investigated case is a particular one, e.g. with a strong labelling pulse added below the rooting zone. This needs to be explicitly stated and the manuscript should discuss in which other situations such a strong influence of spatial variation is to be expected (and where it is not a concern).

AC| To address this general concern, we propose to add a section 3.3 entitled "Progresses and Challenges in soil water isotopic labeling for RWU determination" to the discussion with the following text:

"What can we draw from our findings about the use of isotopic labeling pulses for

RWU analysis? Often in the field, the vertical dynamics of both soil water oxygen and hydrogen isotopic compositions are not strong enough (or show convolutions leading to issues of identifiability) for partitioning RWU among different contributing soil water sources. As a consequence, we unfortunately cannot make use of the natural variability in isotopic abundances for deciphering soil-root transfer processes (Beyer et al., 2018; Burgess et al., 2000). To address this limitation of the isotopic methodology, labeling pulses have been applied locally at different depths in the soil profile (e.g., Beyer et al., 2016) or at the soil upper/lower boundaries under both lab and field conditions by mimicking rain events (e.g., Piayda et al., 2017) and/or rise of the groundwater table (Meunier et al., 2017).

After labeling, we are faced with two problems: (i) the labeling pulse might enhance RWU at the labeling location if the volume of added water significantly changes the value of soil water content. It therefore poses the question of the meaningfulness of the derived RWU profiles, and this independently from the model used (i.e., physically-based soil-root model or statistical multi-source mixing model). In other worlds: are we observing a natural RWU behavior of the plant individual or population or are we seeing the influence of the labeling pulse? Certainly a way to move forward is environmental observatories such as ecotron and field lysimeters (e.g., Groh et al., 2018; Benettin et al., 2018) that provide means to better constrain hydraulic boundary conditions and reduced their isotopic heterogeneity. They allow for a mechanistically and holistic understanding of soil-root processes from stable isotopic analysis.

Another topic of concern is (ii) the difficulty to properly observe in situ (1) the propagation of the labeling pulse in the soil after application and (2) the temporal dynamics of the plant RWU isotopic composition. Beyer and Dubbert (2019) presented a comprehensive review on recent isotopic techniques for non-destructive, online, and continuous determination of soil and plant water isotopic compositions (Rothfuss et al., 2013; Quade et al., 2019; Volkmann et al., 2016) as alternatives of the widely used combination of destructive sampling and offline isotopic analysis following cryogenic vacuum

extraction (Orlowski et al., 2016) or liquid-vapor direct equilibration (Wassenaar et al., 2008). These techniques have the potential for a paradigm change in isotopic studies on RWU processes to the condition that, e.g., isotopic effects during sample collection are fully understood.

The present study highlights the need not to "trust" our isotope data alone and always complement them by information on environmental factors as well as on soil and plant water status to go beyond the simple application of statistical models. This is especially the case in the framework of labeling studies where strong soil water isotopic gradients may induce strong dynamics of the RWU isotopic composition from a low variability of rooting depths."

RC| The study suffers from lack of opportunity for validation: The heterogenous rooting depths cannot be measured in situ and therefore it remains a hypothesis. This is ok. But it requires diligent consideration of other assumptions of the model that may have had a similar effect on the model result. How about the inherent assumptions of a big leaf? Could individual differences in leaf development incur similar results? Those considerations need to enter more than now into the discussion. I propose adding a section dealing with the effect of inherent model assumptions.

AC| We totally agree with the referee. Our analysis shows that the tiller water isotopic signature is very sensitive to rooting depth in that kind of labelling experiment, generating heterogeneity in the aforementioned signatures of the population which in many cases could be confused for temporal variability of tiller water isotopic signature and root water uptake depth. We think this is an important in silico result and clarified in the discussion that its experimental validation would necessitate to estimate the variability of rooting depth in situ, which is currently not possible. Future studies using transparent soils, as in Downie et al. (2012) might take us one step closer to a validation, though distinguishing roots from different plants, for instance with fluorescent roots in mutant lines, would be another challenge.

The assumption that all plants transpire at the same rate ("big leaf") pointed out by the referee was not discussed in the manuscript, though it would be an interesting piece of discussion. Non-uniform patterns of transpiration within the plant population would affect two of our simulated variables: the isotopic signature of the tiller water and the leaf water potential.

- In our analysis, we have shown that large temporal fluctuations of transpiration (Figure 3 panel b) barely affect temporal fluctuations of the isotopic signature of tiller water (continuous grey lines Figure 5 panel a). Hence, we expect that the spatial heterogeneity of transpiration, likely smaller than its temporal heterogeneity, would have an even smaller impact on tiller water isotopic signature heterogeneity. Given the low sensitivity of tiller water isotopic signature to transpiration rate, and the lack of data on transpiration rate spatial heterogeneity, we think it is not worth developing additional simulations to study the effect of this factor in this manuscript. However, we added a paragraph on it in the discussion.

- Unlike the tiller water isotopic signature, leaf water potential turned out to be very sensitive to transpiration rate in our simulations (see temporal fluctuations of grey lines in Figure 5 panel c) and not very sensitive to root distribution (see small variations of leaf water potential across individuals in Figure 5 panel c). This high sensitivity of leaf water potential to transpiration suggests that in this setup the hydraulic conductance of the soil-root system limits shoot water supply more than the distribution of roots Sulis et al. (2019). A consequence of the high sensitivity of leaf water potential to transpiration rate is that if transpiration rate was spatially heterogeneous, substantial deviations of the measured leaf water potential would have been observed, relative to the simulated "baseline" leaf water potential (i.e. leaf water potential in case of uniform transpiration rate across the plant population). Simulated baseline leaf water potentials (for uniform transpiration rates) are shown as grey lines in Figure 5 panel c, and measured leaf water potentials as a green line in the same panel. We were positively surprised to find out that the simulated baseline leaf water potentials fit the measured temporal fluctu-

ations of leaf water potential quite well under the assumption of uniform transpiration rate, despite the high sensitivity of leaf water potential to transpiration rate. This result reinforces the idea that transpiration rate was likely not spatially heterogeneous among the plant population. In consequence, we think that transpiration rate was rather uniform among the plant population (so that the "big leaf" approach was justified), and therefore, the tiller water isotopic signature, whose sensitivity to transpiration rate is already very low, was likely not affected by transpiration rate heterogeneity. This piece of discussion will be added in the revised manuscript.

RC| Detailed comments:

RC| Abstract: Line 18-19: This sentence sounds vague, e.g. "semi-controlled" and "such variables", please formulate more specifically.

AC| We will list in the few sentences after the monitored variables in question. We will remove "semi-controlled" (indeed a vague term) and "such variables" from the text.

RC| Line 23-24: "results underlined the discrepancy.." At this point unclear what is meant

AC| We will put emphasis on "temporal disconnection" instead of "discrepancy".

RC| Line 29-30: The sentence starting with "The physical model.. "is difficult to understand, please reformulate

AC| We will split the sentence in two and name two examples of variables instead of referring to them by "the former" and "the latter".

RC| Line 33: "local increase..." this results is not stated earlier and at this position confusing.

AC| We will not refer to "the" local increase – which indeed was not mentioned earlier – anymore.

RC| Lines 35-62 List of variables. Some variables are missing, please complete. Also

I propose erasing all the repetitions of "units of".

AC| Done. We will add the missing two variables, namely the soil hydraulic conductivity parameter ($\lambda$, dimensionless) and the soil relative water content (Sej). In addition, repetitions of "in units of" will be removed. Instead, we will add headers to the table (i.e., "Name", "Symbol", and "Units").

RC| Later in the paper, it will be useful to express volumetric water content as vol-% and I propose adding it here.

AC| Please see our answer to your specific comment regarding L329ff below.

RC| Introduction

RC| Line 75-76: The description of the "mean value of...weighted by" is confusing, please rephrase

AC| Consider it done!

RC| Material and Methods

RC| Line 114: Please mention that CS616 is a time domain sensor (TDR). Also, reflectometer is a correct, but awkward term for soil moisture sensor. I propose using the latter, just to avoid confusion.

AC| Consider it done!

RC| Line 123: The description of the soil is confusing. Is District Cambisol a typo for "Dystric Cambisol". Otherwise, I am not aware what a District Cambisol would refer to, please explain. Besides, Cambisol refers to a soil in situ and after specific pedogenesis which is completely removed in your experiment. Maybe say "The soil originates from a xx Cambisol". Also, does the bulk density refer to the original soil, and is it required to be mentioned?

AC| There is no typo here. But we agree that it was not properly formulated, and that

the substrate which we filled the rhizotron with could not possibly be referred to as a "cambisol". Therefore we will write that the soil substrate originates from the Lp horizon of an agricultural field part of the Observatory of Environment Research (ORE), INRA Lusignan, France (0°60W, 46°250N) which is classified as District Cambisol (particle size distribution: sand 15%, silt 65%, clay 20%).

RC| Line 125: Add "layer" before "by"

AC| We would propose the following reformulation: "450 kg of soil was filled in the rhizotron by 0.10 m increment a. . ."

RC| Line 128: sols-PST55 sensors are missing above, where installation depths were mentioned. Add there. Please add where they are installed. Line 130: "between its position and measured soil water content" unclear, please rephrase

AC| The retention curves were determined in situ in the same type of macro-rhizotron during another experiment (at the same soil bulk density) of which the results were published by Meunier et al. (2017a). In order to clarify this, we will not mention the type of sensors used – which was indeed misleading – and write: "The closed-form soil water retention curve van Genuchten (1980) was derived in a previous study by Meunier et al. (2017a) from synchronous measurements of soil water content and matric potential from saturated to residual water content (see Appendix B for its hydraulic parameters)."

RC| Lines 132-136: Please shortly state: Were the plants watered? What was the lower boundary condition?

AC| During a period of 165 day following seeding, the tall fescue cover was exclusively watered from the reservoir with local water in order to (i) keep the soil bottom layer (< −1.3 m) close to water saturation, and to (ii) not to disrupt the natural soil water $\delta$18O profile.Done. We will add this information in the revised manuscript.

RC| Line 139: Do you mean "sides" instead of "slides"

[Figure]

AC| Yes, exactly, many thanks for finding this typo!

RC| Line 144: "three plants were sampled" - does it mean the entire plant or some leafs?

AC| We sampled the entire plant. We will insert "whole" after "three".

RC| Line 148-149 I believe you mean "from the atmosphere surrounding the rhizotron". Also, I am assuming the latter means the ambient air in the lab? Would be good to specify.

AC| Thank you for these propositions!

RC| Line 162: "60 tall festucae root systems .." Why 60 plants?

AC| This choice was arbitrary. We estimated that there were about 1500 plants per square meter in the rhizotron, so that there would be 300 plants on total in the experiment. Running simulations for 300 plants would have required a lot of computational resources. That is why we focused on a subset of 60 representative plants, that met our computational capabilities for the inverse modelling scheme. Each "representative plant" was called a "class" of plants that is included in the simulations under the form of a "big root" and "big leaf", with root lengths and transpiration rate corresponding to 5 plants for each class. This is an important point, which we will clarify in the revised manuscript.

RC| Eq. (1) Personally, I do not find this equation obvious. Please motivate the origin. Are there any other assumptions involved besides the big root one?

AC| This equation is indeed not obvious. It was derived by Meunier et al. (2017a) in its Appendix C. The reference was indeed missing. We will add it to the revised manuscript ", as derived by Meunier et al. [2017],".

RC| Line 173-174: "dimension of the domain..." I do not understand this statement

AC| The horizontal domain of simulation typically has two dimensions (X and Y), but

it some cases, the studied problem has an essentially radial dimension between bulk soil and root surface. We will simplify the text as follows: "with B (dimensionless) a geometrical factor simplifying the horizontal dimensions into radial domains between the bulk soil and root surfaces".

RC| Line 176-177: Sentence starting with "The averaged distance .." seems wrong. Maybe erase the last words? AC| The referee is correct. The reported sentence referred to , not to the average distance between roots. We will corrected the text as follows: "It can be deduced from the observed root length density (...)".

Eq (4): Se is not part of the List of variables, plus the S stated there refers to the sink term not saturation. Please use a different abbreviation.

AC| Relative water content (Sej) is both introduced in Eq (4) is defined in Eq. (5). The water "sink term" is introduced in Eq. (9).

RC| Line 188: Could not the measured root length density profiles be used?

AC| Not here because root lengths, and thus root system conductances, are class-specific. The bulk root length densities may not account for the fact that each class has its own root length and RLD at each depth.

RC| Line 191: "were derived" unclear how this was derived? Also, where was k_axial in Eq.(7) taken from? Please explain. Ok, I learn later this was calibrated. Maybe mention this here.

AC| We apologize for the confusion, and will includ this clarification at this point in the revised manuscript: "were calculated as equivalent "big root" specific axial conductance per root system class (kaxial, m4 MPa−1 d−1, to be optimized by inverse modelling) as"

RC| Line 191: "root system class" Unclear what is meant with "class".

AC| See reply to comment on simulated 60 root systems.

[Figure]

RC| Line 194: standard sink distribution is not a standard term and requires a bit more explanation to be convincing.

AC| The term was defined in the following sentence. We moved it ahead for clarity.

RC| Line 195-196: "potential difference between soil and leaf": You are dealing with a soil profile and a leaf canopy. Thus, where in the soil and leaf are you referring to. Please also translate to what this means for your experimental setup either here or in the discussion.

AC| That is a good point. The leaf is a "big leaf", and the soil water potential used in the definition of Ksoil-root is the SSF-averaged bulk soil water potential. We will clarify it in the text: "The variable SSF is the relative distribution of water uptake in each soil layer under vertically homogeneous soil water potential conditions (Couvreur et al., 2012), and Ksoil-root represents the water flow per unit water potential difference between the SSF-averaged bulk soil water potential and the "big leaf" (assuming a negligible stem hydraulic resistance)".

RC| Line 196: "assuming negligible stem conductance": Does this imply that all conductance / resistance happens in the root system? Is this a reasonable assumption?

AC| As far as grass is concerned we think so. The main hydraulic resistances between the bulk soil and the leaf insertion (where leaf water potential is measured) being the drying soil and the root radial resistance (Steudle and Peterson, 1998). Reference will be added in the text.

RC| Line 202: "class" unclear

AC| See previous reply about the 60 root systems.

RC| Line 205: "where axial conductances" this comes too late, please move up.

AC| It was in the right place, but we will rewrite the sentence for clarification as "where K_(soil-root) was assumed to control the compensatory RWU which arise from a het-

erogeneously distributed soil water potential, due to large axial conductances (Couvreur et al., 2012)".

RC| Line 218: I propose moving the inverse modeling procedure out the appendix and add it to the main text. It is important information.

AC| We think that the Material and methods are very dense already, and since the inverse modelling method is a state-of-the-art method in modelling, we think it would be better to leave its detailed description in the appendix.

RC| Line 226: Not sure what is meant with "ten identified potential water sources" .. "10 distinct soil layers" Can you be more specific?

AC| These water sources were defined to originate from 10 distinct soil layers (0.00-0.03, 0.03-0.07, 0.07-0.15, 0.15-0.30, 0.30-0.60, 0.60-0.90, 0.90-1.20, 1.20-1.32, 1.32-1.37, and 1.37-1.44 m). This information be added in the revised version, thank you.

RC| Results and discussion:

RC| Lines 335-252: Small issue: Please add some paragraphs in this section.

AC| Thank you for the tip! We will split the text into three paragraphs which refer to soil water content (§1), soil water oxygen isotopic composition (§2), and root length density (§3) profiles.

RC| Line 281: Do you mean "and" instead of "et"

AC| Consider it done, thank you!

RC| Line 304: With "all the population" do you mean all individuals?

AC| Indeed. We will replace it by "all individuals" for clarification.

RC| Line 311-313: Sentence is difficult to understand, please rephrase.

AC| We will rephrase as follows: "As no correlation could be expected between the drivers, the maximum rooting depth of the sample plants and canopy transpiration rate,

our analysis explains the absence of correlation between $\delta$tiller and $\psi$leaf or transpiration rate".

RC| Lines 329ff: Since there is repeatedly reference to increasing by xx% this may be strongly confusing. Better use vol-% to be on the safe side.

AC| We would like to keep using the information of dimension for soil volumetric water content ($\theta$) rather than using a relative unit. We need this information for, e.g., explaining how we calculate $\theta$ from the soil gravimetric water content ($\theta$grav, in kg kg$-1$). We converted the vol% to m3 m$-3$ back in section 3.2.2. Thank you for pointing out these inconstancies!

RC| Line 340 Lambda is not in the list of variables.

AC| $\lambda$ will be added to the list of variables, thank you.

RC| Also, this information is very compact, and difficult to understand. Please elaborate.

AC| The following clarification will be added: "The optimal value of ksat was quite high (Table 1) but reportedly very correlated to $\lambda$ (i.e. soil unsaturated hydraulic conductivity is proportional to ksat, but also to Se$\lambda$; van Genuchten, 1980), so that the low value of the latter compensated the high value of the former, thus they should be considered as effective rather than physical parameters".

RC| Lines 345ff: I strongly recommend bringing Fig E into the main text. It is discussed and seems therefore sufficiently important. Also, because this is one of the two alternative water uptake profiles which comparison is the main motivation of the manuscript.

AC| The authors agree. Figure E will be part of the main document and become "Figure 7".

RC| Line 360: Replace "first" with "top"

AC| Done. Thank you.

RC| Line 367: Not sure what is meant with "broadcasted"

AC| We will not use the term "broadcasted" anymore and propose to write instead: "This case study highlights (i) the potential limitations of water isotopic labeling techniques for studying RWU: the soil water isotopic artificial gradients induced from water addition result in an improvement in RWU profiles determination to the condition that they are properly characterized spatially and temporally."

RC| Figure 2: I was confused about the positive water potentials. Since they were named psi I was instinctively assuming to see matric potential, but plotted are the water potentials. I propose renaming to h. If you want to stick to psi_soil because of psi_leaf(although I seriously think it would not be an issue), please obviously state the reference elevation to avoid this type of confusion.

AC| Figure 2 shows the profiles of the log transformation of soil water matric potential $\psi$soil (i.e., not the soil hydraulic head, which you term h). $\psi$soil is always negative, therefore for the log transformation we must take "$-\psi$soil". We do not mean by that that $\psi$soil was negative during the experiment.

Benettin, P., Volkmann, T. H. M., von Freyberg, J., Frentress, J., Penna, D., Dawson, T. E., and Kirchner, J.: Effects of climatic seasonality on the isotopic composition of evaporating soil waters, Hydrol. Earth Syst. Sc., 22, 2881-2890, doi:10.5194/hess-22-2881-2018, 2018.

Beyer, M., Koeniger, P., Gaj, M., Hamutoko, J. T., Wanke, H., and Himmelsbach, T.: A deuterium-based labeling technique for the investigation of rooting depths, water uptake dynamics and unsaturated zone water transport in semiarid environments, J. Hydrol., 533, 627-643, doi:10.1016/j.jhydrol.2015.12.037, 2016.

Beyer, M., Hamutoko, J. T., Wanke, H., Gaj, M., and Koeniger, P.: Examination of deep root water uptake using anomalies of soil water stable isotopes,

depth-controlled isotopic labeling and mixing models, J. Hydrol., 566, 122-136, doi:10.1016/j.jhydrol.2018.08.060, 2018.

Beyer, M., and Dubbert, M.: X Water Worlds and how to investigate them: A review and future perspective on in situ measurements of water stable isotopes in soils and plants, Hydrol. Earth Syst. Sci. Discuss., in review doi:10.5194/hess-2019-600, 2019.

Burgess, S. S. O., Adams, M. A., Turner, N. C., and Ward, B.: Characterisation of hydrogen isotope profiles in an agroforestry system: implications for tracing water sources of trees, Agric. Water Manage., 45, 229-241, doi:10.1016/S0378-3774(00)00105-0, 2000.

Couvreur, V., Vanderborght, J., and Javaux, M.: A simple three-dimensional macroscopic root water uptake model based on the hydraulic architecture approach, Hydrol. Earth Syst. Sc., 16, 2957-2971, doi:10.5194/hess-16-2957-2012, 2012.

Downie, H., Holden, N., Otten, W., Spiers, A. J., Valentine, T. A., and Dupuy, L. X.: Transparent Soil for Imaging the Rhizosphere, Plos One, 7, doi:10.1371/journal.pone.0044276, 2012.

Groh, J., Slawitsch, V., Herndl, M., Graf, A., Vereecken, H., and Putz, T.: Determining dew and hoar frost formation for a low mountain range and alpine grassland site by weighable lysimeter, J. Hydrol., 563, 372-381, doi:10.1016/j.jhydrol.2018.06.009, 2018.

Meunier, F., Rothfuss, Y., Bariac, T., Biron, P., Richard, P., Durand, J.-L., Couvreur, V., Vanderborght, J., and Javaux, M.: Measuring and Modeling Hydraulic Lift of Lolium multiflorum Using Stable Water Isotopes, Vadose Zone J., 15 pp., doi:10.2136/vzj2016.12.0134, 2017.

Meunier, F., Rothfuss, Y., Bariac, T., Biron, P., Durand, J.-L., Richard, P., Couvreur, V., J, V., and Javaux, M.: Measuring and modeling Hydraulic Lift of Lolium multiflorum using stable water isotopes, Vadose Zone J., doi:10.2136/vzj2016.12.0134, 2017a.

Orlowski, N., Pratt, D. L., and McDonnell, J. J.: Intercomparison of soil pore water extraction methods for stable isotope analysis, Hydrol. Process., 30, 3434-3449, doi:10.1002/hyp.10870, 2016.

Piayda, A., Dubbert, M., Siegwolf, R., Cuntz, M., and Werner, C.: Quantification of dynamic soil-vegetation feedbacks following an isotopically labelled precipitation pulse, Biogeosciences, 14, 2293-2306, doi:10.5194/bg-14-2293-2017, 2017.

Quade, M., Klosterhalfen, A., Graf, A., Brüggemann, N., Hermes, N., Vereecken, H., and Rothfuss, Y.: In-situ Monitoring of Soil Water Isotopic Composition for Partitioning of Evapotranspiration During One Growing Season of Sugar Beet (Beta vulgaris), Agr. Forest Meteorol., 266–267, 53–64, doi:10.1016/j.agrformet.2018.12.002, 2019.

Rothfuss, Y., Vereecken, H., and Bruggemann, N.: Monitoring water stable isotopic composition in soils using gas-permeable tubing and infrared laser absorption spectroscopy, Water Resour. Res., 49, 3747-3755, doi:10.1002/wrcr.20311, 2013.

Steudle, E., and Peterson, C. A.: How does water get through roots?, J. Exp. Bot., 49, 775-788, doi:10.1093/jxb/49.322.775, 1998.

Sulis, M., Couvreur, V., Keune, J., Cai, G. C., Trebs, I., Junk, J., Shrestha, P., Simmer, C., Kollet, S. J., Vereecken, H., and Vanderborght, J.: Incorporating a root water uptake model based on the hydraulic architecture approach in terrestrial systems simulations, Agr. Forest Meteorol., 269, 28-45, doi:10.1016/j.agrformet.2019.01.034, 2019.

van Genuchten, M. T.: A closed-form equation for predicting the hydraulic conductivity of unsaturated soils, Soil Sci. Soc. Am. J., 44, 892-898, doi:10.2136/sssaj1980.03615995004400050002x, 1980.

Volkmann, T. H., Kühnhammer, K., Herbstritt, B., Gessler, A., and Weiler, M.: A method for in situ monitoring of the isotope composition of tree xylem water using laser spectroscopy, Plant Cell Environ, doi:10.1111/pce.12725, 2016.

Wassenaar, L. I., Hendry, M. J., Chostner, V. L., and Lis, G. P.: High resolution pore water delta2H and delta18O measurements by H2O(liquid)-H2O(vapor) equilibration laser

spectroscopy, Environ. Sci. Technol., 42, 9262-9267, doi:doi.org/10.1021/es802065s, 2008.

---

## Author Response (AR1)

**Additional comment to the reviewers and Editor Prof. Markus Weiler**

Please note that we have decided to opt for another more "catchy" title: "The rollercoaster and the swarm: disentangling plant water isotopic composition variabilities in response to soil water labelling"

For the co-authors,

Valentin Couvreur and Youri Rothfuss

**Anonymous Referee #1**

Couvreur and colleagues present an interesting isotopic labelling experiment and innovative simulations of the processes in the soil-roots interactions. Their study is addressing current research gaps and will thus be of interest to the readership of HESS. The manuscript is well prepared and the figures are mostly informative. I provide two general recommendations and several minor technical comments below. I recommend publication after addressing these comments.

Dear reviewer, we thank you for your general comments as well as technical corrections of our manuscript! You can find our answers below:

General aspects:

The "rollercoaster hypothesis" and the "swarm pattern hypothesis" both focus on the variation of $\delta^{18}O$ in tiller across plants and/or over time, respectively. However, the studied system is likely to be more complex due to heterogeneity of the water flow/capillary rise. Do you see a chance to improve the modelling results when moving from a uniform flow/capillary rise to some kind of dual-permeability approach accounting for potential subsurface isotopic heterogeneity?

This is an excellent comment. Other sources of variability may indeed have affected the variability of measured $\delta^{18}O_{tiller}$ and $\psi_{leaf}$, such as:

- The lateral heterogeneity of soil water isotopic composition (as mentioned by the referee). The idea is that water in micropores is less mobile than water in meso- and macropores, so that it is likely that, in the lower half of the profile, the capillary rise of labelled water affected the signature of water in meso- and macropores more than in micropores. If roots have more access to meso- and macropore water, then the water absorbed by roots would be isotopically enriched, as compared to the "bulk soil water" characterized experimentally. The importance of this possible bias depends on soil texture and heterogeneity (e.g. existence of more isolated "pockets" of soil or compact clusters), as well as on the speed of water mixing between mobile and immobile water fractions. Including this process in the modelling would necessitate sufficient observations to estimate the aforementioned properties, and ideally some quantification of the lateral heterogeneity of soil water isotopic composition at the micro-scale. We think it would be an excellent idea for a future study, but including it in the model in this study would involve extrapolating simulations beyond what we can justify with the measured dataset;
- The lateral heterogeneity of bulk soil water potential and soil water content (or the observational errors) may have slightly affected our estimation of soil water potential, and in turn our predictions of root water uptake distribution. The experiment was designed to maximize vertical gradients and minimize lateral bulk soil water potential gradients by wetting soil from the bottom and letting it drain, so we consider that any lateral heterogeneity must be small. However, in the revised version of the MS we tested the impact on our results of deviations of soil water potential, that could be due to observational errors in soil water content measurements with the gravitational method, and found no qualitative change;
- The lateral heterogeneity of soil hydraulic properties and root distribution may also have participated to the generation of lateral soil water potential heterogeneities, particularly in undisturbed soils. If one had access to data on lateral heterogeneity of soil properties and rooting density, it would be possible to simulate 3-D soil-root water flow with a tool such as R-SWMS (Javaux et al., 2008), using a randomization technique for soil properties distribution as in Kuhlmann et al. (2012), in order to obtain estimations of the relative importance of this type of heterogeneity on $\delta^{18}O_{tiller}$ and $\psi_{leaf}$ variability. However, in this experiment we consider that the substrate and rooting heterogeneity were minimized by the sieving of the soil, and thus focused on the vertical profiling in measurements and modelling.

Overall, our treatment of the soil media in this experiment (sieving, irrigating from the bottom) makes it different from soils in natural systems, which are most likely more heterogeneous laterally. This method allowed us to study specifically the impact of the vertical component of soil water isotopic signature on tiller water isotopic signature. It also justified the use of a simplistic 1-D model adapted to the vertically resolved measurements. This was clarified

in our revisions, and the perspective of comparing bulk soil water isotopic signature to the signature of "mobile water" in meso- and macropores was discussed.
A new section "3.2.3 Other sources of variability and observational error" was added to discuss these points (L388-422):

> Our treatment of the soil media in this experiment (sieving, irrigation from the bottom) makes it laterally more homogeneous than natural soils. This method allowed us to study specifically the impact of the vertical gradients of $\delta_{soil}$ on $\delta_{tiller}$. It also justified the use of a simplistic 1-D model adapted to the vertically resolved measurements. If lateral heterogeneity of soil water content remained and was accounted for, our predictions of root water uptake distribution, $\delta_{tiller}$ and $\psi_{leaf}$ would be altered. Observational errors in the gravimetric soil water content measurement (turned into soil water potential using the soil water retention curve) would as well alter these predictions. In order to quantify the sensitivity of our simulated results to such heterogeneity or observational error, we varied the soil water content input by ± 0.02 $m^3 \, m^{-3}$ at three critical depths (-0.9, -1.1 and -1.3 m, before interpolation), at the last observation time, during which measurements and simulations suggested that hydraulic lift occurred. Our results were mostly sensitive to soil water content alterations at -0.9 m, and barely differed in response to alterations at -1.1 and -1.3 m, though the conclusions were not affected qualitatively. No statistically significant difference between predicted and observed $\delta_{tiller}$ distributions for the overall dataset could be found when pooling 3 simulated $\delta_{tiller}$ randomly at each observation time (predicted and observed $\delta_{tiller}$ distributions were closest to differ when soil water content was reduced by 0.02 $m^3 \, m^{-3}$ at 0.9 m depth; P>0.01 in 76 cases out of 100 repeated drawings). Measured and simulated $\psi_{leaf}$ remained very correlated in all cases (from $R^2$=0.69 to 0.74 when adding or removing 0.02 $m^3 \, m^{-3}$ at 0.9 m depth, respectively). Furthermore, when adding or removing 0.02 $m^3 \, m^{-3}$ at 0.9 m depth, cumulative water exudation at -0.9 m varied between 0.0019 and 0.0035 $m^3 \, m^{-3}$, uptake at -1.1 m varied between 0.0080 and 0.0108 $m^3 \, m^{-3}$, and the simulated change of $\delta_{soil}$ ranged between 0.28 and 0.40 ‰, respectively.
> Lateral heterogeneity of soil water isotopic composition may as well occur at the microscopic scale. As water in micropores is less mobile than water in meso- and macropores (Alletto et al., 2006), it is likely that, in the lower half of the profile, the capillary rise of labelled water affected the signature of water in meso- and macropores more than in micropores. If roots have more access to meso- and macropore water, then the water absorbed by roots would be isotopically enriched, as compared to the "bulk soil water" characterized experimentally. The importance of this possible bias depends on soil texture and heterogeneity (e.g. existence of more isolated "pockets" of soil or compact clusters), as well as on the speed of water mixing between mobile and immobile water fractions (Gazis and Feng, 2004). Including this process in the modelling would necessitate sufficient observations to estimate the aforementioned properties, and ideally some quantification of the lateral heterogeneity of soil water isotopic composition at the micro-scale.
> The lateral heterogeneity of soil hydraulic properties and root distribution may also have participated to the generation of lateral soil water potential heterogeneities, particularly in undisturbed soils. If one had access to data on lateral heterogeneity of soil properties and rooting density, it would be possible to simulate 3-D soil-root water flow with a tool such as R-SWMS (Javaux et al., 2008), using a randomization technique for soil properties distribution as in Kuhlmann et al. (2012), in order to obtain estimations of the relative importance of this type of heterogeneity on $\delta_{tiller}$ and $\psi_{leaf}$ variability."

I was missing a discussion of the uncertainties regarding for example soil moisture estimates and the impact of such uncertainties for the interpretation regarding potential processes (i.e., hydraulic lift).
We agree that this should be added (see second bullet point in our reply to the previous comment and first paragraph of the revision in the reply to the previous comment).

I further think that the implications of their interesting findings (i.e., no match between the ensemble of various simulations and the observations; Fig. 5) for both field studies labelled or with natural isotope compositions and the modelling of the soil-root interactions could be made clearer. This way, the manuscript might have a higher impact and could provide recommendations to overcome limitations in observation techniques and modelling approaches.
We have now removed the regression lines in Figure 4 for which the p-value of the linear model was higher than 0.01, hoping that it clarifies the absence of significant linear correlation between given hydraulic (e.g., Transpiration flux $T$) and isotopic variables (e.g., oxygen stable isotopic composition of tiller water, $\delta_{tiller}$). We provide in a separate discussion section 3.3 "Progresses and Challenges in soil water isotopic labeling for RWU determination" recommendations to overcome the aforementioned limitations.

I appreciate that the authors will upload the data of the study. Are they further intending to make the model code available?
We are indeed, as it may be useful to the scientific community working on such data. We will upload it as soon as the MS is accepted.

Technical comments:

L 77: monotonic gradient? Consider sinusoidal variability across the depth, which would cause issues of identifiability
You are right! It reads (now L83-84):
"…the soil water isotopic composition depth gradient is strong and monotonic (thus avoiding issues of identifiability)"

L 80: Not only GW, also due to increasing dispersion with depth – even if the GW table is several meters deep
The impact of dispersion was added in the revised text (now L86-87):
"…(due to the isotopic influence of the groundwater table and increasing dispersion with depth)"

L 100: This paragraph is kept quite general after a very informative introduction. I suggest to be more specific and especially pose hypothesis or specific research questions.
Indeed, the objectives were not clearly stated in our initial submission. We now write (L107-113):
"Building on the work of Meunier et al. (2017a), the objective of the present study is to (i) model in a physically-based manner (i.e., by accounting for soil and plant environmental factors) the temporal dynamics of the isotopic composition of RWU of a population of *Festuca arundinacae* cv Soni (tall fescue) during a semi-controlled experiment following an isotopic labeling pulse of deep soil water, (ii) investigate the implication of the model-to-data fit quality in terms of meaningfulness of the isotopic information to reconstruct RWU profiles, and finally (iii) confront the simulated root water uptake profiles with estimations obtained on basis of isotopic information alone (i.e., provided by a Bayesian mixing model)."

L 117: Since you provide the variable and unit for soil moisture, you probably should also add that to matric potential.
Done. It now reads (L127):
"…volumetric content ($\theta$, in $m^3\ m^{-3}$) and matric potential ($\psi_{soil}$, in MPa)."

L 140: replace "isotopic" with "δ18O"
Done. It now reads (L154-155):
"That same day at 17:00, the reservoir's water was labelled and its $\delta^{18}O$ measured at +470 ‰."

L 140: How was the sampling done? Soil corer? How much soil was sampled?
We now add (L155-157):

"Soil was sampled before (DaS 166 - 15:45) and after labeling on DaS 167 - 07:00, DaS 167 - 17:00 and DaS 168 - 05:00 using a 2 cm diameter auger through the transparent polycarbonate side of the rhizotron…"

L 149: provide info about temperature, applied vacuum and time of extraction
It now reads (L168-172):
"Water from plant (i.e., tillers and leaves) and soil samples were extracted by vacuum distillation for 14 to 16 hours depending on the sample mass (e.g., ranging between 18 to 28 g for soil) at temperatures of 60 and 90°C, respectively. The residual water vapor pressure at the end of each successful extraction procedure invariably reached $10^{-1}$ mbar. The oxygen isotopic compositions of tiller, leaf, and soil water (i.e., $\delta_{tiller}$, $\delta_{leaf}$, and $\delta_{soil}$) together with…"

L158: Not sure what "(95 m root (g root)$^{-1}$)." Means
We removed "root" from the mention of the dimension for clarifications, so now it now reads (L180-181):
"…specifically for tall fescue (95 m g$^{-1}$)".

Figure 1: The circles connecting the bottom of the profile of Figure 1a and the histogram of 1c are more confusing than helping. I suggest to get rid of them. The same would apply for the arrow connecting to 1b.
Done. We updated Fig. 1:

[Figure]

L 172: All variables should be explained here. For example Lpr is explained in L 216
That is right. We added the meaning of all terms below equation 1 (L195-196):
"with $r_{root}$ (m) the root radius, $l_{root,i,j}$ (m) the root length of plants of group $i$ in soil layer $j$, $L_{pr}$ (m MPa$^{-1}$ d$^{-1}$) the root radial hydraulic conductivity, $k_{soil,j}$ (m$^2$ MPa$^{-1}$ d$^{-1}$) the soil hydraulic conductivity in layer $j$, and $B_j$ (dimensionless) …"

L 181: The variable "n" should be briefly explained as one of the MVG parameters. Also, consider adding n and Sej to the list of variables.
We agree with the referee and made the suggested changes in the revised version of the MS (see list of variables and L205):

"m (dimensionless) and λ (dimensionless) are soil hydraulic parameters (with m = 1 – 2/n)"

L 209: Please define conditions for exudation. I believe it is for Sj<0, but not sure.

The referee is correct. This was clarified in the revised version of the MS (L246-247):

"… the depth at which the transition between nighttime water uptake and exudation ($S_{i,j}<0$, i.e. release of water from root to soil) takes place"

L 239: I do not see how the soil moisture varied notably at 1.3 m depth. What do you mean here? How comes that you refer to 12:00 and 20:00 on DaS 167, while that is not shown in Figure 2a?

Thank you, this was corrected and now reads (L266-267):

"Soil moisture remained unchanged in the top 25 cm during the sampling period ($\theta$ = 0.08 ±0.00 $m^3\ m^{-3}$) as well as at −1.30 m from DaS 166 - 15:45 to DaS 168 - 05:00 ($\theta$ = 0.33 ±0.01 $m^3\ m^{-3}$)."

L 243: Again, you refer to a time (7:30), which is not shown in the Figure and you should refer to it as soil labelled" and not "soil" to be consistent with Figure 2.

Indeed! This was also corrected (now L271):

"$\delta_{soil}$ reached a value of 36.9 ‰ at −1.50 m on DaS 167 - 17:00."

L 244: "lead us to assume" or "leads to the assumption"

Thank you. We now write (L272):

"…and the observed surface $\theta$ values lead us to assume…"

L 262: It is unclear which of the correlations are describe a significant relationship. I suggest to only draw the regression lines for significant relationships in Figure 4.

Thank you for this suggestion: we removed the regression lines for which the p-value of the linear model was lower than 0.01 and now indicate in the caption of Figure 4:

[Figure]

Figure 4. Correlations between measured variables: oxygen isotopic compositions of xylem and leaf waters ($\delta_{tiller}$ and $\delta_{leaf}$, in ‰), transpiration rate ($T$, in m d$^{-1}$), relative humidity (RH, %), and leaf water potential ($\psi_{leaf}$, in MPa). Coefficient of determinations ($R^2$) are reported for all data, and separately for 'day' data (gray symbols) and 'night' data (black symbols) (see Appendix C for definition of 'day' and 'night' experimental periods). Regression lines are drawn for linear models with p-value < 0.01.

L 281: replace "et" with "and"
Thank you ☺! Done.

L 298: Unclear what is meant with "over all dataset". I believe you mean the 60 different root system classes. Please be more specific.
That is right. We clarified the sentence as follows (L336-337):
"The predicted versus observed δtiller distributions including all plant groups and observation times differed noticeably but not significantly (…)".

L 315: It seems to me that in-situ measurements would overcome these limitations. One could sample in parallel several plants and thus, observe the temporal dynamics at individual plant level.
We could not agree more! We now mention these new methodological developments in a dedicated new subsection 3.3 "Progresses and Challenges in soil water isotopic labeling for RWU determination"

L 319: What is the expected accuracy of your volumetric soil moisture measurements. Given that you derived this from gravimetric water content and a bulk density, which was assumed to be constant in the repacked soil. However, relatively small differences in bulk density of just a few g cm$^{-3}$ will affect the estimates of the volumetric water content. It would be good to account for such uncertainties in this discussion.

We now add information on the validity of our assumption that the bulk density was constant in the soil profile L160-162:

> "The hypothesis of a constant value for $\rho_b$ across the reconstructed soil profile was further validated from the quality of the linear fit (coefficient of determination $R^2$ = 1.0) between the $\theta$ values measured by the sensors at the six available depths and (–0.05, –0.10, –0.30, –0.60, – 1.05 and –1.30 m) and those computed from $\theta_{grav}$."

Yet, the impact of observational errors were investigated as a sensitivity analysis in the revised MS, and did not yield any qualitative change in our results (see new section 3.2.3).

L 325: What do you mean with "significantly higher"? Did you apply a statistical test? I believe that you mean that the difference is higher than the measurement uncertainty.
Yes, the p-value is 1.4e-04. It was clarified in the revised version of the MS (L366).

Figure 6b: The title says "standard error", but the caption says "standard deviation". Which one is it? Please correct.
It is standard deviation, thank you for spotting this typo. We have updated Figure 6b accordingly.

L 360: the upper half of the soil profile
Done (now L438)

L 367: "water addition is localized and not broadcasted in the soil" is unclear. What do you mean with "broadcasted"?
We don't use the term "broadcasted" anymore and write instead (L491-493):

> "This case study highlights (i) the potential limitations of water isotopic labeling techniques for studying RWU: the soil water isotopic artificial gradients induced from water addition result in an improvement in RWU profiles determination to the condition that they are properly characterized spatially and temporally."

L 370: "simple"? In addition to the usual struggle of assessing meaningful MVG parameters to describe the soil water transport, also like for example Lpr and Kaxial are needed, which are not easily derived, but its estimation adds to the uncertainty of the uptake depths.
We meant "simple soil-root model", relative to (i) complex soil-root models, which include more parameters (e.g. profile of root hydraulic properties changing with root segment age, etc.), and (ii) absent soil-root models, in the typical Bayesian approach. We clarified that more measurements are needed than with no soil-root model. Extra measurements could be limited if appropriate assumptions on the model parameters can be done (e.g. using soil pedotransfer functions, root hydraulic properties reported in the literature, etc.). The revised text now reads (L495-498):

> "… calls for the use of simple soil-root models (though requiring additional water status measurements, as compared to the traditional Bayesian approach, and making simplifying assumptions in the description of the soil-plant system more explicit) for inversing isotopic data and gain insights into the RWU process."

**Referee #2: Dr. Matthias Beyer**

The manuscript hess-2019-543 'Disentangling temporal and population variability in plant root water uptake from stable isotopic analysis: a labeling study' by Couvreur et al. present a lab-/field- and model-based study of root water uptake during an artificial tracer experiment, where the soil is wetted from below (as opposed to often, via irrigation). They support their isotope analysis by hydraulic measures in order to provide a holistic understanding of RWU.

The authors address the urgent and contemporary need for increasing the reliability of RWU models and improve the understanding root water uptake patterns. It has been often proposed to combine hydraulic, water isotope and other information in order to do so, the presented study is in my opinion a holistic and promising approach. The rollercoaster vs. swarm hypothesis is also a good idea, though (as the authors state themselves) it should be validated further. It is also great that both data from an experiment and modeling are provided, rather than only one of the two. This manuscript is well prepared, and the topic is highly relevant. The figures are suitable and well-explained.

I highly recommend this manuscript for HESS, though I have a number of comments/questions that might help to improve this manuscript further. In brief, a few general comments, which are all rather minor:
Dear Dr. Beyer, we thank you for the time you spent in carefully revising our manuscript! We hope that we have sufficiently addressed the issues you raised in our revised version.

- The discussion on hydraulic redistribution should be strengthened. Do the authors see a clear sign or not? I think strengthening this part would be of utmost interest for many people from the ecohydrological community.
This is indeed an important part of the discussion. In the revised manuscript, we clarified in section 3.2.2 that all measurable signs of hydraulic redistribution are positive (local increase of soil water content, local enrichment of water isotopic signature) and converge with independent simulated results (water exuded at the same time and location, at a rate compatible with measurements) to yield a robust "yes we think that hydraulic lift was happening at that time at that depth" (L377-379):
> "Therefore, all relevant measurements (local increase of soil water content, local enrichment of water isotopic signature) and simulation results ($S<0$, i.e. local water release from roots) clearly converge to the conclusion that hydraulic lift occurred in the vicinity of -0.9 m depth in the early morning of DaS 168."

Furthermore, in the new section 3.2.3, we show that these results are not sensitive to an observational error of +/- 0.02 $m^3$ $m^{-3}$ in the soil water content measured with the gravitational method.

- When reading the results and discussion section, I realized that there are very small differences discussed in the manuscript (e.g. 0,41 per mill, 1 per mil, etc....). I think it is necessary to think about uncertainties in that respect and really decide which of the differences are likely 'true' differences or simply within the variability/uncertainty.
We now discuss in further detail (please see our answers to your specific comments) the problematic of meaningfulness of our isotope data, i.e., whether these differences of isotopic compositions are the result of given processes or the mere translation of e.g., soil lateral heterogeneity. Importantly, the simulated isotopic enrichment below 0.5 ‰ remains despite the consideration of soil water content measurement errors of +/- 0.02 $m^3$ $m^{-3}$ (see new section 3.2.3). In the revised manuscript we also clarify with ANOVA analysis that the locally increased $\delta_{soil}$ at 0.9 m depth significantly differs from $\delta_{soil}$ at previous times (L365-366):
> "… (–6.2 ‰, a value significantly higher than –7.1 ‰ ± 0.1 ‰ at earlier times based on ANOVA analysis, $P<0.01$)…"

- I find the discussion of the physical experiment slightly too weak compared to the results drawn from the modeling (I also indicated this in the detailed comments below).

We hope that we strengthened the discussion of the experimental experiment (see answer to your specific comment to L247-249)

- The authors use δtiller, etc. without providing the water isotope (e.g.δtiller18O). I think this is important to clarify (it was only 18O used, correct?) starting with the symbol description. Why was only oxygen-18 used? (and not 2H in addition?)
For clarification, we now systematically add "oxygen" before "isotopic composition" throughout the manuscript as well as in the "List of variables with symbols and units" (Page 2). We also now write (L116):
"Data on soil and plant oxygen stable isotopic signature and hydraulic status were monitored for 34 hours".
We only measured the water $\delta^{18}O$ our IRMS ("Isoprep 18 - Optima, Fison, Great-Britain" L172-173) and not water $\delta^2H$ and $\delta^{18}O$ simultaneously with, e.g., a laser spectrometer for two reasons: (1) to the contrary of laser spectrometers, IRMS are not affected by the presence of volatile organic substances which should be present in the distillated water from soil and plant samples. (2) The added information on $\delta^2H$ profiles should not be discriminating for determination of RWU profiles as $\delta^2H$ remains constant in the lower half of the soil profile (mostly contributing to RWU) which is influenced by labeling.

- Will the model be made publicly available? It would be very interesting to apply the model with other datasets (e.g. some in situ datasets of joint soil and plant water isotopes)
We are indeed willing to make the code open source, as it may be useful to the scientific community working on such data. We will upload it as soon as the MS is accepted.

I wish the authors good luck and look forward to the final publication.
Greetings and best wishes, Matthias Beyer

Detailed comments:

- Abstract is very well written
Thank you ☺

- L.78/79: depends on how deep the groundwater table is. In thick unsaturated zones, often mixing of old water is also a reason. Further, over short time periods a seasonal pattern might persist in the soil
We agree! If soil water (and eventually groundwater) is replenished by rain events of which the isotopic compositions is highly dynamic in time, it can generally lead to issues of identifiability. This is why we added (L83-84):
"(i) the soil water isotopic composition depth gradient is strong *and monotonic (thus avoiding issues of identifiability)*"

- L.140: in oxygen-18 I guess? Could the authors please add this information?
Done. It now reads (L154-155):
"…the reservoir's water was labelled and its $\delta^{18}O$ measured at +470 ‰."

- L.149: Can the authors please add specifics on the extraction? (Extraction temperature and time for soil and plant samples, how was complete extraction assessed?)The community has been asking in many occasions to provide more transparency of extraction procedures; hence it would be appreciable to add this information.
This information is added L168-170:
"Water from plant (i.e., tillers and leaves) and soil samples were extracted by vacuum distillation (applied vacuum: 10–3 mbar) at temperatures of 60 and 90°C, respectively,…"
In addition, complete extraction was assessed…
"based on the comparison of sample weight loss during distillation and mass of collected distillated water" (L308-309)

- L.152/153: the loss of mass would also include evaporation; was this neglected (please clarify) [I see that this is mentioned later in the text, but perhaps better to clarify here]
Yes, thank you. It has been clarified (L174):
    "transpiration (m d$^{-1}$)" is now replaced by "evapotranspiration rate (in m d$^{-1}$)"

- L.163: literature- Chapter 2.4: The explanation and equations make sense to me, but for a detailed evaluation and/or comments on the equations a true modeler might be considered (e.g. M. Cuntz, Wingate/Ogee group)
This is true! We have already received a comprehensive review from referee #1 on the modeling aspects of our work which we hope to have properly addressed in our answer.

- Chapter 3.1.1: Figure 2 is mentioned first in the text, then Figure 1....hence, those might be switched Results/Discussion
We make reference to Fig. 1 at L187-188 (under section "2.4 Modeling of RWU and $\delta_{tiller}$"), thus before citing Fig. 2 (section 3.1.1).

- L.243-247 and Fig. 2: There were two soil moisture profiles measured, but only one is shown in Fig.2 (or is that averaged over the two?) I am not sure if that justification that no evaporation was present is sufficient, as the moisture profiles oscillate greatly and over one or two days the effect of evaporation might be minimal (which on the other hand supports the assumption ET=T). Still, evaporation is probably occurring (though at a low rate).
There were 1 profile taken per sampling time, thus four profiles are shown in Figure 2: DaS 166 - 15:45 (orange line), DaS 167 - 07:00 (blue), DaS 167 - 15:45 (red), and DaS 168 - 05:00 (black)
We agree with the reviewer that evaporation could have been partly the reason of the observed differences in water content at the soil surface across sampling times, the other reason being the lateral heterogeneity. We can only make the assumption that evapotranspiration = transpiration, assumption that we carefully mention, based also on the very negative values of soil water potential in shallow layers.

- L.247-249: Yes, but these three options should be discussed by the authors
We now strengthen the discussion at this point and added (L275-285):
    "The differences in soil water oxygen isotopic profile observed at the four different sampling dates were therefore either due to lateral heterogeneity (e.g., upper soil layers), to the soil capillary rise of labelled water from the reservoir (deep soil layers), or to the hydraulic redistribution of water through roots (to the condition that the isotopic composition of the redistributed water differs from that of the soil water at the release location). We note an isotopic enrichment of 1.0 ‰ of soil water observed on DaS 168 - 05:00 at – 0.9 m with respect to the mean δsoil value across previous sampling dates. This could partly be due to, e.g., upward preferential flow of labelled water from the bottom soil layers and therefore be the sign of the lateral heterogeneity of the soil. Another reason for this would be hydraulic redistribution of labelled water by the roots. It was however not possible to evaluate the relative importance of these three processes (lateral heterogeneity, capillary rise/preferential flow, and hydraulic redistribution) in the setting of the soil water isotopic profile since the physically-based soil-root model presented in section 2.4 does not account for soil liquid and vapor flow. This was also not the primary intent of the present study."

- L. 250: minimal minimum instead maximal maximum
Thank you. Done.

- L. 252: delete level
Done. We replaced "level" by "value".

- 3.1.2: Again, if results/discussion is mixed here, those differences and diurnal patterns should be discussed and explained here

We agree that section 3.1.2 (as well as 3.1.1) stays rather descriptive. It is the case because we choose to discuss both soil and plant isotopic data in section 3.1.3 by cross-comparing them with soil and plant hydraulic data.

- L.269: 'Rayleigh distillation corrections' – this is not explained in the methods. Could the authors provide details on these corrections and/or provide a citation?
We now add two references to these corrections and how they should be applied:

> "Galewsky, J., Steen-Larsen, H. C., Field, R. D., Worden, J., Risi, C., and Schneider, M.: Stable isotopes in atmospheric water vapor and applications to the hydrologic cycle, Rev. Geophys., 54, 809-865, doi:10.1002/2015rg000512, 2016."
> "Dansgaard, W.: Stable Isotopes in Precipitation, Tellus, 16, 436-468, doi:10.1111/j.2153-3490.1964.tb00181.x, 1964."

- 3.1.3: Well-written and explained
Thank you ☺

- L.298: yes, but still: 3 per mill is notable for 18O...
Indeed, we agree with this comment. We clarified that a difference of 2.9 ‰ between simulated and measured mean $\delta_{tiller}$ is notable, though relatively small compared to the datasets standard deviations (8.4 ‰) and to the isotopic ratio of the labelled water (470 ‰; non-labelled soil water isotopic ratio between -7.4 ‰ and 1.3 ‰). Statistically we could not systematically conclude that simulated and measured $\delta_{tiller}$ differed. By drawing randomly simulated $\delta_{tiller}$ in 3 plants at each time step (as in the measurements), comparing the overall distributions of measured and simulated pooled $\delta_{tiller}$ with an ANOVA analysis, and repeating the random drawings for all 40 observation times 100 times, measured and simulated $\delta_{tiller}$ distributions were not statistically different in 92% of drawings (P>0.01). We reformulated the sentence as (now L336-339):

> "The predicted versus observed $\delta_{tiller}$ distributions including all plant groups and observation times differed noticeably but not significantly (6.6 ± 8.4 ‰ and 3.7 ± 8.4 ‰, respectively) when pooling 3 simulated $\delta_{tiller}$ randomly at each observation time, as in measurements (p-value>0.01 in 92 cases out of 100 repeated drawings)"

- L.325: I am not sure if an 0,9 per mil increase is significant...were replicates taken for each soil depth? What is the std of those (-often this can be in that range already)....if no replicates were taken, this might be well within the uncertainty rather than a true increase
The observed $\delta_{soil}$ at the first three observation times are -7.17 ‰, -7.00 ‰, and -7.21 ‰. We confirm that it differs from -6.2 ‰ with an ANOVA analysis (P<0.01). This was clarified in the revised version of the manuscript (L365-366):

> "…(–6.2 ‰, a value significantly higher than –7.1 ‰ ± 0.1 ‰ at earlier times based on ANOVA analysis, P<0.01)…"

- L.327 depths instead heights
Done. Thank you!

- L.345 model instead models
Done.

- L. 360 upper half instead first half
Done. It now reads (L438):
> "In the upper half of the profile…"

- L.363-365: But couldn't this be implemented to the Bayesian approach via the construction of priors?
Thank you, this is a very keen remark. It is true that we decided to go for (Appendix E, L733-734):

"flat Dirichlet a priori rRWU distribution (i.e., rRWUJ=1/10)"

We were missing an explanation on why we did not implement the construction of priors. This is not done (L442-448):

"Note that the outcome of the statistical model may significantly depend on the definition of the a priori relative RWU profile. In the present study, we set it to follow a "flat" distribution (i.e., rRWUJ = 1/10, see Appendix E), in other word, each layer was initially defined to contribute equally to RWU. To the contrary of other studies (e.g., Mahindawansha et al., 2018), where the a priori rRWU profile was empirically constructed on basis of soil water content and root length density profiles, we decided not to further arbitrarily constrain the Bayesian model for the sake of comparison with the physically-based soil-root model."

Figure 2:
- it's 18O data shown, could this be added to the title (instead of only delta)...OK it's in the figure description, still...
We hope mention of "oxygen" in the title and now repeatedly throughout the manuscript has clarified this.

- why is matric potential 'calculated' shown if it was measured?
$\psi_{soil}$ was calculated on basis of $\theta$ data, and not directly measured. We now clarify this confusion by moving the mention of soil matric potential to section 2.2 (L141-142):

"It was used to compute the soil water matric potential ($\psi_{soil}$, in MPa) on basis of volumetric water content data during the present experiment."

In addition, we added "Measured" in Fig. 2's caption:

"Measured soil volumetric water content ($\theta$, panel a), oxygen isotopic composition ($\delta_{soil}$, panel b), and calculated soil matric potential ($\psi_{soil}$, panel c) profiles during the sampling period"

- Not sure if the inset graphic for the water content is helping the figure
Done. The inset was removed from Fig. 2 in the revised version.

**Anonymous Referee #3**

This manuscript compares two alternative modeling strategies for deriving the sink term (root water uptake) in a controlled ecotron experiment. Strategy 1 uses a simplified root water uptake model which however incorporates the main features of the three dimensional soil water flow, including hydraulic redistribution. The unknown model parameters are calibrated based on isotope data in the tiller and leaf water potentials. Strategy 2 derives root water uptake based on isotope data using Bayesian inference.

The authors find that the results between the two strategies diverge. They show that Bayesian inference yields unphysical fluxes. Based on the model results they conclude that spatial variation ("swarm-like") in tiller isotopic signal is misinterpreted as a strongly fluctuating time series, whereas it actually reflects the different rooting depths of plant individuals. Additionally, they argue that both the root water uptake model and the soil moisture time series suggest hydraulic lift, which cannot be captured by the Bayesian inference based on isotope data alone. Therefore, they conclude that the results obtained based on Bayesian inference could be due to an artifact.

This is a valuable contribution illustrating how sampling choice may affect the interpretation of isotope data. Especially the application of a straightforward process model for comparison with the Bayesian inference together with the dense measurements are extremely helpful to explain the shortcomings of deriving uptake profiles based on isotope data alone. The case is well argued and the methods are sound. I feel the manuscript has potential to making an impact and will find strong interest in the readership of HESS. The paper is mostly well structured, although I have some concerns with the Abstract and Methods section, as well as with some formulations (see below).

Dear reviewer, we thank you for the detailed list of specific comments, for which we hope you will find our answers satisfactory.

I have two general concerns, and a number of editorial remarks (below). The investigated case is a particular one, e.g. with a strong labelling pulse added below the rooting zone. This needs to be explicitly stated and the manuscript should discuss in which other situations such a strong influence of spatial variation is to be expected (and where it is not a concern).

To address this general concern, we added a section 3.3 entitled "Progresses and Challenges in soil water isotopic labeling for RWU determination" (L450-476):

"Often in the field, the vertical dynamics of both soil water oxygen and hydrogen isotopic compositions are not strong enough (or show convolutions leading to issues of identifiability) for partitioning RWU among different contributing soil water sources. As a consequence, we unfortunately cannot make use of the natural variability in isotopic abundances for deciphering soil-root transfer processes [Beyer et al., 2018; Burgess et al., 2000]. To address this limitation of the isotopic methodology, labeling pulses have been applied locally at different depths in the soil profile [e.g., Beyer et al., 2016] or at the soil upper/lower boundaries under both lab and field conditions by mimicking rain events [e.g., Piayda et al., 2017] and/or rise of the groundwater table [Meunier et al., 2017a; Kühnhammer et al., 2019].

After labeling, we are faced with two problems: (i) the labeling pulse might enhance RWU at the labeling location if the volume of added water significantly changes the value of soil water content. It therefore poses the question of the meaningfulness of the derived RWU profiles, and this independently from the model used (i.e., physically-based soil-root model or statistical multi-source mixing model). In other worlds: are we observing a natural RWU behavior of the plant individual or population or are we seeing the influence of the labeling pulse? Certainly a way to move forward is environmental observatories such as ecotron and field lysimeters [e.g., Groh et al., 2018; Benettin et al., 2018] that provide means to better constrain hydraulic boundary conditions and reduced their isotopic heterogeneity. They allow for a mechanistic and holistic understanding of soil-root processes from stable isotopic analysis.

Another topic of concern is (ii) the difficulty to properly observe in situ (1) the propagation of the labeling pulse in the soil after application and (2) the temporal dynamics of the plant RWU isotopic composition. Beyer and Dubbert [2019] presented a comprehensive review on recent isotopic techniques for non-destructive, online, and continuous determination of soil and plant water isotopic compositions [e.g., Rothfuss et al., 2013; Quade et al., 2019; Volkmann et al., 2016a] as alternatives of the widely used combination of destructive sampling and offline isotopic analysis following cryogenic vacuum extraction [Orlowski et al., 2016b] or liquid-vapor direct equilibration [Wassenaar et al., 2008]. These techniques have the potential for a paradigm change in isotopic studies on RWU processes to the condition that, e.g., isotopic effects during sample collection are fully understood.

The present study highlights the need not to "trust" our isotope data alone and always complement them by information on environmental factors as well as on soil and plant water status to go beyond the simple application of statistical models. This is especially the case in the framework of labeling studies where strong soil water isotopic gradients may induce strong dynamics of the RWU isotopic composition from a low variability of rooting depths."

The study suffers from lack of opportunity for validation: The heterogenous rooting depths cannot be measured in situ and therefore it remains a hypothesis. This is ok. But it requires diligent consideration of other assumptions of the model that may have had a similar effect on the model result. How about the inherent assumptions of a big leaf? Could individual differences in leaf development incur similar results? Those considerations need to enter more than now into the discussion. I propose adding a section dealing with the effect of inherent model assumptions.

We totally agree with the referee. Our analysis shows that the tiller water isotopic signature is very sensitive to rooting depth in this kind of labelling experiment, generating heterogeneity in the aforementioned signatures of the population which in many cases could be confused for temporal variability of tiller water isotopic signature and root water uptake depth. We think this is an important *in silico* result and clarified in the discussion that its experimental validation would necessitate to estimate the variability of rooting depth in situ, which is currently not possible. Future studies using transparent soils, as in Downie et al. (2012), and continuous determination of soil and plant water isotopic composition (e.g., Rothfuss et al., 2013; Quade et al., 2019; Volkmann et al., 2016) might take us one step closer to a validation. This is now discussed in the new section 3.3.

The assumption that all plants transpire at the same rate ("big leaf") pointed out by the referee was not discussed in the manuscript, though it would be an interesting piece of discussion. Non-uniform patterns of transpiration within the plant population would affect two of our simulated variables: the isotopic signature of the tiller water and the leaf water potential.

- In our analysis, we have shown that large temporal fluctuations of transpiration (Figure 3 panel b) barely affect temporal fluctuations of the isotopic signature of tiller water (continuous grey lines Figure 5 panel a). Hence, we expect that the spatial heterogeneity of transpiration, likely smaller than its temporal heterogeneity, would have an even smaller impact on tiller water isotopic signature heterogeneity. Given the low sensitivity of tiller water isotopic signature to transpiration rate, and the lack of data on transpiration rate spatial heterogeneity, we think it is not worth developing additional simulations to study the effect of this factor in this manuscript.

- Unlike the tiller water isotopic signature, leaf water potential turned out to be very sensitive to transpiration rate in our simulations (see temporal fluctuations of grey lines in Figure 5 panel c) and not very sensitive to root distribution (see small variations of leaf water potential across individuals in Figure 5 panel d). This high sensitivity of leaf water potential to transpiration suggests that in this setup the hydraulic conductance of the soil-root system limits shoot water supply more than the distribution of roots (Sulis et al., 2019). A consequence of the high sensitivity of leaf water potential to transpiration rate is that if transpiration rate was spatially heterogeneous, substantial deviations of the measured leaf water potential would have been observed, relative to the simulated "baseline" leaf water potential (i.e. leaf water potential in case of uniform transpiration rate across the plant population). Simulated baseline leaf water potentials (for uniform transpiration rates) are shown as grey lines in Figure 5 panel c, and measured leaf water potentials as a

green line in the same panel. We were positively surprised to find out that the simulated baseline leaf water potentials fit the measured temporal fluctuations of leaf water potential quite well under the assumption of uniform transpiration rate, despite the high sensitivity of leaf water potential to transpiration rate. This result reinforces the idea that transpiration rate was likely not spatially heterogeneous among the plant population.

In consequence, we think that transpiration rate was rather uniform among the plant population (so that the "big leaf" approach was justified), and therefore, the tiller water isotopic signature, whose sensitivity to transpiration rate is already very low, was likely not affected by transpiration rate heterogeneity. This piece of discussion was added in the revised manuscript (L423-431):

"Unlike the tiller water isotopic signature, leaf water potential turned out to be very sensitive to transpiration rate in our simulations (see temporal fluctuations of grey lines in Figure 5 panel c) and not very sensitive to root distribution (see small variations of leaf water potential across individuals in Figure 5 panel d). This suggests that in this setup the hydraulic conductance of the soil-root system limited shoot water supply more than the distribution of roots, as in Sulis et al. (2019). Simulated baseline (i.e. for uniform transpiration rates) leaf water potentials are shown as grey lines in Figure 5 panel c, and measured leaf water potentials as a green line in the same panel. The fact that they match well despite the high sensitivity of leaf water potential to transpiration rate, reinforces the idea that transpiration rate was likely not spatially heterogeneous among the plant population. Therefore, the tiller water isotopic signature, whose sensitivity to transpiration rate is already very low, was likely not affected by transpiration rate heterogeneity."

Detailed comments:

Abstract: Line 18-19: This sentence sounds vague, e.g. "semi-controlled" and "such variables", please formulate more specifically.
We list in the few sentences after (now L21-24) the monitored variables in question. We removed "semi-controlled" (indeed a vague term) and "such variables" from the text. It now reads (L20-21):
"In this study, a population of tall fescue (Festuca arundinacae cv Soni) was grown in a macro-rhizotron and monitored for a 34-hours long period following the oxygen stable isotopic ($^{18}O$) labeling of deep soil water"

Line 23-24: "results underlined the discrepancy.." At this point unclear what is meant
We now put emphasis on "temporal disconnection" instead of "discrepancy" (L24-26):
"While there were strong correlations between hydraulic variables as well as between isotopic variables, the experimental results underlined the partial disconnection between temporal dynamics of hydraulic and isotopic variables."

Line 29-30: The sentence starting with "The physical model.. "is difficult to understand, please reformulate
The sentence was clarified in the revised version of the manuscript (L31-33):
"The physical model thus explained the discrepancy between isotopic and hydraulic observations: the variability captured by $\delta_{tiller}$ was spatial and may not correlate with the temporal dynamics of $\psi_{leaf}$."

Line 33: "local increase..." this results is not stated earlier and at this position confusing.
The "local increase" was not mentioned earlier, indeed. We write now instead (L36-37):
"It further supported that concomitant increases of soil water content and isotopic composition observed overnight above the soil region influenced by the labeling were due to hydraulic lift."

Lines 35-62 List of variables. Some variables are missing, please complete. Also I propose erasing all the repetitions of "units of". Later in the paper, it will be useful to express volumetric water content as vol-% and I propose adding it here.

Done. We added the missing two variables, namely the soil hydraulic conductivity parameter ($\lambda$, dimensionless) and the soil relative water content ($Se_j$). In addition, repetitions of "in units of" were discarded from the list of variables. Instead, we added L39 headers to the table "Name", "Symbol", and "Units".

Introduction

Line 75-76: The description of the "mean value of...weighted by" is confusing, please rephrase
Done. The sentence now reads (L81-82):
> "equals the sum of the product between the soil water isotopic composition and relative contribution to RWU across plant water sources."

Material and Methods

Line 114: Please mention that CS616 is a time domain sensor (TDR). Also, reflectometer is a correct, but awkward term for soil moisture sensor. I propose using the latter, just to avoid confusion.
Done. It now reads (L124):
> "The rhizotron was equipped with two sets of CS616 time domain reflectometer (TDR) profiles"

Line 123: The description of the soil is confusing. Is District Cambisol a typo for "Dystric Cambisol". Otherwise, I am not aware what a District Cambisol would refer to, please explain. Besides, Cambisol refers to a soil in situ and after specific pedogenesis which is completely removed in your experiment. Maybe say "The soil originates from a xx Cambisol". Also, does the bulk density refer to the original soil, and is it required to be mentioned?
There is no typo here. But we agree that it was not properly formulated, and that the substrate which we filled the rhizotron with could not possibly be referred to as a "cambisol". Therefore we write now (L134):
> "The soil substrate originates from the Lp horizon of an agricultural field part of the Observatory of Environment Research (ORE), INRA Lusignan, France (0°60W, 46°250N) which is classified as District Cambisol (particle size distribution: sand 15%, silt 65%, clay 20%)."

Line 125: Add "layer" before "by"
We propose the following reformulation (L137-138):
> "450 kg of soil was filled in the rhizotron by 0.10 m increment a.."

Line 128: sols-PST55 sensors are missing above, where installation depths were mentioned. Add there. Please add where they are installed.
Line 130: "between its position and measured soil water content" unclear, please rephrase
The retention curves were determined in situ in the same type of macro-rhizotron during another experiment (at the same soil bulk density) of which the results were published by Meunier et al. (2017a). In order to clarify this, we do not mention the type of sensors used – which was indeed misleading – of we write (L138-141):
> "The closed-form soil water retention curve of van Genuchten (1980) was derived in a previous study by Meunier et al. (2017a) from synchronous measurements of soil water content and matric potential from saturated to residual water content (see Appendix B for its hydraulic parameters)."

Lines 132-136: Please shortly state: Were the plants watered? What was the lower boundary condition?
Done. We added the information (L149-151):
> "During a period of 165 day following seeding, the tall fescue cover was exclusively watered from the reservoir with local water in order to (i) keep the soil bottom layer (< –1.3 m) close to water saturation, and to (ii) not to disrupt the natural soil water $\delta^{18}$O profile."

Line 139: Do you mean "sides" instead of "slides"
Yes, exactly, many thanks for finding this typo!

Line 144: "three plants were sampled" - does it mean the entire plant or some leafs?
We sampled the entire plant. We now insert "whole" after "three" (L163)

Line 148-149 I believe you mean "from the atmosphere surrounding the rhizotron". Also, I am assuming the latter means the ambient air in the lab? Would be good to specify.
Thank you for these propositions! We now write (L166-167):
"In addition, air water vapor was collected from the ambient atmosphere surrounding rhizotron"

Line 162: "60 tall festucae root systems .." Why 60 plants?
This choice was arbitrary. We estimated that there were about 1500 plants per square meter in the rhizotron, so that there would be 300 plants on total in the experiment. Running simulations for 300 plants would have required a lot of computational resources. That is why we focused on a subset of 60 representative plants, that met our computational capabilities for the inverse modelling scheme. Each "representative plant" was called a "class" ("group" in the revised MS) of plants that is included in the simulations under the form of a "big root" and "big leaf", with root lengths and transpiration rate corresponding to 5 plants for each group. This is an important point, which we clarified in the revised manuscript (L184-191):
"The experimental setup included about 300 tall fescue plants. In order to limit the computational requirement in the inverse modelling loop, we only generated 60 virtual root systems whose rooting depths ranged from −1.30 to −1.60 m depth (based on our own observations and those of the litterature, e.g., Schulze et al., 1996; Fan et al., 2016) with the root architecture simulator CRootBox (Schnepf et al., 2018), so that the simulated RLD matched observations (Fig. 1a). In order to reach the right amount of plants, each root system was replicated 5 times, forming a "group". Each group was assumed to occupy one sixtieth of the total horizontal area, and considered as a "big root" hydraulic network (5 identical plants per "big root") with equivalent radial and axial hydraulic conductances (thus neglecting architectural aspects but accounting for each group's respective root length density profile)."

Eq. (1) Personally, I do not find this equation obvious. Please motivate the origin. Are there any other assumptions involved besides the big root one?
This equation is indeed not obvious. It was derived by Meunier et al. (2017) in its Appendix C. The reference was indeed missing. We added it to the revised manuscript (L193):
", as derived by Meunier et al. (2017),".

Line 173-174: "dimension of the domain..." I do not understand this statement
The horizontal domain of simulation typically has two dimensions (X and Y), but it some cases, the studied problem has an essentially radial dimension between bulk soil and root surface.
We simplified the text as follows (L196-197):
"with $B_j$ (dimensionless) a geometrical factor simplifying the horizontal dimensions into radial domains between the bulk soil and root surfaces,…".

Line 176-177: Sentence starting with "The averaged distance .." seems wrong. Maybe erase the last words?
The referee is correct. The reported sentence referred to $\rho$, not to the average distance between roots. We corrected the text as follows (L200-201):
"It can be deduced from the observed root length density (…)".

Eq (4): Se is not part of the List of variables, plus the S stated there refers to the sink term not saturation. Please use a different abbreviation.
Relative water content ($Se_j$) is both introduced in Eq (4) is defined in Eq. (5). The water "sink term" is introduced in Eq. (9).

Line 188: Could not the measured root length density profiles be used?

Not here because root lengths, and thus root system conductances, are group-specific. The bulk root length densities may not account for the fact that each group has its own root length and RLD at each depth.

Line 191: "were derived" unclear how this was derived? Also, where was k_axial in Eq.(7) taken from? Please explain. Ok, I learn later this was calibrated. Maybe mention this here.

We apologize for the confusion, and included this clarification at this point in the revised manuscript (L215-216):

"were calculated as equivalent "big root" specific axial conductance per root system group ($k_{axial}$, m$^4$ MPa$^{-1}$ d$^{-1}$, to be optimized by inverse modelling) as"

Line 191: "root system class" Unclear what is meant with "class".

See reply to comment on simulated 60 root systems.

Line 194: standard sink distribution is not a standard term and requires a bit more explanation to be convincing.

The term was defined in the following sentence. We moved it ahead for clarity.

Line 195-196: "potential difference between soil and leaf": You are dealing with a soil profile and a leaf canopy. Thus, where in the soil and leaf are you referring to. Please also translate to what this means for your experimental setup either here or in the discussion.

That is a good point. The leaf is a "big leaf", and the soil water potential used in the definition of Ksoil-root is the SSF-averaged bulk soil water potential. We clarified it in the text (L220-223):

"The variable *SSF* is the relative distribution of water uptake in each soil layer under vertically homogeneous soil water potential conditions (Couvreur et al., 2012), and $K_{soil\text{-}root}$ represents the water flow per unit water potential difference between the *SSF*-averaged bulk soil water potential and the "big leaf" (assuming a negligible stem hydraulic resistance, [Steudle and Peterson])".

Line 196: "assuming negligible stem conductance": Does this imply that all conductance / resistance happens in the root system? Is this a reasonable assumption?

As far as grass is concerned we think so. The main hydraulic resistances between the bulk soil and the leaf insertion (where leaf water potential is measured) being the drying soil and the root radial resistance (Steudle and Peterson, 1998). Reference added in the text (L223).

Line 202: "class" unclear

See previous reply about the 60 root systems.

Line 205: "where axial conductances" this comes too late, please move up.

It was in the right place, but we rewrote the sentence for clarification (L231-232):

"where $K_{soil-root}$ was assumed to control the compensatory RWU which arise from a heterogeneously distributed soil water potential, due to large axial conductances (Couvreur et al., 2012)".

Line 218: I propose moving the inverse modeling procedure out the appendix and add it to the main text. It is important information.

We think that the Material and methods are very dense already, and since the inverse modelling method is a state-of-the-art method in modelling, we think it would be better to leave its detailed description in the appendix.

Line 226: Not sure what is meant with "ten identified potential water sources" .. "10 distinct soil layers" Can you be more specific?

Done. We now give the 10 soil layers' upper/lower boundaries L252-255:

"These water sources were defined to originate from 10 distinct soil layers (0.00-0.03, 0.03-0.07, 0.07-0.15, 0.15-0.30, 0.30-0.60, 0.60-0.90, 0.90-1.20, 1.20-1.32, 1.32-1.37, and 1.37-1.44 m)…"

Results and discussion:

Lines 335-252: Small issue: Please add some paragraphs in this section.
Done. We now split the text into three paragraphs which refer to soil water content (§1), soil water oxygen isotopic composition (§2), and root length density (§3) profiles.

Line 281: Do you mean "and" instead of "et"
Done, thank you!

Line 304: With "all the population" do you mean all individuals?
Indeed. We replaced it by "all individuals" for clarification (L343-344).

Line 311-313: Sentence is difficult to understand, please rephrase.
We rephrased as follows (L350-352):
> "As no correlation could be expected between the drivers (the maximum rooting depth of the sample plants and canopy transpiration rate) our analysis explains the absence of correlation between $\delta_{tiller}$ and $\psi_{leaf}$ or transpiration rate".

Lines 329ff: Since there is repeatedly reference to increasing by xx% this may be strongly confusing. Better use vol-% to be on the safe side.
We would like to keep using the information of dimension for soil volumetric water content ($\theta$) rather than using a relative unit. We need this information for, e.g., explaining how we calculate $\theta$ from the soil gravimetric water content ($\theta_{grav}$, in kg kg$^{-1}$). We converted the vol% to m$^3$ m$^{-3}$ back in section 3.2.2. Thank you for pointing out these inconstancies!

Line 340 Lambda is not in the list of variables.
$\lambda$ has been added to the list of variables, thank you.

Also, this information is very compact, and difficult to understand. Please elaborate.
The following clarification was added (L384-387):
> "The optimal value of $k_{sat}$ was quite high (Table 1) but reportedly very correlated to $\lambda$ (i.e. soil unsaturated hydraulic conductivity is proportional to $k_{sat}$, but also to $Se^{\lambda}$ (van Genuchten, 1980)), so that the low value of the latter compensated the high value of the former, thus they should be considered as effective rather than physical parameters".

Lines 345ff: I strongly recommend bringing Fig E into the main text. It is discussed and seems therefore sufficiently important. Also, because this is one of the two alternative water uptake profiles which comparison is the main motivation of the manuscript.
Done. Figure E is now in the main document under "Figure 7".

Line 360: Replace "first" with "top"
Done. Thank you.

Line 367: Not sure what is meant with "broadcasted"
We don't use the term "broadcasted" anymore and write instead (L491-493):
> "This case study highlights (i) the potential limitations of water isotopic labeling techniques for studying RWU: the soil water isotopic artificial gradients induced from water addition result in an improvement in RWU profiles determination to the condition that they are properly characterized spatially and temporally."

Figure 2: I was confused about the positive water potentials. Since they were named psi I was instinctively assuming to see matric potential, but plotted are the water potentials. I propose renaming to h. If you want to stick to psi_soil because of psi_leaf(although I seriously think it would not be an issue), please obviously state the reference elevation to avoid this type of confusion.

Figure 2 shows the profiles of the log transformation of soil water matric potential $\psi_{soil}$ (i.e., not the soil hydraulic head, which you term $h$). $\psi_{soil}$ is always negative, therefore for the log transformation we must take "$-\psi_{soil}$". We do not mean by that that $\psi_{soil}$ was negative during the experiment.

[revised manuscript text omitted]

---

## Author Response (AR2)

Authors' answers to **Anonymous Referee #3**

The authors have made an effort to carefully address all the comments put forward by the reviewers and
have improved the manuscript. In my opinion it is almost ready for publication, given some small
amendments.

*Dear reviewer, we thank you for reviewing our manuscript for the second time. You will find our*
*answers in line with your comments below:*

(title)
I am not convinced by the new title of the manuscript. The „rollercoaster" and the „swarm" are not sufficiently
self-explanatory acronyms to allow sense-making of the title. I strongly encourage finding a title that is catchy
because it speaks for itself. Why not including the terms „variation" and „rooting depth" into the title?

*We have changed the title to a version close to its original form, but integrating the important*
*keywords suggested by the referee:*
*"Disentangling temporal and population variability in plant root water uptake from stable*
*isotopic analysis: when rooting depth matters in labeling studies"*

(abstract)
The abstract still includes some sentences that are difficult to understand without having read the paper. I
propose properly rephrasing. Specifically -
Line 32-33: „the variability captured by δ_tiller was spatial and may not correlate with the temporal dynamics
of ψ_leaf."

*We attempted to make our point clearer and edited the sentence as (now L32-35):*
*"…the variability captured by $\delta_{tiller}$ reflected the spatial heterogeneity in rooting depth in the*
*soil region influenced by the labeling and may not correlate with the temporal dynamics of*
*$\psi_{leaf}$. In other words, the strong variations of RWU as deduced from isotopic changes in the*
*tiller water may not translate into significant variations of leaf water potential value."*

Line 36-37: The reference to hydraulic lift comes a bit out of the blue. I expected the last sentence of the
abstract to address further implications, but not a completely new topic. It is confusing. Can you
accommodate this?

*We now underline that the very last sentence in the abstract is not a new topic but illustrates another*
*fundamental difference between the physical and statistical model (now L38-40):*
*"An important difference between the two types of RWU models was the ability of the*
*physical model to simulate the occurrence of hydraulic lift in order to explain concomitant*
*increases of soil water content and isotopic composition observed overnight above the soil*
*labeling region."*

(List of variables)
Se_j is now added to list of variables. However, please adhere to the HESS guidelines which require variable
abbreviations in equations to be one letter only. Therefore, the abbreviation Se should be changed.

*Thank you. The symbol "Se" is standard for soil relative water content in vadose zone hydrology,*
*so we modified it into "$S_{e,j}$" with "e" part of the subscript so that the main symbol "S" is one letter as*
*requested by HESS guidelines. We therefore adapted the text in section "2.4 Modeling of RWU and*
*$\delta_{tiller}$" (now L206-211):*

The soil hydraulic conductivity function of Mualem [1976] and van Genuchten [1980] was
used:

$$k_{soil,j}(t) = k_{sat} \cdot S_{e,j}{}^{\lambda}(t)\left(1-\left(1-S_{e,j}^{\frac{1}{m}}\right)^{m}\right)^{2} \qquad (4)$$

where (…) $S_{e,j}$, the relative water content (dimensionless), is computed from the saturated
($\theta_{sat}$, m3 m–3) and residual ($\theta_{res}$, m3 m–3) water contents as:

$$S_{e,j} = \frac{\theta_j - \theta_{res}}{\theta_{sat} - \theta_{res}} \qquad (5)$$

(Methods)
Line 188: „the right amount of plants" - do you mean „the right number of plants" ? Also, it is unclear what is
meant with „right". Could you please specify?
Thank you for spotting this point that requires clarification. In the revised version of the text, we
rephrased the sentence as follows (now L190-191):
"In order to reach a total number of virtual plants representative of the number of plants in the
experimental setup, each root system was replicated 5 times, forming a "group"."

Line 222: In the response to the review 3 it says the new text would read "big leaf", but in the new manuscript
version it only says „leaf". The „big leaf", as in the response to reviewer 3, would be better.
Thank you for noticing this oversight on our behalf. In the revised version of the text, we used the
term "big leaf" as in the reply to the referee (now L224-226):
"… $K_{soil\text{-}root}$ represents the water flow per unit water potential difference between the SSF-averaged
bulk soil water potential and the "big leaf" (assuming a negligible stem hydraulic resistance [Steudle and
Peterson, 1998]."

Figure 2:
I think panel c would be much easier to read, if the x-axis was simply log-scale instead of plotting log(-
\psi_soil)
You are right! This is done. The new Figure 2 is displayed below:

[revised manuscript text omitted]